# Mechanism Design for LLM Fine-tuning with Multiple Reward Models

**Haoran Sun**[1], **Yurong Chen**[2]*, **Siwei Wang**[3], **Xu Chu**[1], **Wei Chen**[3], **Xiaotie Deng**[1]*

[1] CFCS, School of Computer Science, Peking University
[2] Inria, École Normale Supérieure, PSL Research University
[3] Microsoft Research Asia

sunhaoran0301@stu.pku.edu.cn, yurong.chen@inria.fr,
{chu_xu, xiaotie}@pku.edu.cn, {siweiwang, weic}@microsoft.com,

## Abstract

Fine-tuning large language models (LLMs) to aggregate multiple preferences has attracted considerable research attention. With aggregation algorithms advancing, a potential economic scenario arises where fine-tuning services are provided to agents with different preferences. In this context, agents may benefit from strategically misreporting their preferences, but this could harm the aggregation performance. This paper addresses such incentive issues by framing it as a mechanism design problem: an LLM provider determines the fine-tuning objective (training rule) and the pricing scheme (payment rule) for agents. We primarily focus on training rules that maximize social welfare subject to certain regularizations, referred to as SW-Max rules. First, we show that under most circumstances, truthful reporting is sub-optimal with simply a SW-Max rule, thereby highlighting the necessity of payments. Second, we extend the VCG payment to implement SW-Max rules in dominant-strategy incentive compatibility (DSIC). We characterize sufficient conditions for payment equivalence and derive the necessary conditions for a payment rule to implement a SW-Max rule in DSIC and other principles. Third, we demonstrate that our mechanism is approximately DSIC with perturbed input, showcasing its robustness against the inevitable errors in real-world applications. Experiments on real LLM training results further confirm the practical implications of our results.

## 1  Introduction

As large language models (LLMs) [61, 74] become increasingly widespread, users are seeking models that not only possess general capabilities but also align with their individual values. Reinforcement Learning from Human Feedback (RLHF) [14, 57] has emerged as a mainstream approach to achieve this alignment, where a reward model guides the reinforcement learning process using feedback signals that reflect human preferences.

However, standard RLHF becomes resource-intensive when catering to diverse preferences. Training separate LLMs for every individual or group within a community, each with unique preferences, is often impractical due to prohibitive computational costs and potential data privacy concerns. A more feasible alternative is to train a unified model that reflects collective values while still accommodating distinct needs. Multiple-Objective RLHF (MORLHF) [5, 78], which aims to efficiently integrate multiple preferences into a single model, offers a promising avenue for this. Further studies aim to improve MORLHF algorithms from various perspectives, including efficiency [41, 62, 70], accuracy [18, 26, 63, 82], and fairness [11].

---

*Corresponding Authors.

As these techniques advance, we explore a practical economic scenario: a platform offering a fine-tuning service to aggregate diverse preferences from various groups into a single LLM. These "groups"—such as different departments within a company or hospitals in the same city with various specializations—share the same core values but have slightly different focuses. Given these shared values and the high cost of fine-tuning, developing separate LLMs for each entity is often inefficient. Nevertheless, each group must provide its specific preferences to account for these differing focuses. Finally, the training cost is shared among the groups and can be non-uniform due to their differentiated preferences.

A critical issue in this process is that groups may *strategically misreport their preferences to manipulate* the aggregate objective for a more favorable outcome. As illustrated in a simplified RLHF framework (see Figure 1), a group's true preference ($rm_1$) could be misreported as a polarized one ($\widetilde{rm}_1$) to steer the model toward a more desirable outcome. However, this behavior distorts the training objective, resulting in a suboptimal model for the overall community. Given the potential profitability of such strategies and the growing economic importance of LLMs, ensuring truthful preference reporting is as critical as the training algorithm itself. We therefore formalize this scenario to study its incentives. *Our findings indicate that many commonly used training objectives lead to profitable misreporting strategies. However, we also demonstrate that a simple incentive-compatible cost allocation scheme can incentivize truthful reporting, and under certain conditions, this scheme is uniquely determined.*

Specifically, we model this as a multi-parameter mechanism design problem involving a fine-tuning service provider and multiple groups of agents. The mechanism consists of a *training rule*, which aggregates the reported sizes $w_i$ (representing a group's scale) and preferences from different groups, and a *payment rule* to determine their respective charges. The fine-tuning process is implemented through RLHF, with reward models representing the groups' preferences. Our focus is on training objectives aimed at maximizing social welfare with a regularization constraint, referred to as SW-Max training rules. Our technical contributions, which extend beyond standard mechanism design due to the unique complexities of LLM fine-tuning objectives, are summarized as follows:

1. *We show that mechanisms using only SW-Max training rules are vulnerable to profitable preference misreporting (Theorem 4.2 and Theorem 4.3).* This finding highlights the need for a payment rule to resolve incentive issues.

2. We extend the VCG payment to ensure truthfulness for SW-Max training rules (Proposition 4.4) and *further establish the uniqueness of this payment under certain conditions (Theorem 4.9 and Corollary 4.10).* Based on that, *we derive necessary conditions for payment rules to implement a SW-Max training rule in more principles (Theorem 4.11).*

3. *We demonstrate that our mechanism is approximately DSIC in the presence of input perturbations (Theorem 4.12).* This finding highlights the robustness of our mechanism against the inevitable measurement errors in real-world applications.

4. Experiments on practical LLM setups *empirically validate the existence of profitable misreporting strategies and demonstrate the efficacy of our mechanism in incentivizing truthful reporting* (Section 5).

**Related Work.**   Several recent studies have also examined incentive issues in RLHF and LLMs. Duetting et al. [25] proposed a preference aggregation mechanism that satisfies monotonicity with respect to bids; however, their work does not address strategic misreporting of preferences, which is the central challenge we tackle. Other works that consider strategic preference reporting have different focuses, such as implementing truthful rules with KL-divergence for ad auctions [71], analyzing the implementability of various training rules [59], or modifying the RLHF objective to achieve approximate truthfulness while preserving convergence [10]. In contrast, our work adopts a theoretical perspective to analyze representative training rules, providing a comprehensive understanding of incentive issues in RLHF. Specifically, our analysis of payment equivalence helps characterize *all* possible payment rules that implement a training rule in DSIC.

Our research also connects to classic literature on auction design [52–54] and facility location problems [21, 58]. Compared to the classic auction model, we have to consider the necessary regularization term, which makes the training rule (or the allocation rule in the auction) more complicated and prevents vanilla VCG from being applied. In facility locations, agents can benefit by misreporting a more polarized preference. The idea of such a strategy is similar to our model.

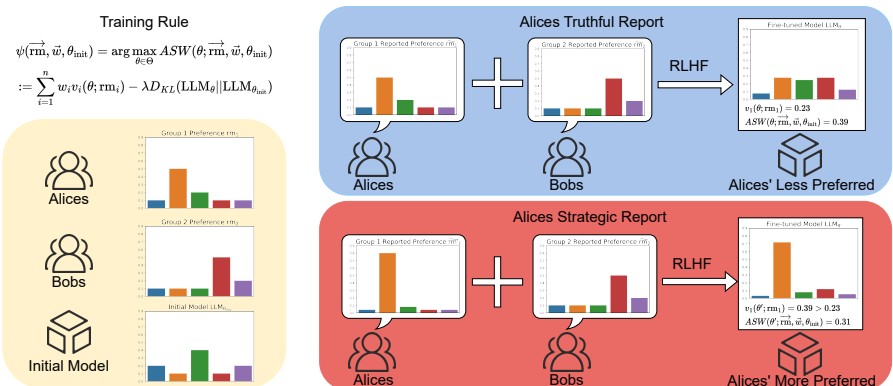

Figure 1: An illustration of the incentive issue in LLM preference aggregation. When using a basic training rule $\psi$ in RLHF for two groups (Alices and Bobs), fixing Bobs' report $\widetilde{\text{rm}}_2$, Alices can gain a higher utility by strategically reporting $\widetilde{\text{rm}}_1' \neq \text{rm}_1$ than truthfully reporting $\widetilde{\text{rm}}_1 = \text{rm}_1$. On the other hand, we have $\text{ASW}(\theta; \overrightarrow{\text{rm}}, \vec{w}, \theta_{\text{init}}) > \text{ASW}(\theta'; \overrightarrow{\text{rm}}, \vec{w}, \theta_{\text{init}})$, which means that such strategic behavior also harms the training objective.

However, due to the complexity of the training rules that aim to catch the LLM fine-tuning scenarios and the normalization constraints of the reward models, the reporting strategies can be more complex. Further, combined with the discretized input spaces of the agents, most of our results cannot be directly derived from existing literature.

**Paper Organization.** The remainder of the paper is organized as follows. Section 2 introduces the necessary preliminaries, and Section 3 formulates the RLHF Game. We then analyze the properties of mechanisms composed of SW-Max training rules and payment rules in Section 4, followed by a presentation of our experimental results in Section 5. Finally, Section 6 offers concluding remarks and discusses potential future research directions.

## 2 Preliminaries

**Large Language Models.** In this paper, LLMs are abstracted as stochastic mappings from a prompt set, denoted by $\mathcal{X}$, to a probability distribution over sequences of length up to $K$ in the output space [25]. Let $T$ represent the set of all tokens, and define $T^* := \emptyset \cup T \cup T^2 \cup \ldots \cup T^K$ as the set of sequences with lengths up to $K$. An LLM parameterized by $\theta$ is a function $\text{LLM}_\theta : \mathcal{X} \to \Delta(T^*)$. The space of LLM parameters is denoted by $\Theta$, and it is assumed that the LLM can express any function within this space. Our theoretical model operates on each prompt independently, so we focus on a fixed prompt scenario and omit its notation for simplicity. We denote $\text{LLM}_\theta(\boldsymbol{x})$ the probability of a sequence $\boldsymbol{x}$ generated by the model $\text{LLM}_\theta$.

**Reward Modeling.** In RLHF, a reward model is a function $\text{rm} : \mathcal{X} \times T^* \to \mathbb{R}$, which maps a prompt-response pair to a real number, indicating humans' satisfaction with the response based on the prompt. Similar to the LLM case, we focus on a fixed prompt scenario, so $\text{rm}(\boldsymbol{x})$ represents the scalar feedback for a response $\boldsymbol{x} \in T^*$. Following prior empirical work for RLHF [57, 78], we mainly consider two types of normalization constraints for the reward model: (1) The summation of the rewards over $T^*$ is normalized to 1, i.e. $\sum_{\boldsymbol{x} \in T^*} \text{rm}(\boldsymbol{x}) = 1$. (2) The maximum of the rewards over $T^*$ is normalized to 1, i.e. $\max_{\boldsymbol{x} \in T^*} \text{rm}(\boldsymbol{x}) = 1$. Furthermore, we also assume that the output rewards are all non-negative, i.e., $\text{rm}(\boldsymbol{x}) \geq 0$ for all $\boldsymbol{x} \in T^*$. The set of all reward model functions satisfying these conditions is denoted by $\mathcal{R}$. Unless otherwise specified, the results in this paper hold under both normalization schemes.

## 3 Formulation of the RLHF Game

In this section, we present the formal description of the RLHF Game. The game involves one LLM *provider* and $n$ *groups of agents*, denoted by $[n] = \{1, 2, \ldots, n\}$. The provider has an initial model

$\text{LLM}_{\theta_{\text{init}}}$ with positive probability for all sequences, i.e., $\text{LLM}_{\theta_{\text{init}}}(\boldsymbol{x}) > 0$ for all $\boldsymbol{x} \in T^*$. Each group $i$ has $w_i$ agents who share the same preference represented by a reward model $\text{rm}_i$. Let $\mathcal{R}$ and $\mathcal{W} \subseteq \mathbb{N}_+$ denote the domains for each group's reward model and group size, respectively. The group size $w$ should be an integer, and we assume an upper bound $\bar{w}$ for $\mathcal{W}$, which is public information. The exact reward model $\text{rm}_i$ and the size $w_i$ are group $i$'s private information. For an agent in group $i$, the valuation when it receives a model $\text{LLM}_\theta$ is denoted by $v_i(\theta; \text{rm}_i)$, defined as follows.

**Definition 3.1.** An agent's valuation of model $\text{LLM}_\theta$ is its expected reward on the sequences generated by it: $v(\theta; \text{rm}) = \mathbb{E}_{\boldsymbol{x} \sim \text{LLM}_\theta} \text{rm}(\boldsymbol{x}) = \sum_{\boldsymbol{x} \in T^*} \text{LLM}_\theta(\boldsymbol{x}) \text{rm}(\boldsymbol{x})$.

In practice, this can be obtained by averaging the reward of the sequences sampled from an LLM. We also discuss the influence of possible errors in this process in Section 4.3.

**Remark on the group size $\vec{w}$.**   We introduce the concept of group size to ensure that our model encompasses a broader range of scenarios. As the scales of different groups may vary, our training objective has to account for this factor to ensure fairness. Groups are also allowed to over-report their sizes to attain a higher status in fine-tuning. The case $\vec{w} = 1$ represents a special scenario where each group consists of exactly one agent and is included in our general model. In certain results, we note that the general model is technically more difficult than the $\vec{w} = 1$ case.

The provider first announces the mechanism, including a training rule $\psi$ and a payment rule $p$,

$$\psi : \mathcal{R}^n \times \mathcal{W}^n \times \Theta \to \Theta, \qquad\qquad p : \mathcal{R}^n \times \mathcal{W}^n \times \Theta \to \mathbb{R}^n.$$

Both rules take $n$ reported reward models, $n$ reported sizes, and an initial model as input and output the objective fine-tuned model and each group's payment, respectively. The provider can choose not to charge the users by setting $p$ always equal to $0$. In this case, the model coincides with most previous work on designing empirical algorithms, where agents' incentives are not considered [18, 26, 41, 63, 76, 78, 82]. Specifically, the training rule seeks the model that maximizes a certain objective function OBJ. That is, $\psi(\overrightarrow{\text{rm}}, \vec{w}, \theta_{\text{init}}) \in \arg\max_{\theta \in \Theta} \text{OBJ}(\theta; \overrightarrow{\text{rm}}, \vec{w}, \theta_{\text{init}})$, with ties broken based on further ordering of $v_i(\theta; \text{rm}_i)$s.

After observing the announced mechanism $(\psi, p)$, each group $i$ reports a reward model, $\widetilde{\text{rm}}_i$, and its group size $\tilde{w}_i$. Based on the reported information, the provider fine-tunes the model and gets the final model with parameter $\theta_{\text{final}} = \psi(\overrightarrow{\widetilde{\text{rm}}}, \vec{\tilde{w}}, \theta_{\text{init}})$. Each member in the group has access to the fine-tuned model, so the valuation for group $i$ is $w_i v_i(\theta_{\text{final}}; \text{rm}_i)$. The provider then charges each group $i$ a one-time payment according to the payment rule, $p_i(\overrightarrow{\widetilde{\text{rm}}}, \vec{\tilde{w}}, \theta_{\text{init}})$. All groups have quasi-linear utilities, i.e., group $i$'s utility is the valuation it attains minus the payment:

$$u_i(\overrightarrow{\widetilde{\text{rm}}}, \vec{\tilde{w}}; \psi, p, \text{rm}_i, w_i) := w_i v_i(\theta_{\text{final}}; \text{rm}_i) - p_i(\overrightarrow{\widetilde{\text{rm}}}, \vec{\tilde{w}}, \theta_{\text{init}}).$$

The groups may strategically report, thus $\overrightarrow{\widetilde{\text{rm}}}$ and $\vec{\tilde{w}}$ do not necessarily equal the true $\overrightarrow{\text{rm}}$ and $\vec{w}$. The LLM provider's goal is to achieve its training objective based on the group's true preferences, taking into account that misreporting may distort the training outcome. To this end, it is crucial to incentivize all groups to report their information truthfully so that the provider has access to the groups' private information. These desiderata for the mechanism are formally defined as follows.

**Definition 3.2.** A mechanism $(\psi, p)$ satisfies *dominant-strategy incentive compatibility* (DSIC) if $\forall i$, $\text{rm}_i, w_i, \text{rm}_i', w_i', \overrightarrow{\text{rm}}_{-i}, \vec{w}_{-i}, \theta_{\text{init}}$, we have

$$u_i((\text{rm}_i, \overrightarrow{\text{rm}}_{-i}), (w_i, \vec{w}_{-i}); \psi, p, \text{rm}_i, w_i) \geq u_i((\text{rm}_i', \overrightarrow{\text{rm}}_{-i}), (w_i', \vec{w}_{-i}); \psi, p, \text{rm}_i, w_i). \quad \text{(DSIC)}$$

**Definition 3.3.** A mechanism $(\psi, p)$ satisfies *individually rationality* (IR) if $\forall i$, $\text{rm}_i, w_i, \overrightarrow{\text{rm}}_{-i}, \vec{w}_{-i}, \theta_{\text{init}}$, we have

$$u_i((\text{rm}_i, \overrightarrow{\text{rm}}_{-i}), (w_i, \vec{w}_{-i}); \psi, p, \text{rm}_i, w_i) \geq 0. \quad \text{(IR)}$$

DSIC means that truthfully reporting the reward model and the group size yields the highest utility for any group, regardless of other groups' reports. IR means that truthfulness always yields non-negative utilities. When a mechanism $(\psi, p)$ satisfies DSIC, IR, or both DSIC and IR, we say that the payment rule $p$ *implements* $\psi$ in DSIC, IR, or both DSIC and IR. When we say the implementability of a training rule, we refer to the property of DSIC.

# 4 Incentives in the RLHF Game

This section explores incentive design within the RLHF Game framework. Our focus is mainly on a set of training rules that aims at maximizing social welfare with regularization, which balances efficiency and fairness and is commonly used in practice to aggregate various preferences [8, 56]. Denote $D_f(p||q) := \mathbb{E}_{q(\boldsymbol{x})} f(p(\boldsymbol{x})/q(\boldsymbol{x}))$ the divergence between probability distributions $p$ and $q$ measured by function $f$, the formal definition follows.

**Definition 4.1** (SW-Max Training Rules). A Social Welfare-Maximizing training rule fine-tunes the model to maximize the summation of the groups' valuations subject to a regularization measured by $f$-divergence [3, 19, 70]. Formally, the training objective is

$$\text{OBJ}(\theta; \overrightarrow{\text{rm}}, \vec{w}, \theta_{\text{init}}) = \text{ASW}(\theta; \overrightarrow{\text{rm}}, \vec{w}, \theta_{\text{init}}) := \sum_{i=1}^{n} w_i v_i(\theta; \text{rm}_i) - D_f(\text{LLM}_\theta || \text{LLM}_{\theta_{\text{init}}}),$$

where $f$ is convex on $\mathbb{R}_+$ and $f(1) = 0$. We use $\text{ASW}(\theta; \overrightarrow{\text{rm}}, \vec{w}, \theta_{\text{init}})$ to denote the affine social welfare.

This defines a set of training rules, and the function $f$ includes the most commonly used regularization terms in training a model. For example, $f(x) = \lambda x \log x$ refers to KL-divergence, $f(x) = \lambda(x-1)^2$ refers to $\chi^2$ divergence, $f(x) = \lambda|x-1|$ refers to total variation. We denote $\psi \in \Psi^{SW}$ that $\psi$ belongs to this set.

In the following subsections, we will first establish the necessity of a payment rule for SW-Max training rules. Then, we construct DSIC mechanisms for these training rules using affine maximizer payments and demonstrate payment equivalence properties for certain distance measures $f$. Next, we address the influence of noise input on the DSIC property. Finally, we discuss the efficient implementations of the mechanisms in practice.

## 4.1 Necessity of Payment Rule

We start by showing that without payment rules, groups have incentives to misreport their preferences under most circumstances. Our discussion focuses on strategies other than simply inflating the group size $w_i$. We assume that for $\forall \overrightarrow{\text{rm}}, \vec{w}, \theta_{\text{init}}$, the fine-tuned model $\theta = \psi(\overrightarrow{\text{rm}}, \vec{w}, \theta_{\text{init}})$ satisfies that $\text{LLM}_\theta(\boldsymbol{x}) > 0$ for $\forall \boldsymbol{x} \in T^*$. This mainly excludes extreme cases where the outcomes remain largely unchanged regardless of input, which may make the analysis meaningless. Based on this, we comprehensively analyze the relationship between optimal strategy and truthful reporting. We start with two cases with strong intuition.

**Theorem 4.2.** *In the RLHF Game with mechanism $(\psi, p)$ that $\psi \in \Psi^{SW}$ and $p \equiv 0$, for group $i$, define $s_i := |\{r | r = \text{rm}_i(x), x \in T^*\}|$ and $\underline{\text{rm}}_i := \min_{\boldsymbol{x} \in T^*} \text{rm}_i(\boldsymbol{x})$:*

1. *If $s_i = 1$, truthfully reporting is the optimal strategy regardless of other groups' reports.*

2. *If $s_i \geq 2$ and $\underline{\text{rm}}_i > 0$, there is a strategy that yields strictly higher utility than truthfully reporting regardless of other groups' reports.*

$s_i = 1$ is an unusual case in which group $i$ has the same preference values for all $\boldsymbol{x}$, resulting in the same valuation for any model $\theta$. In such a case, all strategies bring the same utility and hence are optimal. However, when $s_i \geq 2$ and $\underline{\text{rm}}_i > 0$, group $i$ can report $\text{rm}_i'$ that assigns a lower value to $\boldsymbol{x}_1 = \arg\min_{\boldsymbol{x} \in T^*} \text{rm}_i(\boldsymbol{x})$ (and a larger value to $\boldsymbol{x}_2 = \arg\max_{\boldsymbol{x} \in T^*} \text{rm}_i(\boldsymbol{x})$ in summation normalization). By doing so, group $i$ pretends to prefer $\boldsymbol{x}_1$ less, thereby increasing the likelihood that the resulting fine-tuned model generates the outcomes it prefers more. The condition $\underline{\text{rm}}_i > 0$ ensures that group $i$ is not completely uninterested in any $\boldsymbol{x}$, which is more realistic in practice.

Further, we consider the case that $s_i \geq 2$ and $\underline{\text{rm}}_i = 0$. Since the minimum value is already 0, the strategy above cannot be applied. We need to analyze in more detail how the training results change when one group adjusts its reported preferences. Under certain smoothness conditions of the function $f$, we derive a function $t(\boldsymbol{x})$ to estimate the gradient of the valuation for group $i$ over the reported value $\text{rm}_i(\boldsymbol{x})$. Based on this function, we show that if $t(\boldsymbol{x}) \neq 0$ for some $\boldsymbol{x}$, it is always possible to find a suitable direction and magnitude to report $\text{rm}_i'(\boldsymbol{x}) \neq \text{rm}_i(\boldsymbol{x})$, allowing group $i$ to achieve higher utility. The result is summarized in the following theorem. Due to the complicated form of the function $t$, we provide a detailed version in the Theorem B.2.

**Theorem 4.3** (Simplified version of Theorem B.2). *In the RLHF Game with mechanism $(\psi, p)$ that $\psi \in \Psi^{SW}$ and $p \equiv 0$, when $f$ is strongly convex and $C^2$-smooth, there exists a function $t$, when $t(\boldsymbol{x}, \overrightarrow{rm}, \vec{w}, \theta_{init}) \neq 0$ for some $\boldsymbol{x} \in T^*$, truthfully reporting is not the optimal strategy.*

The properties of $f$ stated in Theorem 4.3 are also considered in optimization theory [48] and encompass a wide range of divergence measures. Combining Theorem 4.2 and Theorem 4.3, we provide a comprehensive analysis that covers the entire space of $s_i$ and $\underline{rm}_i$. While the second theorem offers only a sufficient condition for the suboptimality of truthful reporting, we demonstrate in the proof that *this condition is highly likely to occur*, illustrating the impossibility of a mechanism that aims to maximize social welfare to incentivize truthfulness without payments.

## 4.2 Affine Maximizer Payment

After establishing the necessity of payment rules in this scenario, we mainly address two questions in this part:

1. Given a training rule $\psi$, can we find a payment rule $p$ such that the mechanism $(\psi, p)$ satisfies DSIC? This is the so-called implementability of a training rule $\psi$.

2. For an implementable training rule $\psi$, can we identify the relationship between the payment rules $p$s among all DSIC mechanisms $(\psi, p)$.

For the first question, since there is an additional regularization term, we can not directly apply the vanilla VCG payment [15, 34, 75] to the SW-Max training rules. To address this problem, we define $\text{ASW}_{-i}(\theta; \overrightarrow{rm}, \vec{w}, \theta_{init})$, the affine social welfare function that excludes the contribution of group $i$ from the social welfare:

$$\text{ASW}_{-i}(\theta; \overrightarrow{rm}, \vec{w}, \theta_{init}) := \text{ASW}(\theta; \overrightarrow{rm}, \vec{w}, \theta_{init}) - w_i v_i(\theta; rm_i).$$

Then, the vanilla VCG payment can be generalized to the following form, which is also known as the affine maximizer payment rule [64] $p^{AFF}$:

$$p_i^{AFF}(\overrightarrow{rm}, \vec{w}, \theta_{init}) = \text{ASW}_{-i}(\psi(\overrightarrow{rm}_{-i}, \vec{w}_{-i}, \theta_{init}); \overrightarrow{rm}, \vec{w}, \theta_{init}) - \text{ASW}_{-i}(\psi(\overrightarrow{rm}, \vec{w}, \theta_{init}); \overrightarrow{rm}, \vec{w}, \theta_{init}).$$
$$(1)$$

Following the proof of the classic VCG mechanism, we show that $p^{AFF}$ implements SW-Max training rules in both DSIC and IR, implying that truthfully reporting both reward models and group sizes constitutes a dominant Nash Equilibrium under this mechanism.

**Proposition 4.4.** *For any $\psi \in \Psi^{SW}$, mechanism $(\psi, p^{AFF})$ satisfies DSIC and IR, and the payment is non-negative.*

The availability of the affine maximizer payment derives from the additive property of SW-Max training rules. However, this method does not apply to training rules where the objective function cannot be decomposed into additive components, such as Nash Social Welfare and the fairness-oriented objective defined in MaxMin-RLHF [11]. The implementability of an arbitrary training rule is characterized by the concept of cycle monotonicity, which is discussed in Section E but is not the focus of this paper.

The second question is more general, so we consider the concept of *payment equivalence* [4] as a bridge, which is defined as:

**Definition 4.5** (Payment Equivalence). An implementable training rule $\psi$ satisfies payment equivalence if for any two mechanisms $(\psi, p)$ and $(\psi, p')$ satisfying DSIC, there exists a function $g_i$ such that for $\forall rm_i \in \mathcal{R}, w_i \in \mathcal{W}$

$$p_i'(\overrightarrow{rm}, \vec{w}, \theta_{init}) = p_i(\overrightarrow{rm}, \vec{w}, \theta_{init}) + g_i\left(\overrightarrow{rm}_{-i}, \vec{w}_{-i}, \theta_{init}\right).$$

Or equivalently, when fixing $\overrightarrow{rm}_{-i}$, $\vec{w}_{-i}$ and $\theta_{init}$, there is a constant $c$ such that $p_i'(rm_i, w_i) = p_i(rm_i, w_i) + c$ for all $rm_i \in \mathcal{R}, w_i \in \mathcal{W}$.

Payment equivalence indicates that the only way to modify a mechanism $(\psi, p)$ to $(\psi, p')$ while maintaining the property of DSIC is to add a term that is independent of $i$'s report to group $i$'s payment function $p_i$. Thus, the payment equivalence of $\psi$ is sometimes interpreted as the uniqueness of the payment rule $p$ that implements it in DSIC. This notion is particularly useful in the case that

we can figure out a certain DSIC mechanism $(\psi, p)$ for $\psi$ because any other payment rules $p'$ that also implement it in DSIC can be divided into $p$ and an independent part.

In the context of the RLHF Game, the domain of the reward models and group sizes affects payment equivalence. When $\vec{w} \equiv 1$, groups only report reward models, with the domain $\mathcal{R}$ containing all normalized reward models rm. Since this forms a connected set in Euclidean space, we can apply the result from Nisan et al. [55] to show:

**Proposition 4.6.** *When $\vec{w} \equiv 1$ is public information, and the agents only report the reward models, all implementable training rules satisfy payment equivalence.*

However, when the group size $\vec{w}$ is also a part of the private information for all groups, *the domain of the whole private information becomes $\mathcal{R} \times \mathcal{W}$ that is no longer a connected set because $\mathcal{W} \subseteq \mathbb{N}_+$.* To get a more meticulous characterization of the property, we define the continuity of a training rule.

**Definition 4.7** (Continuous Training Rule)**.** A training rule $\psi$ is continuous if for any $\epsilon > 0$, there exists a $\delta > 0$ such that for any $\theta_{\text{init}}$, $\overrightarrow{\text{rm}}$, $\overrightarrow{\text{rm}}'$, $\vec{w}$ and $\vec{w}'$, if $\max_{\boldsymbol{x} \in T^*} |\sum_{i=1}^{n} (w_i \text{rm}_i(\boldsymbol{x}) - w_i' \text{rm}_i'(\boldsymbol{x}))| \leq \delta$, then $\max_{\boldsymbol{x} \in T^*} |\text{LLM}_\theta(\boldsymbol{x}) - \text{LLM}_{\theta'}(\boldsymbol{x})| \leq \epsilon$, where $\theta := \psi(\overrightarrow{\text{rm}}, \vec{w}, \theta_{\text{init}})$ and $\theta' := \psi(\overrightarrow{\text{rm}}', \vec{w}', \theta_{\text{init}})$.

The continuity requests that the training outcome be similar if the reported values are similar. This definition is natural, and we identify several continuous SW-Max training rules.

**Proposition 4.8.** *SW-Max training rules with regularizations KL-divergence, $f_{\text{KL}}(x) = \lambda x \log x$, and $\chi^2$ divergence, $f_2(x) = \lambda(x - 1)^2$ ($\lambda > 0$ is a constant) are continuous.*

Based on the continuity, we show a sufficient condition of payment equivalence for general training rules.

**Theorem 4.9.** *An implementable training rule $\psi$ satisfies payment equivalence if it is continuous and for $\forall i$, $\overrightarrow{\text{rm}}_{-i}$, $\vec{w}_{-i}$, $\theta_{init}$ there exists $rm_i^*$ and $\theta$ such that $\psi((rm_i^*, \overrightarrow{rm}_{-i}), (w_i, \vec{w}_{-i}), \theta_{init}) \equiv \theta$ for all $w_i \in \mathcal{W}$. In the maximum normalization case, $rm_i^*$ must be $\mathbb{1}$.*

We provide some intuitions of the theorem. Here, when fixing $\overrightarrow{\text{rm}}_{-i}$, $\vec{w}_{-i}$, and $\theta_{\text{init}}$, if we can find a $\text{rm}_i^*$ such that when group $i$ reports $\text{rm}_i^*$ then the reported $w_i$ will not affect the training result, $\text{rm}_i^*$ actually serves to connect different $w_i \in \mathcal{W}$. For SW-Max training rules, we observe that the reward model rm that assigns the same value for all $\boldsymbol{x}$s, i.e., $\forall \boldsymbol{x}$, $\text{rm}(\boldsymbol{x}) = 1$ for maximum normalization, and $\text{rm}(\boldsymbol{x}) = 1/|T^*|$ for summation normalization, serves the role of $\text{rm}_i^*$. With the continuity of the training rule, this makes the domain of $\mathcal{R} \times \mathcal{W}$ connected in another sense that can also induce payment equivalence. Based on this, we derive the payment equivalence property:

**Corollary 4.10.** *Each continuous training rule $\psi \in \Psi^{SW}$ satisfies payment equivalence.*

As a continuous SW-Max training rule always satisfies payment equivalence, we can establish the relationship between $p^{AFF}$ and any other payment rule that implements it in DSIC. Combined with the inherent property of $p^{AFF}$, we derive the necessary conditions for a payment rule to satisfy more conditions, such as non-negativity and IR.

**Theorem 4.11.** *Given a continuous training rule $\psi \in \Psi^{SW}$ and a payment rule $p$ implements it in DSIC: If $p$ is always non-negative, it holds that for all $i$, $\overrightarrow{rm}$, $\vec{w}$, and $\theta_{init}$,*

$$p_i(\overrightarrow{rm}, \vec{w}, \theta_{init}) \geq p_i^{AFF}(\overrightarrow{rm}, \vec{w}, \theta_{init}).$$

*If $p$ implements $\psi$ in IR, then for any $\epsilon > 0$ and $i$, there exists $\overrightarrow{rm}_{-i}$, $\vec{w}_{-i}$, and $\theta_{init}$, such that for all $rm_i$ and $w_i$,*

$$p_i(\overrightarrow{rm}, \vec{w}, \theta_{init}) \leq p_i^{AFF}(\overrightarrow{rm}, \vec{w}, \theta_{init}) + \epsilon.$$

This result implies that if we want to design a payment $p$ to satisfy all these properties, $p^{AFF}$ is a "lower bound" for $p$, and $p$ should be sufficiently close to $p^{AFF}$ in some inputs.

## 4.3 Approximate Valuation

In this part, we study the influence of errors generated in practice on the incentive property in the RLHF Game. We abstract it as an approximate valuation problem [13]. Formally, when group $i$ reports its reward model $\text{rm}_i$, the mechanism may not use $\text{rm}_i$ but rather a noisy reward model $\widehat{\text{rm}}_i$

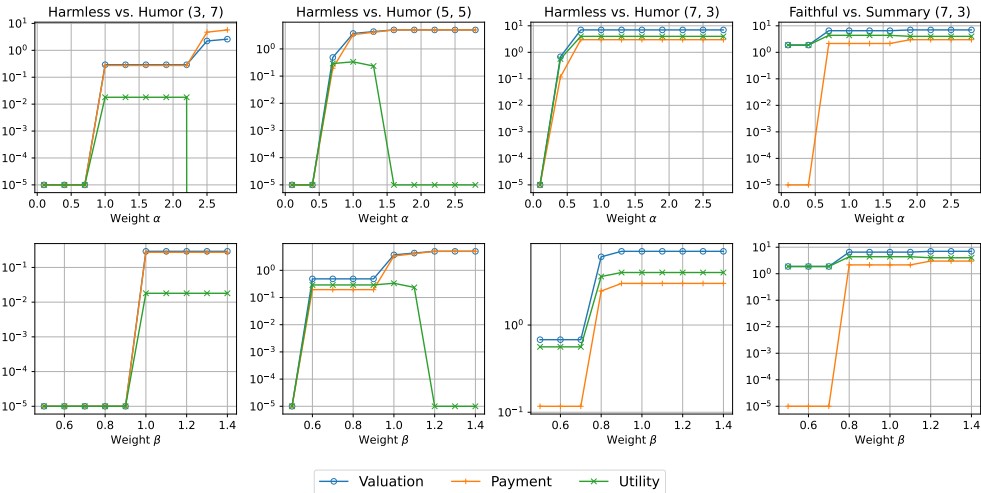

Figure 2: The simulation result for the mechanism $(\psi, p^{AFF})$ on real LLM setup. We set the group number $n = 2$, and the group size $(w_1, w_2)$ for each column is in the title. We report the valuation, the payment, and the utility for group 1 when adopting different reporting parameters $\alpha$ and $\beta$ (defined in Section 5). Truthfully reporting ($\alpha = 1$ and $\beta = 1$) brings the highest utility for all cases.

as the input. We assume that the noise is independently generated and there is an underlying joint distribution $F(\cdot|\overrightarrow{\mathrm{rm}})$ for the $\overrightarrow{\widehat{\mathrm{rm}}}$. This abstraction captures various errors that may occur in practical training. One example is that the calculation of valuation defined in Definition 3.1 requires sampling sequences from LLM, which may result in a deviation from the true valuation.

When the groups are rational, they could be aware of the noise and consider the influence of that on their utility. For group $i$ with reward model $\mathrm{rm}_i$ and group size $w_i$, it will computes an expected utility $U_i$ for reporting $(\mathrm{rm}'_i, w'_i)$ given by

$$U_i((\mathrm{rm}'_i, \overrightarrow{\mathrm{rm}}_{-i}), (w'_i, \vec{w}_{-i}); \psi, p, \mathrm{rm}_i, w_i) = \mathbb{E}_{\overrightarrow{\widehat{\mathrm{rm}}} \sim F(\cdot|(\mathrm{rm}'_i, \overrightarrow{\mathrm{rm}}_{-i}))} u_i(\overrightarrow{\widehat{\mathrm{rm}}}, (w'_i, \vec{w}_{-i}); \psi, p, \mathrm{rm}_i, w_i).$$

We consider the case that the noisy input reward models $\widehat{\mathrm{rm}}_i$ and the reported reward models $\mathrm{rm}_i$ are close. In that case, we show that when using a training rule $\psi \in \Psi^{SW}$, the distance between the true optimal point and the training outcome with noisy input is bounded. Based on that, we calculate the utility of a group under the mechanism $(\psi, p^{AFF})$ and derive the approximate incentive compatibility of the mechanism.

**Theorem 4.12.** *Assume that for any noisy input $\overrightarrow{\widehat{rm}}$ generated from $F(\cdot|\overrightarrow{rm})$, and $i \in [n]$, there is*

$$\max_{\boldsymbol{x} \in T^*} |\widehat{rm}_i(\boldsymbol{x}) - rm_i(\boldsymbol{x})| \le \epsilon.$$

*Then, with a training rule $\psi \in \Psi^{SW}$, $(\psi, p^{AFF})$ ensures that each group $i$ can benefit at most $2w_i\epsilon$ from misreporting the reward model.*

This theoretical result guarantees a considerable utility for truthful reporting. Since the maximum gain of misreporting for group $i$ is less than $2w_i\epsilon$ regardless of the others' reports, groups will tend to truthfully report in cases where finding the optimal strategy and modifying its reward model is costlier than $2w_i\epsilon$.

## 4.4 Efficient Implementation of the Mechanism

At the end of the whole section, we discuss how $p^{AFF}$ can be implemented in practice, as Proposition 4.4 and Theorem 4.11 show that it is "unique" to implement SW-Max training rules in DSIC. As is defined in Equation (1), we have to compute $\psi(\overrightarrow{\mathrm{rm}}_{-i}, \vec{w}_{-i}, \theta_{\mathrm{init}})$ for each $i$ aside from the final model $\theta^* := \psi(\overrightarrow{\mathrm{rm}}, \vec{w}, \theta_{\mathrm{init}})$. From the definition $\psi(\overrightarrow{\mathrm{rm}}_{-i}, \vec{w}_{-i}, \theta_{\mathrm{init}}) := \max_{\theta \in \Theta} \mathrm{OBJ}(\theta; \overrightarrow{\mathrm{rm}}_{-i}, \vec{w}_{-i}, \theta_{\mathrm{init}})$, finding a maximum over whole space $\Theta$ requires a whole training process. This results in $n$ additional

trainings when we have $n$ groups. To address this problem, we propose two heuristic methods that approximate the payment computation; the core of both is to take the maximum on a constraint space $\Theta'$ instead of the whole space $\Theta$.

- **Heuristic 1: Intermediate Models**
  During the training to obtain the model $\psi(\overrightarrow{\text{rm}}, \vec{w}, \theta_{\text{init}})$, we usually save intermediate models at different training steps. We can set $\Theta'$ to be these intermediate models. This requires no additional training and maintains payment non-negativity since $\theta^*$ is also in $\Theta'$, but DSIC is not strictly guaranteed as $\Theta'$ depends on group $i$'s report. However, the complex dependency makes strategic manipulation practically difficult.

- **Heuristic 2: Early-Stopped Training**
  We can perform early-stopped training to compute the $\psi(\overrightarrow{\text{rm}}_{-i}, \vec{w}_{-i}, \theta_{\text{init}})$. This means that we use a less powerful $\Theta'$ that is only dependent on $\overrightarrow{\text{rm}}_{-i}, \vec{w}_{-i}, \theta_{\text{init}}$. Since the independence of group $i$, this preserves DSIC theoretically. However, the payment may be negative as $\psi(\overrightarrow{\text{rm}}, \vec{w}, \theta_{\text{init}})$ may outperform the early-stopped $\psi(\overrightarrow{\text{rm}}_{-i}, \vec{w}_{-i}, \theta_{\text{init}})$ in terms of $\text{ASW}_{-i}$.

These heuristics provide a practical trade-off: Heuristic 1 offers maximum computational efficiency with relaxed theoretical guarantees, while Heuristic 2 preserves DSIC with moderate additional cost. From a theory perspective, we can derive the following result based on Theorem 4.11.

**Corollary 4.13.** *Given a continuous training rule $\psi \in \Psi^{SW}$, if the payment rule $p$ implements it in DSIC, IR and is always non-negative, then for any $\epsilon > 0$, there exists $i$, $\overrightarrow{rm}_{-i}$, $\vec{w}_{-i}$ and $\theta_{init}$, such that for all $rm_i$ and $w_i$, denote $rm = (rm_i, \overrightarrow{rm}_{-i})$ and $w = (w_i, \vec{w}_{-i})$, we have*

$$p_i^{AFF}(rm, w, \theta_{init}) \le p_i(rm, w, \theta_{init}) \le p_i^{AFF}(rm, w, \theta_{init}) + \epsilon.$$

This indicates that any payment rule $p$ that satisfies all these properties must closely approximate $p^{AFF}$ in certain inputs. This somewhat showcases a *tradeoff between theoretical guarantees and computational efficiency*. A more rigorous analysis of the efficiency loss caused by these heuristics or an "impossibility theorem" regarding efficient implementation is left for future work.

## 5 Empirical Study

In this section, we present an empirical evaluation of the proposed mechanism. Our objectives are twofold: first, to demonstrate that in practical LLM settings, agents can benefit from misreporting their preferences and distorting the learning outcomes; and second, to intuitively show how our mechanism incentivizes truthful reporting[2].

**Models and Datasets.** Our experimental setup follows the literature on Multiple-Objective RLHF [62, 70, 78]. We consider two tasks: the Helpful Assistants task [5] and the Reddit Summary task [72], using Llama-2 7b [74] as the base model for both. For the Helpful Assistants task, the initial model $\text{LLM}_{\theta_{\text{init}}}$ is obtained by supervised fine-tuning a Llama-2 7b model on the Anthropic-HH dataset [5]. We then apply two reward models during the RLHF process to measure harmlessness and humor, respectively. For the Reddit Summary task, the model is fine-tuned on the Summarize-from-Feedback dataset [72], with two reward models assessing the summary's quality and faithfulness.

We formulate these tasks as two mechanism design scenarios: the "Harmless vs. Humor" game for the Helpful Assistants task, and the "Faithful vs. Summary" game for the Reddit Summary task. In each game, we assume that there are two groups whose joint preferences are captured by a reward model. For example, in "Harmless vs. Humor, " group 1 prioritizes harmlessness, while group 2 values humor. The corresponding reward models for these preferences are denoted as $\text{rm}_1$ (harmlessness) and $\text{rm}_2$ (humor), with synthetic group size vectors $(w_1, w_2)$ selected from $\{(3, 7), (5, 5), (7, 3)\}$, varying across different settings.

**Implementation Details.** We implement the basic training rule from Definition 4.1, using the KL-divergence as the distance measure $f$. To balance model optimality with training cost, we simplify the problem by replacing the entire parameter space $\Theta$ with a representative finite set $\Theta'$.

---

[2]The code for the simulation is available at GitHub.

Models are first trained using single reward models and then combined via the Rewarded Soups technique [62] to produce a set of hybrid models, $\{\theta_1, \theta_2, \ldots, \theta_K\}$, which constitute $\Theta'$. Optimality is then defined over this finite set. As shown in Rame et al. [62], this approach reduces training costs while maintaining performance comparable to full multi-objective fine-tuning.

Given the large space of potential misreporting strategies, we focus on two simple ones:

- **Strategy (1)**: $\widetilde{\mathrm{rm}}_i = \mathrm{rm}_i$ and $\tilde{w}_i = \alpha w_i$
  *Naïve overstatement*: Exaggerating group size to gain more influence, requiring no knowledge of other groups.
- **Strategy (2)**: $\widetilde{\mathrm{rm}}_i = \beta \mathrm{rm}_i + (1 - \beta)\mathrm{rm}_{-i}$ and $\tilde{w}_i = w_i$
  *Strategic manipulation*: Leveraging other groups' preferences to downplay opposing outcomes, requiring some information about conflicts.

Intuitively, $\alpha = 1$ and $\beta = 1$ represent truthful reporting. Increasing $\alpha$ or $\beta$ allows a group to gain more influence in the training process. Our experiments confirm that both strategies can be profitable. However, the DSIC of our mechanism ensures that truthful reporting yields higher utility than any misreporting strategy.

**Result Analysis.** Since the outputs of different reward models have varying scales, we normalize all reward values to $[0, 1]$, where the maximum and minimum values are 1 and 0, respectively. We then report the normalized valuations, payments, and utilities of group $i$ for different reporting strategies in Figure 2. Each column represents a specific RLHF Game with a given group size $(w_1, w_2)$.

As shown in the figure, increasing $\alpha$ or $\beta$ leads to a higher valuation for the group, confirming that groups can benefit from simple misreporting in the absence of payments. However, when the payment $p^{AFF}$ is applied, it increases with $\alpha$ or $\beta$, offsetting the gains in valuation. This ensures that truthful reporting ($\alpha = 1$, $\beta = 1$) maximizes utility in all cases. Additional simulation settings are provided in Appendix F.

## 6 Conclusion and Future Work

This paper studies incentive issues in a potential economic scenario where a platform offers LLM fine-tuning services to aggregate preferences and agents strategically report to get a preferred outcome. We focus on aggregation objectives that maximize social welfare subject to regularization constraints, referred to as SW-Max rules. Through a comprehensive analysis of strategic reporting, we demonstrate the critical role of payment schemes in incentivizing truthful reporting under SW-Max rules. We derive sufficient conditions for payment equivalence and identify necessary conditions for implementing SW-Max rules within additional constraints. Moreover, we analyze how perturbed input will influence the mechanism to account for practical errors that inevitably arise and show that the mechanism satisfies approximate DSIC. Finally, we conduct experiments within real-world LLM setups, showcasing how the proposed mechanism effectively incentivizes truthful reporting.

Building on our proposed scenario and formulated model, we identify several promising directions for future research from both theoretical and empirical perspectives. First, exploring and modeling more general training rules could enhance our understanding of the framework. As noted in Appendix E, cycle monotonicity is a necessary and sufficient condition for implementability, but its validation is complex. Identifying a simpler condition to ensure implementability and investigating properties like payment equivalence for these rules are critical next steps. Second, studying preference aggregation across multiple models, particularly with diversity considerations, is a valuable direction. Third, as discussed in Section 4.4, developing mechanisms or criteria that balance computational efficiency and incentive compatibility in the RLHF Game could improve its real-world applicability. Finally, applying mechanism design theory to other large language model contexts, such as API pricing, retrieval-augmented generation (RAG), and prompt engineering, offers significant opportunities for further exploration.

## Acknowledgments

This work is supported by the National Natural Science Foundation of China (Grant No. 62572010 and No. 62506010). We thank all anonymous reviewers for their helpful feedback.

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

## Limitation

The main limitation of this paper is that we mainly consider the SW-Max training rules and their theoretical properties. Further study could consider more training rules and extend our model to the DPO scenario, in which each group only provides pairs of data rather than a reward model.

## A    Further Related Work

In this section, we review relevant research across various domains that are related to our paper, including works on RLHF with multiple reward models, multi-parameter auctions, and the intersection of game theory and LLMs.

### A.1    RLHF with Multiple Reward Models

Research involving multiple reward models primarily focuses on developing algorithms to enhance practical performance. Some studies design methods simultaneously satisfying multiple preferences [11, 41, 62, 63, 70, 78, 81]. They develop more efficient algorithms to extend the Pareto frontier among different objectives [41, 62, 70, 81] and balance issues from various perspectives [11, 59, 63].

Additionally, there is a body of work that trains multiple models for a single preference and then ensembles them to improve the robustness of RLHF [18, 82], mitigate the influence of incorrect and ambiguous preferences in the dataset [76], and reduce reward hacking [26]. Unlike these approaches, our work considers how to collect misaligned preferences truthfully from different agents. As we have mentioned, these works are often assumed to be accessible to humans' actual preferences, neglecting the incentive issue for motivating rational agents to report truthfully.

### A.2    Multi-parameter Auctions

Several studies have explored the properties relevant to our paper in various multi-parameter auction scenarios, such as implementability [4, 7, 16, 49, 65, 67] and payment equivalence [6, 38, 39, 60]. Another central topic in auction theory is to design mechanisms that satisfy DSIC and IR while maximizing the expected revenue for the auctioneer. Although the single-parameter scenario has been resolved by Myerson [53], the optimal auction design for multi-parameter settings remains an open question. Therefore, there is a stream of research focusing on a specific subset, affine maximizer auctions, which inherently satisfy DSIC and IR [9, 42, 44, 64, 68, 73], and proposing optimizations to enhance empirical performance [20, 22, 23]. Compared to these works, we are the first to discuss the property of payment equivalence and the revenue-maximizing solution for SW-Max training rules in the scenario of fine-tuning LLMs.

### A.3    Game Theory and LLMs

In addition to the work we review in the primary related work, others have explored the intersection of game theory and large language models from different perspectives. A line of work studies other LLM-related scenarios from the algorithmic game theory perspective. Laufer et al. [43] abstracted the fine-tuning process as a bargaining game and characterized the perfect sub-game equilibria. Dubey et al. [24] proposed an auction where bidders compete to place their content within a summary generated by an LLM. Conitzer et al. [17] considered incorporating social choice theory in LLM alignment. Feizi et al. [29] explored the potential for leveraging LLMs in online advertising systems.

More broadly, some research has proposed algorithms for training LLMs inspired by concepts in game theory, such as Nash learning from human feedback [51], consensus game [40], direct Nash optimization [66], and Gemp et al. [33]. And various studies assess LLMs from a game-theoretical perspective, examining aspects such as rationality [12, 28], behavior in matrix games [2, 32, 45], and performance in strategic games like auctions [35, 36], Werewolf [79, 80], Avalon [77], Diplomacy [27, 50], card game [30] and electronic game [1, 47, 69]. There are also comprehensive surveys [31, 37, 83].

# B  Omitted Proofs in Section 4.1

**Theorem 4.2.** *In the RLHF Game with mechanism $(\psi, p)$ that $\psi \in \Psi^{SW}$ and $p \equiv 0$, for group $i$, define $s_i := |\{r | r = rm_i(x), x \in T^*\}|$ and $\underline{rm_i} := \min_{x \in T^*} rm_i(x)$:*

1. *If $s_i = 1$, truthfully reporting is the optimal strategy regardless of other groups' reports.*

2. *If $s_i \geq 2$ and $\underline{rm_i} > 0$, there is a strategy that yields strictly higher utility than truthfully reporting regardless of other groups' reports.*

*Proof.* If $s_i = 1$, the group gets the same utility from all training outcomes. Therefore, any strategy is optimal. We then analyze the case $s_i \geq 2$ and $\underline{rm_i} > 0$ in the following. First, the optimization of $\psi$ can be written as an equivalent constraint programming problem on the $\text{LLM}_\theta$:

$$\arg\max_{\text{LLM}_\theta} \quad \sum_{i=1}^n w_i v_i(\theta; rm_i) - \sum_{x \in T^*} \text{LLM}_{\theta_{\text{init}}}(x) f\left(\frac{\text{LLM}_\theta(x)}{\text{LLM}_{\theta_{\text{init}}}(x)}\right)$$

$$\text{s.t.} \quad \sum_{x \in T^*} \text{LLM}_\theta(x) = 1$$

$$\text{LLM}_\theta(x) \geq 0 \quad \forall x \in T^*$$

Because of the assumption that the optimal policy satisfies $\text{LLM}_\theta(x) > 0$ for all $x \in T^*$, we can infer that the condition $\text{LLM}_\theta(x) \geq 0, \forall x \in T^*$ is not active for the optimal solution. Since the convexity of the function $f$, by KKT condition, the necessary condition for the optimal $\theta^*$ is that there exists $\mu \in \mathbb{R}$ [46], such that

$$\sum_{i=1}^n w_i \frac{\partial v_i}{\partial \text{LLM}_\theta(x)} - f'\left(\frac{\text{LLM}_\theta(x)}{\text{LLM}_{\theta_{\text{init}}}(x)}\right) = \mu \quad \forall x \in T^*, \quad \sum_{x \in T^*} \text{LLM}_\theta(x) = 1.$$

Under Definition 3.1, $\frac{\partial v_i}{\partial \text{LLM}_\theta(x)} = rm_i(x)$, so we have

$$\sum_{i=1}^n w_i rm_i(x) - f'\left(\frac{\text{LLM}_\theta(x)}{\text{LLM}_{\theta_{\text{init}}}(x)}\right) = \mu \quad \forall x \in T^*. \tag{OPT}$$

We mainly discuss the strategies other than simply over-reporting the group size $\vec{w}$. We omit the notation $\vec{w}$ for simplicity.

Next, our main technique is to construct a report reward model $rm_i' \neq rm_i$ for group $i$ such that $v_i(\psi((rm_i', \overrightarrow{rm}_{-i}), \theta_{\text{init}}); rm_i) > v_i(\psi((rm_i, \overrightarrow{rm}), \theta_{\text{init}}); rm_i)$ holds for all $\overrightarrow{rm}_{-i}$ and $\theta_{\text{init}}$.

**The Summation Normalization Case.** We first discuss the case of the reward model being normalized by summation. We take the $x_1 \in \arg\max_{x \in T^*} rm_i(x), x_2 \in \arg\min_{x \in T^*} rm_i(x)$. Since $\min_{x \in T^*} rm_i(x) > 0$, we have $rm_i(x_1) < 1$ and $rm_i(x_2) > 0$. Then we take a small $\epsilon < \min\{1 - rm_i(x_1), rm_i(x_2)\}$ and define $rm_i'$ as:

$$rm_i'(x) = \begin{cases} rm_i(x) + \epsilon, & x = x_1, \\ rm_i(x) - \epsilon, & x = x_2 \\ rm_i(x), & x \neq x_1, x \neq x_2. \end{cases}$$

Intuitively, by reporting $rm_i'$, group $i$ pretends to value more for the most preferred $x$ and less for the least preferred $x$. Let $\theta = \psi((rm_i, \overrightarrow{rm}_{-i}), \theta_{\text{init}})$ and $\theta' = \psi((rm_i', \overrightarrow{rm}_{-i}), \theta_{\text{init}})$, we use $\mu$ and $\mu'$ to denote the variable in the necessary condition for $\text{LLM}_\theta$ and $\text{LLM}_{\theta'}$, and we can derive the following results.

(a) $\text{LLM}_{\theta'}(x_1) > \text{LLM}_\theta(x_1)$ and $\text{LLM}_{\theta'}(x_2) < \text{LLM}_\theta(x_2)$. We prove the former by contradiction: if $\text{LLM}_{\theta'}(x_1) \leq \text{LLM}_\theta(x_1)$, then by the convexity of $f$, we have

$$f'\left(\frac{\text{LLM}_{\theta'}(x_1)}{\text{LLM}_{\theta_{\text{init}}}(x)}\right) \leq f'\left(\frac{\text{LLM}_\theta(x_1)}{\text{LLM}_{\theta_{\text{init}}}(x)}\right).$$

With $rm_i'(x_1) > rm_i(x_1)$, we can infer that $\mu' > \mu$. However, since for all $x \neq x_1$, we have $rm_i'(x) \leq rm_i(x)$, to satisfy the optimal condition in (OPT), there must be for all $x \neq x_1$,

$$f'\left(\frac{\text{LLM}_{\theta'}(x)}{\text{LLM}_{\theta_{\text{init}}}(x)}\right) < f'\left(\frac{\text{LLM}_\theta(x)}{\text{LLM}_{\theta_{\text{init}}}(x)}\right).$$

Which is equivalent to $\text{LLM}_{\theta'}(\boldsymbol{x}) < \text{LLM}_{\theta}(\boldsymbol{x})$, and hence results in $\sum_{\boldsymbol{x}\in T^*}\text{LLM}_{\theta'}(\boldsymbol{x}) < \sum_{\boldsymbol{x}\in T^*}\text{LLM}_{\theta}(\boldsymbol{x}) = 1$. The latter, $\text{LLM}_{\theta'}(\boldsymbol{x}_2) < \text{LLM}_{\theta}(\boldsymbol{x}_2)$, can be proved by totally same method.

(b) The order of $\text{LLM}_{\theta}(\boldsymbol{x})$ and $\text{LLM}_{\theta'}(\boldsymbol{x})$ for all $\boldsymbol{x} \notin \{\boldsymbol{x}_1, \boldsymbol{x}_2\}$ is consistent. Without loss of generality, we assume there is $\boldsymbol{x}_3 \notin \{\boldsymbol{x}_1, \boldsymbol{x}_2\}$ such that $\text{LLM}_{\theta'}(\boldsymbol{x}_3) \geq \text{LLM}_{\theta}(\boldsymbol{x}_3)$. Then we have

$$f'\left(\frac{\text{LLM}_{\theta'}(\boldsymbol{x}_3)}{\text{LLM}_{\theta_{\text{init}}}(\boldsymbol{x})}\right) \geq f'\left(\frac{\text{LLM}_{\theta}(\boldsymbol{x}_3)}{\text{LLM}_{\theta_{\text{init}}}(\boldsymbol{x})}\right).$$

Then, we can infer that $\mu' \leq \mu$. For all $\boldsymbol{x} \notin \{\boldsymbol{x}_1, \boldsymbol{x}_2\}$, to satisfy Equation (OPT), there must be

$$f'\left(\frac{\text{LLM}_{\theta'}(\boldsymbol{x})}{\text{LLM}_{\theta_{\text{init}}}(\boldsymbol{x})}\right) \geq f'\left(\frac{\text{LLM}_{\theta}(\boldsymbol{x})}{\text{LLM}_{\theta_{\text{init}}}(\boldsymbol{x})}\right).$$

which is equivalent to $\text{LLM}_{\theta'}(\boldsymbol{x}) \geq \text{LLM}_{\theta}(\boldsymbol{x})$. Similarly, if there is $\boldsymbol{x}_3 \notin \{\boldsymbol{x}_1, \boldsymbol{x}_2\}$ such that $\text{LLM}_{\theta'}(\boldsymbol{x}_3) \leq \text{LLM}_{\theta}(\boldsymbol{x}_3)$, then for all $\boldsymbol{x} \notin \{\boldsymbol{x}_1, \boldsymbol{x}_2\}$, there is $\text{LLM}_{\theta'}(\boldsymbol{x}) \leq \text{LLM}_{\theta}(\boldsymbol{x})$.

Finally, with the results in (a) and (b), when $\text{LLM}_{\theta'}(\boldsymbol{x}) \leq \text{LLM}_{\theta}(\boldsymbol{x})$ for all $\boldsymbol{x} \notin \{\boldsymbol{x}_1, \boldsymbol{x}_2\}$, the change in the utility of group $i$ can be calculated by

$$\sum_{\boldsymbol{x}\in T^*}(\text{LLM}_{\theta'}(\boldsymbol{x}) - \text{LLM}_{\theta}(\boldsymbol{x}))\,\text{rm}_i(\boldsymbol{x})$$

$$= \sum_{\boldsymbol{x}\neq\boldsymbol{x}_1,\boldsymbol{x}\in T^*}(\text{LLM}_{\theta'}(\boldsymbol{x}) - \text{LLM}_{\theta}(\boldsymbol{x}))\,\text{rm}_i(\boldsymbol{x}) + (\text{LLM}_{\theta'}(\boldsymbol{x}_1) - \text{LLM}_{\theta}(\boldsymbol{x}_1))\,\text{rm}_i(\boldsymbol{x}_1)$$

$$= -\sum_{\boldsymbol{x}\neq\boldsymbol{x}_1,\boldsymbol{x}\in T^*}(\text{LLM}_{\theta}(\boldsymbol{x}) - \text{LLM}_{\theta'}(\boldsymbol{x}))\,\text{rm}_i(\boldsymbol{x}) + (\text{LLM}_{\theta'}(\boldsymbol{x}_1) - \text{LLM}_{\theta}(\boldsymbol{x}_1))\,\text{rm}_i(\boldsymbol{x}_1)$$

$$\overset{(2)}{\geq} -\sum_{\boldsymbol{x}\neq\boldsymbol{x}_1,\boldsymbol{x}\in T^*}(\text{LLM}_{\theta}(\boldsymbol{x}) - \text{LLM}_{\theta'}(\boldsymbol{x}))\,\text{rm}_i(\boldsymbol{x}_1) + (\text{LLM}_{\theta'}(\boldsymbol{x}_1) - \text{LLM}_{\theta}(\boldsymbol{x}_1))\,\text{rm}_i(\boldsymbol{x}_1)$$

$$= \text{rm}_i(\boldsymbol{x}_1)\left(\text{LLM}_{\theta'}(\boldsymbol{x}_1) - \text{LLM}_{\theta}(\boldsymbol{x}_1) - \sum_{\boldsymbol{x}\neq\boldsymbol{x}_1,\boldsymbol{x}\in T^*}(\text{LLM}_{\theta}(\boldsymbol{x}) - \text{LLM}_{\theta'}(\boldsymbol{x}))\right)$$

$$= \text{rm}_i(\boldsymbol{x}_1)\sum_{\boldsymbol{x}\in T^*}(\text{LLM}_{\theta'}(\boldsymbol{x}) - \text{LLM}_{\theta}(\boldsymbol{x})) = 0.$$

When $\text{LLM}_{\theta'}(\boldsymbol{x}) \geq \text{LLM}_{\theta}(\boldsymbol{x})$ for all $\boldsymbol{x} \neq \boldsymbol{x}_1, \boldsymbol{x}_2$, the change in the utility of group $i$ can be calculated by

$$\sum_{\boldsymbol{x}\in T^*}(\text{LLM}_{\theta'}(\boldsymbol{x}) - \text{LLM}_{\theta}(\boldsymbol{x}))\,\text{rm}_i(\boldsymbol{x})$$

$$= \sum_{\boldsymbol{x}\neq\boldsymbol{x}_2,\boldsymbol{x}\in T^*}(\text{LLM}_{\theta'}(\boldsymbol{x}) - \text{LLM}_{\theta}(\boldsymbol{x}))\,\text{rm}_i(\boldsymbol{x}) + (\text{LLM}_{\theta'}(\boldsymbol{x}_2) - \text{LLM}_{\theta}(\boldsymbol{x}_2))\,\text{rm}_i(\boldsymbol{x}_2)$$

$$= \sum_{\boldsymbol{x}\neq\boldsymbol{x}_2,\boldsymbol{x}\in T^*}(\text{LLM}_{\theta'}(\boldsymbol{x}) - \text{LLM}_{\theta}(\boldsymbol{x}))\,\text{rm}_i(\boldsymbol{x}) - (\text{LLM}_{\theta}(\boldsymbol{x}_2) - \text{LLM}_{\theta'}(\boldsymbol{x}_2))\,\text{rm}_i(\boldsymbol{x}_2)$$

$$\overset{(3)}{\geq} \sum_{\boldsymbol{x}\neq\boldsymbol{x}_2,\boldsymbol{x}\in T^*}(\text{LLM}_{\theta'}(\boldsymbol{x}) - \text{LLM}_{\theta}(\boldsymbol{x}))\,\text{rm}_i(\boldsymbol{x}_2) - (\text{LLM}_{\theta}(\boldsymbol{x}_2) - \text{LLM}_{\theta'}(\boldsymbol{x}_2))\,\text{rm}_i(\boldsymbol{x}_2)$$

$$= \text{rm}_i(\boldsymbol{x}_2)\left(\sum_{\boldsymbol{x}\neq\boldsymbol{x}_2,\boldsymbol{x}\in T^*}(\text{LLM}_{\theta'}(\boldsymbol{x}) - \text{LLM}_{\theta}(\boldsymbol{x})) - (\text{LLM}_{\theta}(\boldsymbol{x}_2) - \text{LLM}_{\theta'}(\boldsymbol{x}_2))\right)$$

$$= \text{rm}_i(\boldsymbol{x}_2)\sum_{\boldsymbol{x}\in T^*}(\text{LLM}_{\theta'}(\boldsymbol{x}) - \text{LLM}_{\theta}(\boldsymbol{x})) = 0.$$

Note that both (2) and (3) are because of $\text{rm}_i(\boldsymbol{x}_1) \geq \text{rm}_i(\boldsymbol{x}_2)$. And unless $\text{rm}_i(\boldsymbol{x}_1) = \text{rm}_i(\boldsymbol{x}_2)$, which is excluded by $s_i \geq 2$, the ">"s are hold.

**The Maximum Normalization Case.** The case of the reward model being normalized by the maximum is similar. We take the $\boldsymbol{x}_1 \in \arg\min_{\boldsymbol{x} \in T^*} \mathrm{rm}_i(\boldsymbol{x})$. Since $\min_{\boldsymbol{x} \in T^*} \mathrm{rm}_i(\boldsymbol{x}) > 0$, we have $\mathrm{rm}_i(\boldsymbol{x}_1) > 0$. Then we take a small $\epsilon < \mathrm{rm}_i(\boldsymbol{x}_1)$ and define $\mathrm{rm}_i'$ as:

$$\mathrm{rm}_i'(\boldsymbol{x}) = \begin{cases} \mathrm{rm}_i(\boldsymbol{x}) - \epsilon, & \boldsymbol{x} = \boldsymbol{x}_1, \\ \mathrm{rm}_i(\boldsymbol{x}), & \boldsymbol{x} \neq \boldsymbol{x}_1. \end{cases}$$

With the same technique, we first show that $\mathrm{LLM}_{\theta'}(\boldsymbol{x}_1) < \mathrm{LLM}_\theta(\boldsymbol{x}_1)$ and $\mathrm{LLM}_{\theta'}(\boldsymbol{x}) > \mathrm{LLM}_\theta(\boldsymbol{x})$ for all $\boldsymbol{x} \neq \boldsymbol{x}_1$. After that, it is easy to derive that when $s_i \geq 2$, the change in the utility of group $i$ satisfies

$$\sum_{\boldsymbol{x} \in T^*} \left( \mathrm{LLM}_{\theta'}(\boldsymbol{x}) - \mathrm{LLM}_\theta(\boldsymbol{x}) \right) \mathrm{rm}_i(\boldsymbol{x}) > 0.$$

$\square$

**Lemma B.1.** *When the training rule $\psi \in \Psi^{SW}$, and the divergence function $f$ is $\alpha$-strongly convex and $C^2$-smooth, then $\psi$ satisfies Definition 4.7.*

*Proof.* As is shown in the proof of Theorem 4.2, we have two Lagrangian variables $\mu$ and $\mu'$ for $(\overrightarrow{\mathrm{rm}}, \vec{w})$ and $(\overrightarrow{\mathrm{rm}}, \vec{w})$, respectively. We have the following equations:

$$\sum_{i=1}^n w_i \mathrm{rm}_i(\boldsymbol{x}) - f'\left( \frac{\mathrm{LLM}_\theta(\boldsymbol{x})}{\mathrm{LLM}_{\theta_{\mathrm{init}}}(\boldsymbol{x})} \right) = \mu, \quad \forall \boldsymbol{x} \in T^*.$$

$$\sum_{i=1}^n w_i' \mathrm{rm}_i'(\boldsymbol{x}) - f'\left( \frac{\mathrm{LLM}_{\theta'}(\boldsymbol{x})}{\mathrm{LLM}_{\theta_{\mathrm{init}}}(\boldsymbol{x})} \right) = \mu', \quad \forall \boldsymbol{x} \in T^*.$$

Firstly, we have $|\mu' - \mu| \leq \max_{\boldsymbol{x} \in T^*} |\sum_{i=1}^n w_i \mathrm{rm}_i(\boldsymbol{x}) - \sum_{i=1}^n w_i' \mathrm{rm}_i'(\boldsymbol{x})|$. Otherwise, without loss of generality, assume that $\mu' - \mu > \max_{\boldsymbol{x} \in T^*} |\sum_{i=1}^n w_i \mathrm{rm}_i(\boldsymbol{x}) - \sum_{i=1}^n w_i' \mathrm{rm}_i'(\boldsymbol{x})|$, then we can derive that $\forall \boldsymbol{x} \in T^*$,

$$f'\left( \frac{\mathrm{LLM}_\theta(\boldsymbol{x})}{\mathrm{LLM}_{\theta_{\mathrm{init}}}(\boldsymbol{x})} \right) < f'\left( \frac{\mathrm{LLM}_{\theta'}(\boldsymbol{x})}{\mathrm{LLM}_{\theta_{\mathrm{init}}}(\boldsymbol{x})} \right).$$

This means that $\mathrm{LLM}_\theta(\boldsymbol{x}) < \mathrm{LLM}_{\theta'}(\boldsymbol{x})$ for all $\boldsymbol{x}$, which leads the contradiction. Therefore, we have for all $\boldsymbol{x} \in T^*$

$$\left| f'\left( \frac{\mathrm{LLM}_\theta(\boldsymbol{x})}{\mathrm{LLM}_{\theta_{\mathrm{init}}}(\boldsymbol{x})} \right) - f'\left( \frac{\mathrm{LLM}_{\theta'}(\boldsymbol{x})}{\mathrm{LLM}_{\theta_{\mathrm{init}}}(\boldsymbol{x})} \right) \right| = \left| \sum_{i=1}^n w_i \mathrm{rm}_i(\boldsymbol{x}) - \sum_{i=1}^n w_i' \mathrm{rm}_i'(\boldsymbol{x}) + \mu' - \mu \right|$$

$$\leq 2 \left| \sum_{i=1}^n w_i \mathrm{rm}_i(\boldsymbol{x}) - \sum_{i=1}^n w_i' \mathrm{rm}_i'(\boldsymbol{x}) \right|.$$

By $C^2$-smoothness of $f$ and the $\alpha$-strongly convexity, we have for all $\boldsymbol{x} \in T^*$

$$|\mathrm{LLM}_\theta(\boldsymbol{x}) - \mathrm{LLM}_{\theta'}(\boldsymbol{x})| \leq \frac{\mathrm{LLM}_{\theta_{\mathrm{init}}}(\boldsymbol{x})}{\alpha} \left| f'\left( \frac{\mathrm{LLM}_\theta(\boldsymbol{x})}{\mathrm{LLM}_{\theta_{\mathrm{init}}}(\boldsymbol{x})} \right) - f'\left( \frac{\mathrm{LLM}_{\theta'}(\boldsymbol{x})}{\mathrm{LLM}_{\theta_{\mathrm{init}}}(\boldsymbol{x})} \right) \right|$$

$$\leq \frac{2\mathrm{LLM}_{\theta_{\mathrm{init}}}(\boldsymbol{x})}{\alpha} \left| \sum_{i=1}^n w_i \mathrm{rm}_i(\boldsymbol{x}) - \sum_{i=1}^n w_i' \mathrm{rm}_i'(\boldsymbol{x}) \right|.$$

Therefore, for any $\epsilon > 0$, if $|\sum_{i=1}^n w_i \mathrm{rm}_i(\boldsymbol{x}) - \sum_{i=1}^n w_i' \mathrm{rm}_i'(\boldsymbol{x})| < \frac{\alpha\epsilon}{2}$, then $|\mathrm{LLM}_\theta(\boldsymbol{x}) - \mathrm{LLM}_{\theta'}(\boldsymbol{x})| \leq \epsilon$. $\square$

**Theorem B.2** (Detailed version of Theorem 4.3). *In the RLHF Game with mechanism $(\psi, p)$ that $\psi \in \Psi^{SW}$ and $p \equiv 0$, when $f$ is $\alpha$-strongly convex and $C^2$-smooth, suppose group $i$ has preference $\mathrm{rm}_i$ and group size $w_i$, other groups report $(\overrightarrow{\mathrm{rm}}_{-i}, \vec{w}_{-i})$ and the initial model $\theta_{init}$, we define*

$$t(\boldsymbol{z}) := \sum_{\boldsymbol{x} \in T^*} \frac{(\mathrm{rm}_i(\boldsymbol{z}) - \mathrm{rm}_i(\boldsymbol{x}))\mathrm{LLM}_{\theta_{init}}(\boldsymbol{x})}{f''\left( \frac{\mathrm{LLM}_\theta(\boldsymbol{x})}{\mathrm{LLM}_{\theta_{init}}(\boldsymbol{x})} \right)},$$

*in which $\theta = \psi(\overrightarrow{\mathrm{rm}}, \vec{w}, \theta_{init})$. When $s_i \geq 2$ and $\underline{\mathrm{rm}}_i = 0$:*

1. *For the maximum normalization case, if there exist $\boldsymbol{x}_1 \in T^*$, $t(\boldsymbol{x}_1) \neq 0$ and $0 < \mathrm{rm}_i(\boldsymbol{x}_1) < 1$, truthful reporting is not the optimal strategy.*

2. *For the summation normalization case, if there exist $\boldsymbol{x}_1 \in T^*$, $t(\boldsymbol{x}_1) < 0$ and $0 < \mathrm{rm}_i(\boldsymbol{x}_1) < 1$, truthful reporting is not the optimal strategy.*

3. *For the summation normalization case, if there exist $\boldsymbol{x}_1 \in T^*$, $t(\boldsymbol{x}_1) > 0$ and we can also find $\boldsymbol{x}_2 \in T^*$, such that $1 > \mathrm{rm}_i(\boldsymbol{x}_1) \geq \mathrm{rm}_i(\boldsymbol{x}_2) > 0$ and $\frac{1}{\mathrm{LLM}_{\theta_{init}}(\boldsymbol{x}_1)} f'' \left( \frac{\mathrm{LLM}_\theta(\boldsymbol{x}_1)}{\mathrm{LLM}_{\theta_{init}}(\boldsymbol{x}_1)} \right) < \frac{1}{\mathrm{LLM}_{\theta_{init}}(\boldsymbol{x}_2)} f'' \left( \frac{\mathrm{LLM}_\theta(\boldsymbol{x}_2)}{\mathrm{LLM}_{\theta_{init}}(\boldsymbol{x}_2)} \right)$, truthful reporting is not the optimal strategy.*

*Proof.* As is shown in the proof of Theorem 4.2, the necessary condition for the solution $\theta$ is that there exists a $\mu \in \mathbb{R}$ such that

$$\sum_{i=1}^{n} w_i \mathrm{rm}_i(\boldsymbol{x}) - f' \left( \frac{\mathrm{LLM}_\theta(\boldsymbol{x})}{\mathrm{LLM}_{\theta_{init}}(\boldsymbol{x})} \right) = \mu \quad \forall \boldsymbol{x} \in T^*, \quad \sum_{\boldsymbol{x} \in T^*} \mathrm{LLM}_\theta(\boldsymbol{x}) = 1.$$

And by Lemma B.1, we can also use the condition Definition 4.7.

**The Maximum Normalization Case (1).** Without loss of generality, we assume that there exists $\boldsymbol{x}_1$ such that $t(\boldsymbol{x}_1) > 0$, we take $0 < \epsilon < 1 - \mathrm{rm}_i(\boldsymbol{x}_1)$ to construct a report $\mathrm{rm}_i'$

$$\mathrm{rm}_i'(\boldsymbol{x}) = \begin{cases} \mathrm{rm}_i(\boldsymbol{x}) + \epsilon, & \boldsymbol{x} = \boldsymbol{x}_1, \\ \mathrm{rm}_i(\boldsymbol{x}), & \boldsymbol{x} \neq \boldsymbol{x}_1. \end{cases}$$

Suppose that $\mu'$ is the Lagrangian variable for the optimal solution $\theta'$ when reporting $\mathrm{rm}_i'$, we can derive that

$$\mu' - \mu = w_i \epsilon \mathbb{I}_{\boldsymbol{x} = \boldsymbol{x}_1} - \left( f' \left( \frac{\mathrm{LLM}_{\theta'}(\boldsymbol{x})}{\mathrm{LLM}_{\theta_{init}}(\boldsymbol{x})} \right) - f' \left( \frac{\mathrm{LLM}_\theta(\boldsymbol{x})}{\mathrm{LLM}_{\theta_{init}}(\boldsymbol{x})} \right) \right) \quad \forall \boldsymbol{x} \in T^*.$$

With a similar analyze in the proof of Theorem 4.2, we can induce that $\mu' > \mu$ and $\mathrm{LLM}_{\theta'}(\boldsymbol{x}) < \mathrm{LLM}_\theta(\boldsymbol{x})$ for all $\boldsymbol{x} \neq \boldsymbol{x}_1$. By the $C^2$-smoothness of $f$, for each $\boldsymbol{x} \neq \boldsymbol{x}_1$, there exits a $\mathrm{LLM}_{\theta'}(\boldsymbol{x}) \leq \boldsymbol{z} \leq \mathrm{LLM}_\theta(\boldsymbol{x})$ such that

$$\mu' - \mu = -f''(\frac{\boldsymbol{z}}{\mathrm{LLM}_{\theta_{init}}(\boldsymbol{x})}) \left( \frac{\mathrm{LLM}_{\theta'}(\boldsymbol{x}) - \mathrm{LLM}_\theta(\boldsymbol{x})}{\mathrm{LLM}_{\theta_{init}}(\boldsymbol{x})} \right).$$

For convenience, we let $\mathrm{LLM}_{\theta''}(\boldsymbol{x})$ refer to the corresponding $\boldsymbol{z}$ for $\boldsymbol{x}$, note that $\mathrm{LLM}_{\theta''}$ is not necessarily a distribution. We then compute the change in the group $i$'s utility:

$$\sum_{\boldsymbol{x} \in T^*} \mathrm{rm}_i(\boldsymbol{x})(\mathrm{LLM}_{\theta'}(\boldsymbol{x}) - \mathrm{LLM}_\theta(\boldsymbol{x}))$$

$$= \mathrm{rm}_i(\boldsymbol{x}_1)(\mathrm{LLM}_{\theta'}(\boldsymbol{x}_1) - \mathrm{LLM}_\theta(\boldsymbol{x}_1)) + \sum_{\boldsymbol{x} \neq \boldsymbol{x}_1} \mathrm{rm}_i(\boldsymbol{x})(\mathrm{LLM}_{\theta'}(\boldsymbol{x}) - \mathrm{LLM}_\theta(\boldsymbol{x}))$$

$$= \mathrm{rm}_i(\boldsymbol{x}_1) \sum_{\boldsymbol{x} \neq \boldsymbol{x}_1} (\mathrm{LLM}_\theta(\boldsymbol{x}) - \mathrm{LLM}_{\theta'}(\boldsymbol{x})) - \sum_{\boldsymbol{x} \neq \boldsymbol{x}_1} \mathrm{rm}_i(\boldsymbol{x})(\mathrm{LLM}_\theta(\boldsymbol{x}) - \mathrm{LLM}_{\theta'}(\boldsymbol{x}))$$

$$= \sum_{\boldsymbol{x} \neq \boldsymbol{x}_1} \frac{(\mu' - \mu)(\mathrm{rm}_i(\boldsymbol{x}_1) - \mathrm{rm}_i(\boldsymbol{x}))\mathrm{LLM}_{\theta_{init}}(\boldsymbol{x})}{f''(\frac{\mathrm{LLM}_{\theta''}(\boldsymbol{x})}{\mathrm{LLM}_{\theta_{init}}(\boldsymbol{x})})}.$$

Then, we show that when the $\epsilon$ we choose is sufficiently small, the above term is positive. We define the lower bound:

$$\delta_1 := \min_{\boldsymbol{x} \in T^*} f''(\frac{\mathrm{LLM}_\theta(\boldsymbol{x})}{\mathrm{LLM}_{\theta_{init}}(\boldsymbol{x})}).$$

Since function $f$ is $\alpha$-strongly convex, $\delta_1 \geq \alpha > 0$. By the $C^2$-smoothness of the $f$, there exists an $\delta_2 > 0$, such that for each $\theta, \theta'$ satisfying $\max_{\boldsymbol{x}} |\mathrm{LLM}_\theta(\boldsymbol{x}) - \mathrm{LLM}_{\theta'}(\boldsymbol{x})| < \delta_2$, we have

$$\max_{\boldsymbol{x} \in T^*} \left| f''(\frac{\mathrm{LLM}_\theta(\boldsymbol{x})}{\mathrm{LLM}_{\theta_{init}}(\boldsymbol{x})}) - f''(\frac{\mathrm{LLM}_{\theta'}(\boldsymbol{x})}{\mathrm{LLM}_{\theta_{init}}(\boldsymbol{x})}) \right| \leq \min\{\frac{\delta_1}{2}, \frac{\delta_1^2 t(\boldsymbol{x}_1)}{4|T^*|}\}.$$

Besides, because of the Definition 4.7, there exists $\delta_3$, such that for each $(\vec{w}, \overrightarrow{\mathrm{rm}})$ and $(\vec{w}', \overrightarrow{\mathrm{rm}}')$ satisfying $\max_{\boldsymbol{x} \in T^*} |\sum_{i=1}^n w_i \mathrm{rm}_i(\boldsymbol{x}) - \sum_{i=1}^n w_i' \mathrm{rm}_i'(\boldsymbol{x})| \le \delta_3$, we have $\max_{\boldsymbol{x}} |\mathrm{LLM}_\theta(\boldsymbol{x}) - \mathrm{LLM}_{\theta'}(\boldsymbol{x})| < \delta_2$.

Combining these, we set $\epsilon < \frac{\delta_3}{w_i}$, then it is suffice to show that

$$\left| \sum_{\boldsymbol{x} \ne \boldsymbol{x}_1} \frac{(\mathrm{rm}_i(\boldsymbol{x}_1) - \mathrm{rm}_i(\boldsymbol{x}))\mathrm{LLM}_{\theta_{\text{init}}}(\boldsymbol{x})}{f''(\frac{\mathrm{LLM}_{\theta''}(\boldsymbol{x})}{\mathrm{LLM}_{\theta_{\text{init}}}(\boldsymbol{x})})} - \sum_{\boldsymbol{x} \ne \boldsymbol{x}_1} \frac{(\mathrm{rm}_i(\boldsymbol{x}_1) - \mathrm{rm}_i(\boldsymbol{x}))\mathrm{LLM}_{\theta_{\text{init}}}(\boldsymbol{x})}{f''(\frac{\mathrm{LLM}_\theta(\boldsymbol{x})}{\mathrm{LLM}_{\theta_{\text{init}}}(\boldsymbol{x})})} \right|$$

$$= \left| \sum_{\boldsymbol{x} \ne \boldsymbol{x}_1} \frac{(\mathrm{rm}_i(\boldsymbol{x}_1) - \mathrm{rm}_i(\boldsymbol{x})) \left( f''(\frac{\mathrm{LLM}_\theta(\boldsymbol{x})}{\mathrm{LLM}_{\theta_{\text{init}}}(\boldsymbol{x})}) - f''(\frac{\mathrm{LLM}_{\theta''}(\boldsymbol{x})}{\mathrm{LLM}_{\theta_{\text{init}}}(\boldsymbol{x})}) \right) \mathrm{LLM}_{\theta_{\text{init}}}(\boldsymbol{x})}{f''(\frac{\mathrm{LLM}_{\theta''}(\boldsymbol{x})}{\mathrm{LLM}_{\theta_{\text{init}}}(\boldsymbol{x})}) \cdot f''(\frac{\mathrm{LLM}_\theta(\boldsymbol{x})}{\mathrm{LLM}_{\theta_{\text{init}}}(\boldsymbol{x})})} \right|$$

$$\le \sum_{\boldsymbol{x} \ne \boldsymbol{x}_1} \frac{|\mathrm{rm}_i(\boldsymbol{x}_1) - \mathrm{rm}_i(\boldsymbol{x})| \left| f''(\frac{\mathrm{LLM}_\theta(\boldsymbol{x})}{\mathrm{LLM}_{\theta_{\text{init}}}(\boldsymbol{x})}) - f''(\frac{\mathrm{LLM}_{\theta''}(\boldsymbol{x})}{\mathrm{LLM}_{\theta_{\text{init}}}(\boldsymbol{x})}) \right| \mathrm{LLM}_{\theta_{\text{init}}}(\boldsymbol{x})}{f''(\frac{\mathrm{LLM}_{\theta''}(\boldsymbol{x})}{\mathrm{LLM}_{\theta_{\text{init}}}(\boldsymbol{x})}) \cdot f''(\frac{\mathrm{LLM}_\theta(\boldsymbol{x})}{\mathrm{LLM}_{\theta_{\text{init}}}(\boldsymbol{x})})}$$

$$< |T^*| \cdot \frac{\delta_1^2 t(\boldsymbol{x}_1)}{4|T^*|} \cdot \frac{2}{\delta_1 \cdot \delta_1} = \frac{t(\boldsymbol{x}_1)}{2}.$$

This means that

$$\sum_{\boldsymbol{x} \ne \boldsymbol{x}_1} \frac{(\mathrm{rm}_i(\boldsymbol{x}_1) - \mathrm{rm}_i(\boldsymbol{x}))\mathrm{LLM}_{\theta_{\text{init}}}(\boldsymbol{x})}{f''(\frac{\mathrm{LLM}_{\theta''}(\boldsymbol{x})}{\mathrm{LLM}_{\theta_{\text{init}}}(\boldsymbol{x})})} > \sum_{\boldsymbol{x} \ne \boldsymbol{x}_1} \frac{(\mathrm{rm}_i(\boldsymbol{x}_1) - \mathrm{rm}_i(\boldsymbol{x}))\mathrm{LLM}_{\theta_{\text{init}}}(\boldsymbol{x})}{f''(\frac{\mathrm{LLM}_\theta(\boldsymbol{x})}{\mathrm{LLM}_{\theta_{\text{init}}}(\boldsymbol{x})})} - \frac{t(\boldsymbol{x}_1)}{2}$$

$$= t(\boldsymbol{x}_1) - \frac{t(\boldsymbol{x}_1)}{2} = \frac{t(\boldsymbol{x}_1)}{2} > 0.$$

Combined with $\mu' > \mu$, the proof concludes.

**The Summation Normalization Case (2).** Assume that there exists $\boldsymbol{x}_1$ such that $t(\boldsymbol{x}_1) < 0$, we select $\boldsymbol{x}_2 := \arg\max_{\boldsymbol{x} \in T^*} \mathrm{rm}_i(\boldsymbol{x})$ and take $0 < \epsilon < \min\{\mathrm{rm}_i(\boldsymbol{x}_1), 1 - \mathrm{rm}_i(\boldsymbol{x}_2)\}$ to construct a report $\mathrm{rm}_i'$

$$\mathrm{rm}_i'(\boldsymbol{x}) = \begin{cases} \mathrm{rm}_i(\boldsymbol{x}) - \epsilon, & \boldsymbol{x} = \boldsymbol{x}_1, \\ \mathrm{rm}_i(\boldsymbol{x}) + \epsilon, & \boldsymbol{x} = \boldsymbol{x}_2, \\ \mathrm{rm}_i(\boldsymbol{x}), & \boldsymbol{x} \notin \{\boldsymbol{x}_1, \boldsymbol{x}_2\}. \end{cases}$$

Still, we use $\mu'$ to denote the Lagrangian variable for the optimal solution $\theta'$ when reporting $\mathrm{rm}_i'$. Then, there are two possibilities for the relationship between $\mu$ and $\mu'$. If $\mu \le \mu'$, by the optimal condition OPT, for all $\boldsymbol{x} \ne \boldsymbol{x}_2$, we have $\mathrm{LLM}_\theta(\boldsymbol{x}) \ge \mathrm{LLM}_{\theta'}(\boldsymbol{x})$. Since $\boldsymbol{x}_2$ has the highest reward value, such a change in the training outcome must be more preferred by the group $i$. Therefore, we only have to consider the case that $\mu > \mu'$. Similarly, in this case, for all $\boldsymbol{x} \ne \boldsymbol{x}_1$, we have $\mathrm{LLM}_\theta(\boldsymbol{x}) < \mathrm{LLM}_{\theta'}(\boldsymbol{x})$. By the $C^2$-smoothness of $f$, for each $\boldsymbol{x} \ne \boldsymbol{x}_1$, there exits a $\mathrm{LLM}_\theta(\boldsymbol{x}) \le z \le \mathrm{LLM}_{\theta'}(\boldsymbol{x})$ such that

$$\mu' - \mu = w_i \epsilon \mathbb{I}_{\boldsymbol{x} = \boldsymbol{x}_2} - f''(\frac{z}{\mathrm{LLM}_{\theta_{\text{init}}}(\boldsymbol{x})})(\frac{\mathrm{LLM}_{\theta'}(\boldsymbol{x}) - \mathrm{LLM}_\theta(\boldsymbol{x})}{\mathrm{LLM}_{\theta_{\text{init}}}(\boldsymbol{x})}).$$

Let $\mathrm{LLM}_{\theta''}(\boldsymbol{x})$ refer to the corresponding $z$ for $\boldsymbol{x}$, we then compute the change in the group $i$'s utility:

$$\sum_{\boldsymbol{x} \in T^*} \mathrm{rm}_i(\boldsymbol{x})(\mathrm{LLM}_{\theta'}(\boldsymbol{x}) - \mathrm{LLM}_\theta(\boldsymbol{x}))$$

$$= \mathrm{rm}_i(\boldsymbol{x}_1)(\mathrm{LLM}_{\theta'}(\boldsymbol{x}_1) - \mathrm{LLM}_\theta(\boldsymbol{x}_1)) + \sum_{\boldsymbol{x} \ne \boldsymbol{x}_1} \mathrm{rm}_i(\boldsymbol{x})(\mathrm{LLM}_{\theta'}(\boldsymbol{x}) - \mathrm{LLM}_\theta(\boldsymbol{x}))$$

$$= \mathrm{rm}_i(\boldsymbol{x}_1) \sum_{\boldsymbol{x} \ne \boldsymbol{x}_1} (\mathrm{LLM}_\theta(\boldsymbol{x}) - \mathrm{LLM}_{\theta'}(\boldsymbol{x})) - \sum_{\boldsymbol{x} \ne \boldsymbol{x}_1} \mathrm{rm}_i(\boldsymbol{x})(\mathrm{LLM}_\theta(\boldsymbol{x}) - \mathrm{LLM}_{\theta'}(\boldsymbol{x}))$$

$$= \sum_{\boldsymbol{x} \ne \boldsymbol{x}_1} \frac{(\mu' - \mu)(\mathrm{rm}_i(\boldsymbol{x}_1) - \mathrm{rm}_i(\boldsymbol{x}))\mathrm{LLM}_{\theta_{\text{init}}}(\boldsymbol{x})}{f''(\frac{\mathrm{LLM}_{\theta''}(\boldsymbol{x})}{\mathrm{LLM}_{\theta_{\text{init}}}(\boldsymbol{x})})} - w_i \epsilon \frac{(\mathrm{rm}_i(\boldsymbol{x}_1) - \mathrm{rm}_i(\boldsymbol{x}_2))\mathrm{LLM}_{\theta_{\text{init}}}(\boldsymbol{x}_2)}{f''(\frac{\mathrm{LLM}_{\theta''}(\boldsymbol{x}_2)}{\mathrm{LLM}_{\theta_{\text{init}}}(\boldsymbol{x}_2)})}$$

$$\ge \sum_{\boldsymbol{x} \ne \boldsymbol{x}_1} \frac{(\mu' - \mu)(\mathrm{rm}_i(\boldsymbol{x}_1) - \mathrm{rm}_i(\boldsymbol{x}))\mathrm{LLM}_{\theta_{\text{init}}}(\boldsymbol{x})}{f''(\frac{\mathrm{LLM}_{\theta''}(\boldsymbol{x})}{\mathrm{LLM}_{\theta_{\text{init}}}(\boldsymbol{x})})}.$$

With the same technique we used in the maximum normalized case (1), we can show that with sufficient small $\epsilon > 0$, the above term $\sum_{\boldsymbol{x} \neq \boldsymbol{x}_1} \frac{(\mathrm{rm}_i(\boldsymbol{x}_1) - \mathrm{rm}_i(\boldsymbol{x}))\mathrm{LLM}_{\theta_{\mathrm{init}}}(\boldsymbol{x})}{f''(\frac{\mathrm{LLM}_{\theta''}(\boldsymbol{x})}{\mathrm{LLM}_{\theta_{\mathrm{init}}}(\boldsymbol{x})})} < \frac{t(\boldsymbol{x}_1)}{2} < 0$. Combined with $\mu' < \mu$, the proof concludes.

**The Summation Normalization Case (3).** Assume that there exists $\boldsymbol{x}_1$ such that $t(\boldsymbol{x}_1) > 0$, and $\boldsymbol{x}_2$, $\mathrm{rm}_i(\boldsymbol{x}_1) \geq \mathrm{rm}_i(\boldsymbol{x}_2) > 0$, we take $0 < \epsilon < \min\{\mathrm{rm}_i(\boldsymbol{x}_2), 1 - \mathrm{rm}_i(\boldsymbol{x}_1)\}$ to construct a report $\mathrm{rm}_i'$

$$\mathrm{rm}_i'(\boldsymbol{x}) = \begin{cases} \mathrm{rm}_i(\boldsymbol{x}) + \epsilon, & \boldsymbol{x} = \boldsymbol{x}_1, \\ \mathrm{rm}_i(\boldsymbol{x}) - \epsilon, & \boldsymbol{x} = \boldsymbol{x}_2, \\ \mathrm{rm}_i(\boldsymbol{x}), & \boldsymbol{x} \notin \{\boldsymbol{x}_1, \boldsymbol{x}_2\}. \end{cases}$$

Still, we use $\mu'$ to denote the Lagrangian variable for the optimal solution $\theta'$ when reporting $\mathrm{rm}_i'$. Since we know for sure that $\mathrm{LLM}_\theta(\boldsymbol{x}_1) < \mathrm{LLM}_{\theta'}(\boldsymbol{x}_1)$ and $\mathrm{LLM}_\theta(\boldsymbol{x}_2) > \mathrm{LLM}_{\theta'}(\boldsymbol{x}_2)$, by the $C^2$-smoothness of $f$, $\mathrm{LLM}_{\theta'}(\boldsymbol{x}_2) \leq \mathrm{LLM}_{\theta''}(\boldsymbol{x}_2) \leq \mathrm{LLM}_\theta(\boldsymbol{x}_2)$ and $\mathrm{LLM}_\theta(\boldsymbol{x}_1) \leq \mathrm{LLM}_{\theta''}(\boldsymbol{x}_1) \leq \mathrm{LLM}_{\theta'}(\boldsymbol{x}_1)$ such that

$$\begin{aligned} \mu' - \mu &= w_i\epsilon - f''(\frac{\mathrm{LLM}_{\theta''}(\boldsymbol{x}_1)}{\mathrm{LLM}_{\theta_{\mathrm{init}}}(\boldsymbol{x}_1)})\frac{\mathrm{LLM}_{\theta'}(\boldsymbol{x}_1) - \mathrm{LLM}_\theta(\boldsymbol{x}_1)}{\mathrm{LLM}_{\theta_{\mathrm{init}}}(\boldsymbol{x}_1)}, \\ \mu' - \mu &= -w_i\epsilon - f''(\frac{\mathrm{LLM}_{\theta''}(\boldsymbol{x}_2)}{\mathrm{LLM}_{\theta_{\mathrm{init}}}(\boldsymbol{x}_2)})\frac{\mathrm{LLM}_{\theta'}(\boldsymbol{x}_2) - \mathrm{LLM}_\theta(\boldsymbol{x}_2)}{\mathrm{LLM}_{\theta_{\mathrm{init}}}(\boldsymbol{x}_2)}. \end{aligned} \tag{2}$$

Let $\delta_1 := \min_{\boldsymbol{x}} \mathrm{LLM}_{\theta_{\mathrm{init}}}(\boldsymbol{x})$, by the $C^2$-smoothness of the $f$, there exists an $\delta_2 > 0$, such that for each $\theta, \theta'$ satisfying $\max_{\boldsymbol{x}} |w_i\mathrm{rm}_i(\boldsymbol{x}) - w_i'\mathrm{rm}_i'(\boldsymbol{x})| < \delta_2$, we have

$$\max_{\boldsymbol{x} \in T^*} \left| f''(\frac{\mathrm{LLM}_\theta(\boldsymbol{x})}{\mathrm{LLM}_{\theta_{\mathrm{init}}}(\boldsymbol{x})}) - f''(\frac{\mathrm{LLM}_{\theta'}(\boldsymbol{x})}{\mathrm{LLM}_{\theta_{\mathrm{init}}}(\boldsymbol{x})}) \right| \leq \frac{\frac{\delta_1}{\mathrm{LLM}_{\theta_{\mathrm{init}}}(\boldsymbol{x}_2)}f''\left(\frac{\mathrm{LLM}_\theta(\boldsymbol{x}_2)}{\mathrm{LLM}_{\theta_{\mathrm{init}}}(\boldsymbol{x}_2)}\right) - \frac{\delta_1}{\mathrm{LLM}_{\theta_{\mathrm{init}}}(\boldsymbol{x}_1)}f''\left(\frac{\mathrm{LLM}_\theta(\boldsymbol{x}_1)}{\mathrm{LLM}_{\theta_{\mathrm{init}}}(\boldsymbol{x}_1)}\right)}{3}. \tag{3}$$

**We take $\epsilon < \frac{\delta_2}{w_i}$ and first prove that when taking such $\epsilon$, there is $\mu \leq \mu'$.** By contradiction, if $\mu' < \mu$, then by condition Equation (OPT), for all $\boldsymbol{x} \notin \{\boldsymbol{x}_1, \boldsymbol{x}_2\}$, there is $\mathrm{LLM}_{\theta'}(\boldsymbol{x}) > \mathrm{LLM}_\theta(\boldsymbol{x})$. Therefore, $\mathrm{LLM}_{\theta'}(\boldsymbol{x}_1) - \mathrm{LLM}_\theta(\boldsymbol{x}_1) = \sum_{\boldsymbol{x} \notin \{\boldsymbol{x}_1, \boldsymbol{x}_2\}}(\mathrm{LLM}_\theta(\boldsymbol{x}) - \mathrm{LLM}_{\theta'}(\boldsymbol{x})) + \mathrm{LLM}_\theta(\boldsymbol{x}_2) - \mathrm{LLM}_{\theta'}(\boldsymbol{x}_2) < \mathrm{LLM}_\theta(\boldsymbol{x}_2) - \mathrm{LLM}_{\theta'}(\boldsymbol{x}_2)$. However, by Equation (2), if $\mu' < \mu$, we get

$$f''\left(\frac{\mathrm{LLM}_{\theta''}(\boldsymbol{x}_1)}{\mathrm{LLM}_{\theta_{\mathrm{init}}}(\boldsymbol{x}_1)}\right)\frac{\mathrm{LLM}_{\theta'}(\boldsymbol{x}_1) - \mathrm{LLM}_\theta(\boldsymbol{x}_1)}{\mathrm{LLM}_{\theta_{\mathrm{init}}}(\boldsymbol{x}_1)} > f''\left(\frac{\mathrm{LLM}_{\theta''}(\boldsymbol{x}_2)}{\mathrm{LLM}_{\theta_{\mathrm{init}}}(\boldsymbol{x}_2)}\right)\frac{\mathrm{LLM}_\theta(\boldsymbol{x}_2) - \mathrm{LLM}_{\theta'}(\boldsymbol{x}_2)}{\mathrm{LLM}_{\theta_{\mathrm{init}}}(\boldsymbol{x}_2)}$$

By Equation (3), we can derive that

$$f''\left(\frac{\mathrm{LLM}_{\theta''}(\boldsymbol{x}_1)}{\mathrm{LLM}_{\theta_{\mathrm{init}}}(\boldsymbol{x}_1)}\right)\frac{1}{\mathrm{LLM}_{\theta_{\mathrm{init}}}(\boldsymbol{x}_1)} < f''\left(\frac{\mathrm{LLM}_{\theta''}(\boldsymbol{x}_2)}{\mathrm{LLM}_{\theta_{\mathrm{init}}}(\boldsymbol{x}_2)}\right)\frac{1}{\mathrm{LLM}_{\theta_{\mathrm{init}}}(\boldsymbol{x}_2)},$$

and thus, we get

$$\mathrm{LLM}_{\theta'}(\boldsymbol{x}_1) - \mathrm{LLM}_\theta(\boldsymbol{x}_1) > \mathrm{LLM}_\theta(\boldsymbol{x}_2) - \mathrm{LLM}_{\theta'}(\boldsymbol{x}_2),$$

which brings the contradiction.

**After proving that $\mu \leq \mu'$, we know that for all $\boldsymbol{x} \notin \{\boldsymbol{x}_1, \boldsymbol{x}_2\}$, $\mathrm{LLM}_\theta(\boldsymbol{x}) \geq \mathrm{LLM}_{\theta'}(\boldsymbol{x})$.** Then, by the $C^2$-smoothness of $f$, for each $\boldsymbol{x} \neq \boldsymbol{x}_1$, there exits a $\mathrm{LLM}_{\theta'}(\boldsymbol{x}) \leq \boldsymbol{z} \leq \mathrm{LLM}_\theta(\boldsymbol{x})$ such that

$$\mu' - \mu = -w_i\epsilon\mathbb{I}_{\boldsymbol{x}=\boldsymbol{x}_2} - f''(\frac{\boldsymbol{z}}{\mathrm{LLM}_{\theta_{\mathrm{init}}}(\boldsymbol{x})})(\frac{\mathrm{LLM}_{\theta'}(\boldsymbol{x}) - \mathrm{LLM}_\theta(\boldsymbol{x})}{\mathrm{LLM}_{\theta_{\mathrm{init}}}(\boldsymbol{x})}).$$

Let $\mathrm{LLM}_{\theta''}(\boldsymbol{x})$ refer to the corresponding $\boldsymbol{z}$ for $\boldsymbol{x}$, we then compute the change in the group $i$'s utility:

$$\sum_{\boldsymbol{x}\in T^*} \mathrm{rm}_i(\boldsymbol{x})(\mathrm{LLM}_{\theta'}(\boldsymbol{x}) - \mathrm{LLM}_\theta(\boldsymbol{x}))$$

$$=\mathrm{rm}_i(\boldsymbol{x}_1)(\mathrm{LLM}_{\theta'}(\boldsymbol{x}_1) - \mathrm{LLM}_\theta(\boldsymbol{x}_1)) + \sum_{\boldsymbol{x}\neq\boldsymbol{x}_1} \mathrm{rm}_i(\boldsymbol{x})(\mathrm{LLM}_{\theta'}(\boldsymbol{x}) - \mathrm{LLM}_\theta(\boldsymbol{x}))$$

$$=\mathrm{rm}_i(\boldsymbol{x}_1)\sum_{\boldsymbol{x}\neq\boldsymbol{x}_1} (\mathrm{LLM}_\theta(\boldsymbol{x}) - \mathrm{LLM}_{\theta'}(\boldsymbol{x})) - \sum_{\boldsymbol{x}\neq\boldsymbol{x}_1} \mathrm{rm}_i(\boldsymbol{x})(\mathrm{LLM}_\theta(\boldsymbol{x}) - \mathrm{LLM}_{\theta'}(\boldsymbol{x}))$$

$$= \sum_{\boldsymbol{x}\neq\boldsymbol{x}_1} \frac{(\mu'-\mu)(\mathrm{rm}_i(\boldsymbol{x}_1) - \mathrm{rm}_i(\boldsymbol{x}))\mathrm{LLM}_{\theta_{\mathrm{init}}}(\boldsymbol{x})}{f''(\frac{\mathrm{LLM}_{\theta''}(\boldsymbol{x})}{\mathrm{LLM}_{\theta_{\mathrm{init}}}(\boldsymbol{x})})} + w_i\epsilon\frac{(\mathrm{rm}_i(\boldsymbol{x}_1) - \mathrm{rm}_i(\boldsymbol{x}_2))\mathrm{LLM}_{\theta_{\mathrm{init}}}(\boldsymbol{x}_2)}{f''(\frac{\mathrm{LLM}_{\theta''}(\boldsymbol{x}_2)}{\mathrm{LLM}_{\theta_{\mathrm{init}}}(\boldsymbol{x}_2)})}$$

$$\geq \sum_{\boldsymbol{x}\neq\boldsymbol{x}_1} \frac{(\mu'-\mu)(\mathrm{rm}_i(\boldsymbol{x}_1) - \mathrm{rm}_i(\boldsymbol{x}))\mathrm{LLM}_{\theta_{\mathrm{init}}}(\boldsymbol{x})}{f''(\frac{\mathrm{LLM}_{\theta''}(\boldsymbol{x})}{\mathrm{LLM}_{\theta_{\mathrm{init}}}(\boldsymbol{x})})}.$$

With the same technique we used in the maximum normalized case (1), we can show that with sufficient small $\epsilon > 0$, the above term $\sum_{\boldsymbol{x}\neq\boldsymbol{x}_1} \frac{(\mathrm{rm}_i(\boldsymbol{x}_1) - \mathrm{rm}_i(\boldsymbol{x}))\mathrm{LLM}_{\theta_{\mathrm{init}}}(\boldsymbol{x})}{f''(\frac{\mathrm{LLM}_{\theta''}(\boldsymbol{x})}{\mathrm{LLM}_{\theta_{\mathrm{init}}}(\boldsymbol{x})})} > \frac{t(\boldsymbol{x}_1)}{2} > 0$. Combined with $\mu' < \mu$, the proof concludes. $\qquad\square$

## C  Omitted Proofs in Section 4.2

**Proposition 4.4.** *For any $\psi \in \Psi^{SW}$, mechanism $(\psi, p^{AFF})$ satisfies DSIC and IR, and the payment is non-negative.*

*Proof.* We assume that for group $i$, the true reward model is $\mathrm{rm}_i$, and the agent number is $w_i$. The reports of other groups are $(\overrightarrow{\mathrm{rm}}_{-i}, \vec{w}_{-i})$ and the initial model is $\theta_{\mathrm{init}}$.

(1) $(\psi, p^{AFF})$ satisfies DSIC.

We compare the utility between reporting $(\mathrm{rm}_i, w_i)$ and any other $(\mathrm{rm}'_i, w'_i)$. For convenience, we first simplify the notations by letting

$$\theta = \psi(((\mathrm{rm}_i, \overrightarrow{\mathrm{rm}}_{-i}), (w_i, \vec{w}_{-i})), \theta_{\mathrm{init}}),$$
$$\theta' = \psi(((\mathrm{rm}'_i, \overrightarrow{\mathrm{rm}}_{-i}), (w'_i, \vec{w}_{-i})), \theta_{\mathrm{init}}).$$

The valuation of group $i$ is the valuation for each agent multiplied by the real agent number:

$$v_i = w_i v_i(\theta; \mathrm{rm}_i),$$
$$v'_i = w_i v_i(\theta'; \mathrm{rm}_i).$$

According to the payment rule $p^{AFF}$, the payment $p_i$ for $(\mathrm{rm}_i, w_i)$ and $p'_i$ for $(\mathrm{rm}'_i, w'_i)$ is

$$p_i = \mathrm{ASW}_{-i}(\psi(\overrightarrow{\mathrm{rm}}_{-i}, \vec{w}_{-i}, \theta_{\mathrm{init}}); \overrightarrow{\mathrm{rm}}_{-i}, \vec{w}_{-i}, \theta_{\mathrm{init}}) - \mathrm{ASW}_{-i}(\theta; \overrightarrow{\mathrm{rm}}_{-i}, \vec{w}_{-i}, \theta_{\mathrm{init}})$$
$$p'_i = \mathrm{ASW}_{-i}(\psi(\overrightarrow{\mathrm{rm}}_{-i}, \vec{w}_{-i}, \theta_{\mathrm{init}}); \overrightarrow{\mathrm{rm}}_{-i}, \vec{w}_{-i}, \theta_{\mathrm{init}}) - \mathrm{ASW}_{-i}(\theta'; \overrightarrow{\mathrm{rm}}_{-i}, \vec{w}_{-i}, \theta_{\mathrm{init}})$$

Therefore, we can calculate the change in the utility:

$$\begin{aligned}
u'_i - u_i &= (v'_i - p'_i) - (v_i - p_i) \\
&= \left(w_i v_i(\theta'; \mathrm{rm}_i) + \mathrm{ASW}_{-i}(\theta'; \overrightarrow{\mathrm{rm}}_{-i}, \vec{w}_{-i}, \theta_{\mathrm{init}})\right) \\
&\quad - \left(w_i v_i(\theta; \mathrm{rm}_i) + \mathrm{ASW}_{-i}(\theta; \overrightarrow{\mathrm{rm}}_{-i}, \vec{w}_{-i}, \theta_{\mathrm{init}})\right) \\
&= \mathrm{ASW}((\theta'; (\mathrm{rm}_i, \overrightarrow{\mathrm{rm}}_{-i}), (w_i, \vec{w}_{-i})), \theta_{\mathrm{init}}) - \mathrm{ASW}((\theta; (\mathrm{rm}_i, \overrightarrow{\mathrm{rm}}_{-i}), (w_i, \vec{w}_{-i})), \theta_{\mathrm{init}}) \\
&\leq 0.
\end{aligned}$$

The last inequality holds by the definition of $\theta$

$$\theta = \psi(((\mathrm{rm}_i, \overrightarrow{\mathrm{rm}}_{-i}), (w_i, \vec{w}_{-i})), \theta_{\mathrm{init}}) = \arg\max_{\hat{\theta}\in\Theta} \mathrm{ASW}((\hat{\theta}; (\mathrm{rm}_i, \overrightarrow{\mathrm{rm}}_{-i}), (w_i, \vec{w}_{-i})), \theta_{\mathrm{init}}).$$

Therefore, we can conclude that, for all $\overrightarrow{\text{rm}}$, $\vec{w}$, $\text{rm}'_i$, $w'_i$, we have

$$u_i((\overrightarrow{\text{rm}}, \vec{w}); \psi, p^{AFF}, \text{rm}_i, w_i) \geq u_i((\text{rm}'_i, \overrightarrow{\text{rm}}_{-i}), (w'_i, \vec{w}_{-i}); \psi, p^{AFF}, \text{rm}_i, w_i).$$

(2) $(\psi, p^{AFF})$ satisfies IR.

We reuse the notations above and denote $\theta_{-i}$ to be the optimal parameter for groups except for $i$, i.e. $\theta_{-i} = \psi(\overrightarrow{\text{rm}}_{-i}, \vec{w}_{-i}, \theta_{\text{init}})$. When group $i$ truthfully report its reward model $\text{rm}_i$ and agent number $w_i$, the utility can be written as:

$$
\begin{aligned}
u_i &= v_i - p_i \\
&= w_i v_i(\theta; \text{rm}_i) - \text{ASW}_{-i}(\theta_{-i}; \overrightarrow{\text{rm}}_{-i}, \vec{w}_{-i}, \theta_{\text{init}}) + \text{ASW}_{-i}(\theta; \overrightarrow{\text{rm}}_{-i}, \vec{w}_{-i}, \theta_{\text{init}}) \\
&= w_i v_i(\theta; \text{rm}_i) + \text{ASW}_{-i}(\theta; \overrightarrow{\text{rm}}_{-i}, \vec{w}_{-i}, \theta_{\text{init}}) - \text{ASW}_{-i}(\theta_{-i}; \overrightarrow{\text{rm}}_{-i}, \vec{w}_{-i}, \theta_{\text{init}}) \\
&= \text{ASW}(\theta; \overrightarrow{\text{rm}}, \vec{w}, \theta_{\text{init}}) - \text{ASW}_{-i}(\theta_{-i}; \overrightarrow{\text{rm}}_{-i}, \vec{w}_{-i}, \theta_{\text{init}}) \\
&\geq \text{ASW}(\theta_{-i}; \overrightarrow{\text{rm}}, \vec{w}, \theta_{\text{init}}) - \text{ASW}_{-i}(\theta_{-i}; \overrightarrow{\text{rm}}_{-i}, \vec{w}_{-i}, \theta_{\text{init}}) \\
&= w_i v_i(\theta_{-i}; \text{rm}_i) + \text{ASW}_{-i}(\theta_{-i}; \overrightarrow{\text{rm}}, \vec{w}, \theta_{\text{init}}) - \text{ASW}_{-i}(\theta_{-i}; \overrightarrow{\text{rm}}_{-i}, \vec{w}_{-i}, \theta_{\text{init}}) \\
&= w_i v_i(\theta_{-i}; \text{rm}_i) \geq 0.
\end{aligned}
$$

Therefore, we can conclude that, for all $\overrightarrow{\text{rm}}$, $\vec{w}$, we have

$$u_i((\overrightarrow{\text{rm}}, \vec{w}); \psi, p^{AFF}, \text{rm}_i, w_i) \geq 0.$$

$\square$

**Proposition 4.6.** *When $\vec{w} \equiv 1$ is public information, and the agents only report the reward models, all implementable training rules satisfy payment equivalence.*

*Proof.* We follow the result Theorem 1.37 in Nisan et al. [55].

**Lemma C.1** (Theorem 1.37 in Nisan et al. [55])**.** *Let $\mathcal{R}_i$ be group $i$'s preference domain. Assume that the $\mathcal{R}_1, \mathcal{R}_2, \ldots, \mathcal{R}_n$ are connected sets in the Euclidean space, then all implementable training rules $\psi$ satisfy payment equivalence.*

In our paper, we assume that for all $i \in [n]$, $\mathcal{R}_i$ is the set of all non-negative and normalized $|T^*|$-dim vectors. Either in the summation normalization case or the maximum normalization case, this is a connected set in the Euclidean space. Hence, the theorem holds. $\square$

**Proposition 4.8.** *SW-Max training rules with regularizations KL-divergence, $f_{\text{KL}}(x) = \lambda x \log x$, and $\chi^2$ divergence, $f_2(x) = \lambda(x-1)^2$ ($\lambda > 0$ is a constant) are continuous.*

*Proof.* (1) For $f_{\text{KL}}(x) = \lambda x \log x$ (KL-divergence), since $T^*$ is a finite set, we can rewrite the training rule $\psi$ as an optimization problem as follows:

$$
\begin{aligned}
\arg\max_{\text{LLM}_\theta} &\sum_{\boldsymbol{x} \in T^*} \left( \text{LLM}_\theta(\boldsymbol{x}) \sum_{i=1}^n w_i \text{rm}_i(\boldsymbol{x}) - \lambda \text{LLM}_\theta(\boldsymbol{x}) \log \frac{\text{LLM}_\theta(\boldsymbol{x})}{\text{LLM}_{\theta_{\text{init}}}(\boldsymbol{x})} \right) \\
\text{s.t.} \quad &\sum_{\boldsymbol{x} \in T^*} \text{LLM}_\theta(\boldsymbol{x}) = 1 \\
&\text{LLM}_\theta(\boldsymbol{x}) \geq 0 \quad \forall \boldsymbol{x} \in T^*.
\end{aligned}
$$

Since for $KL$ divergence, the optimal model $\text{LLM}_\theta$ must satisfy that $\text{LLM}_\theta(\boldsymbol{x}) > 0$, for all $\boldsymbol{x} \in T^*$. The necessary condition for an optimal $\theta$ is that there exists $\mu \in \mathbb{R}$, such that

$$\sum_{i=1}^n w_i \text{rm}_i(\boldsymbol{x}) - \lambda \log \frac{\text{LLM}_\theta(\boldsymbol{x})}{\text{LLM}_{\theta_{\text{init}}}(\boldsymbol{x})} - \lambda = \mu \quad \forall \boldsymbol{x} \in T^*, \quad \sum_{\boldsymbol{x} \in T^*} \text{LLM}_\theta(\boldsymbol{x}) = 1.$$

Similarly, for the input $(\overrightarrow{\text{rm}}', \vec{w}')$, there exists $\mu' \in \mathbb{R}$, such that the optimal $\theta'$ satisfies

$$\sum_{i=1}^n w'_i \text{rm}'_i(\boldsymbol{x}) - \lambda \log \frac{\text{LLM}_{\theta'}(\boldsymbol{x})}{\text{LLM}_{\theta_{\text{init}}}(\boldsymbol{x})} - \lambda = \mu' \quad \forall \boldsymbol{x} \in T^*, \quad \sum_{\boldsymbol{x} \in T^*} \text{LLM}_{\theta'}(\boldsymbol{x}) = 1.$$

For convenience, we define $\Delta(\boldsymbol{x}) = \sum_{i=1}^{n} w_i' \text{rm}_i'(\boldsymbol{x}) - \sum_{i=1}^{n} w_i \text{rm}_i(\boldsymbol{x})$. Then the relationship between $\text{LLM}_\theta(\boldsymbol{x})$ and $\text{LLM}_{\theta'}(\boldsymbol{x})$ is given by

$$\text{LLM}_{\theta'}(\boldsymbol{x}) = \text{LLM}_\theta(\boldsymbol{x}) e^{\frac{1}{\lambda}(\Delta(\boldsymbol{x}) + \mu - \mu')}.$$

Note that we also have the condition

$$\sum_{\boldsymbol{x} \in T^*} \text{LLM}_{\theta'}(\boldsymbol{x}) = \sum_{\boldsymbol{x} \in T^*} \text{LLM}_\theta(\boldsymbol{x}) e^{\frac{1}{\lambda}(\Delta(\boldsymbol{x}) + \mu - \mu')} = 1.$$

Since $\sum_{\boldsymbol{x} \in T^*} \text{LLM}_\theta(\boldsymbol{x}) e^{\frac{1}{\lambda}(\Delta(\boldsymbol{x}) + \mu - \mu')} = e^{\frac{1}{\lambda}(\mu - \mu')} \sum_{\boldsymbol{x} \in T^*} \text{LLM}_\theta(\boldsymbol{x}) e^{\frac{1}{\lambda}\Delta(\boldsymbol{x})}$, we can infer that

$$1 = e^{\frac{1}{\lambda}(\mu - \mu')} \sum_{\boldsymbol{x} \in T^*} \text{LLM}_\theta(\boldsymbol{x}) e^{\frac{1}{\lambda}\Delta(\boldsymbol{x})} \leq e^{\frac{1}{\lambda}(\mu - \mu')} \max_{\boldsymbol{x} \in T^*} e^{\frac{1}{\lambda}\Delta(\boldsymbol{x})},$$

$$1 = e^{\frac{1}{\lambda}(\mu - \mu')} \sum_{\boldsymbol{x} \in T^*} \text{LLM}_\theta(\boldsymbol{x}) e^{\frac{1}{\lambda}\Delta(\boldsymbol{x})} \geq e^{\frac{1}{\lambda}(\mu - \mu')} \min_{\boldsymbol{x} \in T^*} e^{\frac{1}{\lambda}\Delta(\boldsymbol{x})}.$$

This is equivalent to

$$\min_{\boldsymbol{x} \in T^*} \Delta(\boldsymbol{x}) \leq \mu' - \mu \leq \max_{\boldsymbol{x} \in T^*} \Delta(\boldsymbol{x}).$$

Thus, the difference for $\text{LLM}_\theta(\boldsymbol{x})$ and $\text{LLM}_{\theta'}(\boldsymbol{x})$ can be bounded by

$$\begin{aligned}
|\text{LLM}_{\theta'}(\boldsymbol{x}) - \text{LLM}_\theta(\boldsymbol{x})| &= \left| 1 - e^{\frac{1}{\lambda}(\Delta(\boldsymbol{x}) + \mu - \mu')} \right| \text{LLM}_\theta(\boldsymbol{x}) \\
&\leq \left| 1 - e^{\frac{1}{\lambda}(\Delta(\boldsymbol{x}) + \mu - \mu')} \right| \\
&\leq \max\{ \max_{\boldsymbol{x} \in T^*} e^{\frac{2\Delta(\boldsymbol{x})}{\lambda}} - 1, \max_{\boldsymbol{x} \in T^*} 1 - e^{\frac{2\Delta(\boldsymbol{x})}{\lambda}} \}.
\end{aligned}$$

For any $\delta > 0$, when we set $\max_{\boldsymbol{x} \in T^*} |\Delta(\boldsymbol{x})| \leq \min\{\frac{\lambda}{2} \log \frac{1}{1-\delta}, \frac{\lambda}{2} \log(1 + \delta)\}$, we have

$$|\text{LLM}_{\theta'}(\boldsymbol{x}) - \text{LLM}_\theta(\boldsymbol{x})| \leq \max\{ \max_{\boldsymbol{x} \in T^*} e^{\frac{2\Delta(\boldsymbol{x})}{\lambda}} - 1, \max_{\boldsymbol{x} \in T^*} 1 - e^{\frac{2\Delta(\boldsymbol{x})}{\lambda}} \} \leq \delta.$$

(2) For $f_2(x) = \lambda(x - 1)^2$ ($\chi^2$ divergence), since $T^*$ is a finite set, we can rewrite the training rule $\psi$ as an optimization problem as follows:

$$\arg\max_{\text{LLM}_\theta} \sum_{x \in T^*} \left( \text{LLM}_\theta(x) \sum_{i=1}^{n} w_i \text{rm}_i(x) - \lambda \frac{(\text{LLM}_\theta(x) - \text{LLM}_{\theta_{\text{init}}}(x))^2}{\text{LLM}_{\theta_{\text{init}}}(\boldsymbol{x})} \right)$$

$$\text{s.t.} \quad \sum_{x \in T^*} \text{LLM}_\theta(x) = 1$$

$$\text{LLM}_\theta(x) \geq 0 \quad \forall x \in T^*.$$

Since we have assumed a relatively large $\lambda$ so that the optimal model $\text{LLM}_\theta$ satisfies that $\text{LLM}_\theta(x) > 0$, for all $x \in T^*$. The necessary condition for an optimal $\theta$ is that there exists $\mu \in \mathbb{R}$, such that

$$\sum_{i=1}^{n} w_i \text{rm}_i(x) - 2\lambda \frac{\text{LLM}_\theta(x) - \text{LLM}_{\theta_{\text{init}}}(x)}{\text{LLM}_{\theta_{\text{init}}}(x)} = \mu \quad \forall x \in T^*, \quad \sum_{\boldsymbol{x} \in T^*} \text{LLM}_\theta(\boldsymbol{x}) = 1.$$

Similarly, for the input $(\overrightarrow{\text{rm}}', \vec{w}')$, there exists $\mu' \in \mathbb{R}$, such that the optimal $\theta'$ satisfies

$$\sum_{i=1}^{n} w_i' \text{rm}_i'(x) - 2\lambda \frac{\text{LLM}_{\theta'}(x) - \text{LLM}_{\theta_{\text{init}}}(x)}{\text{LLM}_{\theta_{\text{init}}}(x)} = \mu' \quad \forall x \in T^*, \quad \sum_{\boldsymbol{x} \in T^*} \text{LLM}_{\theta'}(\boldsymbol{x}) = 1.$$

For convenience, we define $\Delta(x) = \sum_{i=1}^{n} w_i' \text{rm}_i'(x) - \sum_{i=1}^{n} w_i \text{rm}_i(x)$ Then the relationship between $\text{LLM}_\theta(x)$ and $\text{LLM}_{\theta'}(x)$ is given by

$$\text{LLM}_{\theta'}(x) = \text{LLM}_\theta(x) + \frac{\text{LLM}_{\theta_{\text{init}}}(x)}{2\lambda} (\Delta(x) + \mu - \mu').$$

Note that we also have the condition

$$\sum_{x \in T^*} \text{LLM}_{\theta'}(x) = \sum_{x \in T^*} \text{LLM}_{\theta}(x) + \frac{\text{LLM}_{\theta_{\text{init}}}(x)}{2\lambda}(\Delta(x) + \mu - \mu') = 1.$$

Since $\sum_{x \in T^*} \text{LLM}_{\theta}(x) = 1$, we can infer that

$$\sum_{x \in T^*} \frac{\text{LLM}_{\theta_{\text{init}}}(x)}{2\lambda}(\Delta(x) + \mu - \mu') = 0.$$

This is equivalent to

$$\mu' - \mu = \sum_{x \in T^*} \text{LLM}_{\theta_{\text{init}}}(x)\Delta(x).$$

Thus, the difference for $\text{LLM}_{\theta}(x)$ and $\text{LLM}_{\theta'}(x)$ can be bounded by

$$|\text{LLM}_{\theta'}(x) - \text{LLM}_{\theta}(x)| = \left| \frac{\text{LLM}_{\theta_{\text{init}}}(x)}{2\lambda}(\Delta(x) + \mu - \mu') \right| \leq \frac{1}{\lambda} \max_{x \in T^*} |\Delta(x)|$$

For any $\delta > 0$, when we set $\max_{x \in T^*} |\Delta(x)| \leq \lambda\delta$, we have

$$|\text{LLM}_{\theta'}(x) - \text{LLM}_{\theta}(x)| \leq \frac{1}{\lambda} \max_{x \in T^*} |\Delta(x)| \leq \delta.$$

$\square$

**Theorem 4.9.** *An implementable training rule $\psi$ satisfies payment equivalence if it is continuous and for $\forall i$, $\overrightarrow{rm}_{-i}$, $\vec{w}_{-i}$, $\theta_{init}$ there exists $rm_i^*$ and $\theta$ such that $\psi((rm_i^*, \overrightarrow{rm}_{-i}), (w_i, \vec{w}_{-i}), \theta_{init}) \equiv \theta$ for all $w_i \in \mathcal{W}$. In the maximum normalization case, $rm_i^*$ must be $\mathbb{1}$.*

*Proof.* We prove the equivalent version of payment equivalence: For any group $i$, when fixing other groups reports $(\overrightarrow{rm}_{-i}, \vec{w}_{-i})$ and $\theta_{\text{init}}$, any two payment rules $p$, $p'$ that implement $\psi$ in DSIC must satisfy that there exists a constant $c$, such that $p_i(rm_i, w_i) - p_i'(rm_i, w_i) = c$ for any $rm_i$ and $w_i$. Therefore, in the rest of the proof, we suppose fixed $(\overrightarrow{rm}_{-i}, \vec{w}_{-i})$ and $\theta_{init}$ and will omit these notations.

Firstly, we introduce a new notation $t_i$ to represent the combination $(rm_i, w_i)$, whose domain is $\mathcal{R} \times \mathcal{W}$. Without specially claim, $t_i$ is used to represented for the $rm_i$ and $w_i$ with the same superscript and subscript, for example, $t_i^k = (rm_i^k, w_i^k)$. Then, we define the functions $l(\cdot, \cdot)$ and $V(\cdot, \cdot)$ as follows. $l(t_i', t_i)$ is the change in valuation from misreporting type $t_i'$ to reporting type $t_i$ truthfully. In formal,

$$l(t_i', t_i) := w_i v_i(\psi(t_i); rm_i) - w_i v_i(\psi(t_i'); rm_i).$$

And $V(t_i', t_i)$ refers to the smallest values of $l$ on a finite and distinct path from $t_i'$ to $t_i$

$$V(t_i', t_i) := \inf_{\substack{\text{A finite and distinct sequence} \\ [t_i^0 := t_i', t_i^1, \dots, t_i^k, t_i^{k+1} := t_i]}} \sum_{j=0}^{k} l(t_i^j, t_i^{j+1}).$$

We prove the following lemma, which is a special case in Heydenreich et al. [38],

**Lemma C.2** (Heydenreich et al. [38]). *In the RLHF Game, an implemented training rule $\psi$ satisfies payment equivalence if for any agent $i$, and any types $t_i$, $t_i'$, we have*

$$V(t_i, t_i') = -V(t_i', t_i).$$

*Proof.* Assume there is a mechanism $(\psi, p)$ that satisfies DSIC. For any two types $t_i$, $t_i'$ and a finite and distinct sequence $[t_i', t_i^1, \dots, t_i^k, t_i]$, let $t_i^0 = t_i'$ and $t_i^{k+1} = t_i$, we have that

$$w_i^{j+1} v_i(\psi(t_i^{j+1}), rm_i^{j+1}) - p_i(t_i^{j+1}) \geq w_i^{j+1} v_i(\psi(t_i^j), rm_i^{j+1}) - p_i(t_i^j) \quad \forall 0 \leq j \leq k.$$

This can be rewritten as

$$w_i^{j+1} v_i(\psi(t_i^{j+1}), rm_i^{j+1}) - w_i^{j+1} v_i(\psi(t_i^j), rm_i^{j+1}) \geq p_i(t_i^{j+1}) - p_i(t_i^j) \quad \forall 0 \leq j \leq k.$$

Sum over $j$, we get the following inequality

$$\sum_{j=0}^{k} l(t_i^j, t_i^{j+1}) = \sum_{j=0}^{k} w_i^{j+1} v_i(\psi(t_i^{j+1}), \mathrm{rm}_i^{j+1}) - w_i^{j+1} v_i(\psi(t_i^j), \mathrm{rm}_i^{j+1})$$

$$\geq \sum_{j=0}^{k} p_i(t_i^{j+1}) - p_i(t_i^j) = p(t_i) - p(t_i').$$

Since this holds for arbitrary finite and distinct sequences, we can infer that $V(t_i', t_i) \geq p(t_i) - p(t_i')$. Similarly, there is $V(t_i, t_i') \geq p(t_i') - p(t_i)$. Combining these results with $V(t_i, t_i') = -V(t_i', t_i)$, there is

$$V(t_i, t_i') = -V(t_i', t_i) \leq p(t_i') - p(t_i) \leq V(t_i, t_i'),$$

which means that $p(t_i') - p(t_i) = V(t_i, t_i')$. Note that this holds for arbitrary $t_i$ and $t_i'$. Therefore, when for some $t_i$, the payment $p(t_i)$ is determined, then the payment for all other $t_i'$s is determined. For example, if there are any two payment rules $p$ and $p'$ both implement $\psi$ in DSIC, and we set the payment when $i$ reports preference rm defined in Equation (5) and $w_i = 1$ as $p^*$ and $p'^*$ respectively, then $\forall t_i$

$$p_i(t_i) - p_i'(t_i)$$
$$= (p_i(t_i) - p^*) - (p_i'(t_i) - p'^*) + p^* - p'^*$$
$$= V((\mathrm{rm}, 1), t_i) - V((\mathrm{rm}, 1), t_i) + p^* - p'^*$$
$$= p^* - p'^*.$$

Note that $p^*$ and $p'^*$ are not influenced by $i$'s report, but they may vary for different $\overrightarrow{\mathrm{rm}}_{-i}$, $\vec{w}_{-i}$ and $\theta_{\mathrm{init}}$, which means that we can consider the term $p^* - p'^*$ as a function $f$ on $(\overrightarrow{\mathrm{rm}}_{-i}, \theta_{\mathrm{init}})$. $\square$

Then, we show that the training rule satisfying the conditions in Theorem 4.9 is sufficient for the condition stated in Lemma C.2. Firstly, we show that for any $t_i, t_i'$, we have $V(t_i, t_i') + V(t_i', t_i) \geq 0$. By definition of the function $V(\cdot, \cdot)$, $V(t_i, t_i')$ and $V(t_i', t_i)$ correspond to the shortest path from $t_i$ to $t_i'$ and from $t_i'$ to $t_i$ respectively, which means that $V(t_i, t_i') + V(t_i', t_i)$ is the shortest weight for a cycle that goes through $t_i$ and $t_i'$. Since the SW-Max training rule is implementable, we know that the weight for any cycle is non-negative by cycle monotonicity [65]. Therefore, $V(t_i, t_i') + V(t_i', t_i) \geq 0$ must be satisfied.

Then we show that for any $t_i, t_i'$ and $\epsilon > 0$, $V(t_i, t_i') + V(t_i', t_i) \leq \epsilon$. We prove this by constructing a finite and distinct sequence $[t_i, t_i^1, \dots, t_i^k, t_i']$ such that

$$\sum_{j=0}^{k} l(t_i^j, t_i^{j+1}) + \sum_{j=0}^{k} l(t_i^{j+1}, t_i^j) \leq \epsilon. \tag{4}$$

This suffices for proving $V(t_i, t_i') + V(t_i', t_i) \leq \epsilon$ since $V(t_i, t_i')$ and $V(t_i', t_i)$ are the lower bound for $\sum_{j=0}^{k} l(t_i^j, t_i^{j+1})$ and $\sum_{j=0}^{k} l(t_i^{j+1}, t_i^j)$ respectively.

Initially, we rewrite the LHS of Equation (4) by using the definition of the function $l(\cdot, \cdot)$.

$$\sum_{j=0}^{k} l(t_i^j, t_i^{j+1}) + \sum_{j=0}^{k} l(t_i^{j+1}, t_i^j)$$

$$= \sum_{j=1}^{k} \left( w_i^{j+1} v_i(\psi(t_i^{j+1}), \mathrm{rm}_i^{j+1}) - w_i^{j+1} v_i(\psi(t_i^j), \mathrm{rm}_i^{j+1}) \right) + \sum_{j=0}^{k} \left( w_i^j v_i(\psi(t_i^j), \mathrm{rm}_i^j) - w_i^j v_i(\psi(t_i^{j+1}), \mathrm{rm}_i^j) \right)$$

$$= \sum_{j=0}^{k} w_i^{j+1} (\mathrm{LLM}_{\theta^{j+1}} - \mathrm{LLM}_{\theta^j}) \cdot \mathrm{rm}_i^{j+1} + \sum_{j=0}^{k} w_i^j (\mathrm{LLM}_{\theta^j} - \mathrm{LLM}_{\theta^{j+1}}) \cdot \mathrm{rm}_i^j$$

$$= \sum_{j=0}^{k} (\mathrm{LLM}_{\theta^{j+1}} - \mathrm{LLM}_{\theta^j}) \cdot (w_i^{j+1} \mathrm{rm}_i^{j+1} - w_i^j \mathrm{rm}_i^j)$$

$$= \sum_{j=0}^{k} \sum_{x \in T^*} (\mathrm{LLM}_{\theta^{j+1}}(x) - \mathrm{LLM}_{\theta^j}(x))(w_i^{j+1} \mathrm{rm}_i^{j+1}(x) - w_i^j \mathrm{rm}_i^j(x)).$$

In the above equations, $\theta^j = \psi(t_i^j)$ for $0 \le j \le k$ refers to the fine-tuned model when group $i$ reports $t_i^j$.

By the condition, when $\overrightarrow{\text{rm}}_{-i}$, $\vec{w}_{-i}$ and $\theta_{\text{init}}$ are fixed, there exits $\delta > 0$ such that if $\max_{x \in T^*} |w_i \text{rm}_i(x) - w_i' \text{rm}_i'(x)| \le \delta$, then $\max_{x \in T^*} |\text{LLM}_\theta(x) - \text{LLM}_{\theta'}(x)| \le \frac{\epsilon}{4\bar{w}}$ (in maximum normalization case, we take $\frac{\epsilon}{4\bar{w}|T^*|}$), where $\theta := \psi((\text{rm}_i, \overrightarrow{\text{rm}}_{-i}), (w_i, \vec{w}_{-i}); \theta_{\text{init}})$ and $\theta' := \psi((\text{rm}_i', \overrightarrow{\text{rm}}_{-i}), (w_i', \vec{w}_{-i}); \theta_{\text{init}})$.

We construct the sequence $P$ as follows: we set $k = 2n$, $n \ge \frac{\bar{w}}{\delta} + 1$ and let $t_i^0 = t_i, t_i^{k+1} = t_i'$. For each $0 \le j \le n$,

$$w_i^j = w_i, \quad \text{rm}_i^j = \text{rm} + j(\frac{\text{rm}_i^* - \text{rm}}{n}).$$

And for each $n + 1 \le j \le 2n + 1$,

$$w_i^j = w_i', \quad \text{rm}_i^j = \text{rm}_i^* + (j - n - 1)(\frac{\text{rm}' - \text{rm}_i^*}{n}).$$

Note that the $\text{rm}_i^*$ is the one given by the condition in Theorem 4.9. In this construction, any $\text{rm}_i^j$ is either an weighted average of rm and $\text{rm}_i^*$ or $\text{rm}'$ and $\text{rm}_i^*$. This ensures that all reward models in the sequence are valid (normalized by summation or maximum and non-negative). We can then divide the above equation into three parts, making the $w_i$ the same in the first and the last parts.

$$\sum_{j=0}^{k} \sum_{x \in T^*} (\text{LLM}_{\theta^{j+1}}(x) - \text{LLM}_{\theta^j}(x))(w_i^{j+1}\text{rm}_i^{j+1}(x) - w_i^j \text{rm}_i^j(x))$$

$$= \sum_{j=0}^{n-1} \sum_{x \in T^*} w_i(\text{LLM}_{\theta^{j+1}}(x) - \text{LLM}_{\theta^j}(x))(\text{rm}_i^{j+1}(x) - \text{rm}_i^j(x)) \tag{a}$$

$$+ \sum_{x \in T^*} (\text{LLM}_{\theta^{n+1}}(x) - \text{LLM}_{\theta^n}(x))(w_i'\text{rm}_i^{n+1}(x) - w_i\text{rm}_i^n(x)) \tag{b}$$

$$+ \sum_{j=n+1}^{2n} \sum_{x \in T^*} w_i'(\text{LLM}_{\theta^{j+1}}(x) - \text{LLM}_{\theta^j}(x))(\text{rm}_i^{j+1}(x) - \text{rm}_i^j(x)) \tag{c}$$

We first claim that (b) equals 0. This is because of the property of $\text{rm}_i^n = \text{rm}_i^{n+1} = \text{rm}_i^*$, which can induces $\text{LLM}_{\theta^n} = \text{LLM}_{\theta^{n+1}}$.

Then we turn to (a). By the construction, for any $x \in T^*$ and $0 \le j \le n - 1$, $|w_i^j \text{rm}_i^j(x) - w_i^j \text{rm}_i^{j+1}(x)| \le \frac{\bar{w}}{n} \le \delta$, so that $|\text{LLM}_{\theta^j}(x) - \text{LLM}_{\theta^{j+1}}(x)| \le \frac{\epsilon}{4\bar{w}}$ holds for all $x$. Then we can derive that:

$$\sum_{j=0}^{n-1} \sum_{x \in T^*} w_i(\text{LLM}_{\theta^{j+1}}(x) - \text{LLM}_{\theta^j}(x))(\text{rm}_i^{j+1}(x) - \text{rm}_i^j(x))$$

$$= \sum_{j=0}^{n-1} \sum_{x \in T^*} w_i(\text{LLM}_{\theta^{j+1}}(x) - \text{LLM}_{\theta^j}(x))\frac{\text{rm}_i^*(x) - \text{rm}_i(x)}{n}$$

$$\le \sum_{j=0}^{n-1} \sum_{x \in T^*} w_i \frac{\epsilon}{4\bar{w}} \frac{|\text{rm}_i^*(x) - \text{rm}_i(x)|}{n}$$

$$\le \sum_{x \in T^*} \frac{\epsilon}{4}|\text{rm}_i^*(x) - \text{rm}_i(x)|$$

$$\le \sum_{x \in T^*} \frac{\epsilon}{4}(\text{rm}_i^*(x) + \text{rm}_i(x)) \le \frac{\epsilon}{2}.$$

The case is similar to (c). By the construction, for any $x \in T^*$ and $n + 1 \le j \le 2n$, $|w_i^j \text{rm}_i^j(x) - w_i^j \text{rm}_i^{j+1}(x)| \le \frac{\bar{w}}{n} \le \delta$, so that $|\text{LLM}_{\theta^j}(x) - \text{LLM}_{\theta^{j+1}}(x)| \le \frac{\epsilon}{4\bar{w}}$ holds for all $x$. Then we can

derive that:

$$\sum_{j=n+1}^{2n} \sum_{x \in T^*} w_i(\mathrm{LLM}_{\theta^{j+1}}(x) - \mathrm{LLM}_{\theta^j}(x))(\mathrm{rm}_i^{j+1}(x) - \mathrm{rm}_i^j(x))$$

$$= \sum_{j=n+1}^{2n} \sum_{x \in T^*} w_i(\mathrm{LLM}_{\theta^{j+1}}(x) - \mathrm{LLM}_{\theta^j}(x)) \frac{\mathrm{rm}_i'(x) - \mathrm{rm}_i^*(x)}{n}$$

$$\leq \sum_{j=n+1}^{2n} \sum_{x \in T^*} w_i \frac{\epsilon}{4\bar{w}} \frac{|\mathrm{rm}_i'(x) - \mathrm{rm}_i^*(x)|}{n}$$

$$\leq \sum_{x \in T^*} \frac{\epsilon}{4} |\mathrm{rm}_i'(x) - \mathrm{rm}_i^*(x)|$$

$$\leq \sum_{x \in T^*} \frac{\epsilon}{4} (\mathrm{rm}_i'(x) + \mathrm{rm}_i^*(x)) \leq \frac{\epsilon}{2}.$$

Combining the results from (a), (b), and (c), we have that under this construction,

$$\sum_{j=0}^{k} l(t_i^j, t_i^{j+1}) + \sum_{j=0}^{k} l(t_i^{j+1}, t_i^j) \leq \frac{\epsilon}{2} + \frac{\epsilon}{2} = \epsilon.$$

By the arbitrariness of $\epsilon > 0$, this is suffice to demonstrate that $V(t_i, t_i') + V(t_i, t_i') \leq 0$.

Therefore, it is proven that

$$V(t_i, t_i') + V(t_i, t_i') = 0.$$

which means that $V(t_i, t_i') = -V(t_i', t_i)$. By Lemma C.2, this is a sufficient condition for the payment equivalence of $\psi$. $\qquad \square$

**Corollary 4.10.** *Each continuous training rule $\psi \in \Psi^{SW}$ satisfies payment equivalence.*

*Proof.* We construct the reward model as follows and show that this satisfies the condition in Corollary 4.10 for when the mechanism uses SW-Max training rules.

$$\mathrm{rm}^*(x) = \begin{cases} \dfrac{1}{|T^*|} & \text{Summation Normalization Case,} \\ 1 & \text{Maximum Normalization Case.} \end{cases} \tag{5}$$

We prove this by contradiction. Assuming that there exist $i, \overrightarrow{\mathrm{rm}}_{-i}, \vec{w}_{-i}, \theta_{\mathrm{init}}, w_i, w_i'$ such that

$$\theta := \psi((\mathrm{rm}_i^*, \overrightarrow{\mathrm{rm}}_{-i}), (w_i, \vec{w}_{-i}), \theta_{\mathrm{init}}) \neq \psi((\mathrm{rm}_i^*, \overrightarrow{\mathrm{rm}}_{-i}), (w_i', \vec{w}_{-i}), \theta_{\mathrm{init}}) =: \theta'$$

We denote the further tie-breaking rule as $\succ_{\overrightarrow{\mathrm{rm}}}$. Then, considering the optimality of $\theta$, we have one of the following satisfied.

$$\mathrm{ASW}(\theta; (\mathrm{rm}_i^*, \overrightarrow{\mathrm{rm}}_{-i}), (w_i, \vec{w}_{-i}), \theta_{\mathrm{init}}) > \mathrm{ASW}(\theta'; (\mathrm{rm}_i^*, \overrightarrow{\mathrm{rm}}_{-i}), (w_i, \vec{w}_{-i}), \theta_{\mathrm{init}}),$$

or

$$\mathrm{ASW}(\theta; (\mathrm{rm}_i^*, \overrightarrow{\mathrm{rm}}_{-i}), (w_i, \vec{w}_{-i}), \theta_{\mathrm{init}}) = \mathrm{ASW}(\theta'; (\mathrm{rm}_i^*, \overrightarrow{\mathrm{rm}}_{-i}), (w_i, \vec{w}_{-i}), \theta_{\mathrm{init}}), \text{ and } \mathrm{LLM}_\theta \succ_{\overrightarrow{\mathrm{rm}}} \mathrm{LLM}_{\theta'}.$$

Note that $v_i(\theta; \mathrm{rm}_i^*) = v_i(\theta'; \mathrm{rm}_i^*)$, and $\mathrm{ASW}(\theta; (\mathrm{rm}_i^*, \overrightarrow{\mathrm{rm}}_{-i}), (w_i, \vec{w}_{-i}), \theta_{\mathrm{init}}) = (w_i' - w_i)v_i(\theta; \mathrm{rm}_i^*)$ $+ \mathrm{ASW}(\theta; (\mathrm{rm}_i^*, \overrightarrow{\mathrm{rm}}_{-i}), (w_i', \vec{w}_{-i}), \theta_{\mathrm{init}})$, we have

$$\mathrm{ASW}(\theta; (\mathrm{rm}_i^*, \overrightarrow{\mathrm{rm}}_{-i}), (w_i', \vec{w}_{-i}), \theta_{\mathrm{init}}) > \mathrm{ASW}(\theta'; (\mathrm{rm}_i^*, \overrightarrow{\mathrm{rm}}_{-i}), (w_i', \vec{w}_{-i}), \theta_{\mathrm{init}})$$

or

$$\mathrm{ASW}(\theta; (\mathrm{rm}_i^*, \overrightarrow{\mathrm{rm}}_{-i}), (w_i', \vec{w}_{-i}), \theta_{\mathrm{init}}) = \mathrm{ASW}(\theta'; (\mathrm{rm}_i^*, \overrightarrow{\mathrm{rm}}_{-i}), (w_i', \vec{w}_{-i}), \theta_{\mathrm{init}}), \text{ and } \mathrm{LLM}_\theta \succ_{\overrightarrow{\mathrm{rm}}} \mathrm{LLM}_{\theta'}.$$

Both cases contradicted the optimality of $\theta'$. $\qquad \square$

**Theorem 4.11.** *Given a continuous training rule $\psi \in \Psi^{SW}$ and a payment rule $p$ implements it in DSIC: If $p$ is always non-negative, it holds that for all $i$, $\overrightarrow{rm}$, $\vec{w}$, and $\theta_{init}$,*

$$p_i(\overrightarrow{rm}, \vec{w}, \theta_{init}) \geq p_i^{AFF}(\overrightarrow{rm}, \vec{w}, \theta_{init}).$$

*If $p$ implements $\psi$ in IR, then for any $\epsilon > 0$ and $i$, there exists $\overrightarrow{rm}_{-i}$, $\vec{w}_{-i}$, and $\theta_{init}$, such that for all $rm_i$ and $w_i$,*

$$p_i(\overrightarrow{rm}, \vec{w}, \theta_{init}) \leq p_i^{AFF}(\overrightarrow{rm}, \vec{w}, \theta_{init}) + \epsilon.$$

*Proof.* For a continuous SW-Max training rule $\psi$, we know that it satisfies payment equivalence. By the definition of payment equivalence, for any other payment rule $p$ that also implements $\psi$ in DSIC, there exists a function $g_i$ such that

$$p_i(\overrightarrow{rm}, \vec{w}, \theta_{\text{init}}) = p_i^{AFF}(\overrightarrow{rm}, \vec{w}, \theta_{\text{init}}) + g_i(\overrightarrow{rm}_{-i}, \vec{w}_{-i}, \theta_{\text{init}}).$$

**Non-negative Payment.** To ensure that $p_i(\overrightarrow{rm}, \vec{w}, \theta_{\text{init}}) \geq 0$ always satisfied, we have the equivalent condition:

$$g_i(\overrightarrow{rm}_{-i}, \vec{w}_{-i}, \theta_{\text{init}}) \geq - \inf_{rm_i, w_i} p_i^{AFF}(\overrightarrow{rm}, \vec{w}, \theta_{\text{init}}).$$

However, for any $\overrightarrow{rm}_{-i}, \vec{w}_{-i}, \theta_{\text{init}}$, when we set $rm_i$ to the uniform reward model Equation (5), we have shown in the previous proof that this will not change the training outcome regardless of the value of $w_i$ and hence does not impact the $\text{ASW}_{-i}$. This means that the payment defined by the affine maximizer is exactly 0, and the RHS of the above equation will always be non-negative. Therefore, there must be $g_i \geq 0$ for all inputs, which means that for all $i$, $\overrightarrow{rm}$, $\vec{w}$, and $\theta_{\text{init}}$, we have $p_i(\overrightarrow{rm}, \vec{w}, \theta_{\text{init}}) \geq p_i^{AFF}(\overrightarrow{rm}, \vec{w}, \theta_{\text{init}})$.

**Individually Rationality.** To ensure the utility of any group is not negative, we have to constrain the function $g_i$ as follows:

$$g_i(\overrightarrow{rm}_{-i}, \vec{w}_{-i}, \theta_{\text{init}}) \leq \inf_{rm_i, w_i} u_i^{AFF}(\overrightarrow{rm}, \vec{w}, \theta_{\text{init}}),$$

where we denote $u_i^{AFF}$ the utility of group $i$ under the mechanism. We construct an extreme case such that the RHS can be sufficiently small. Without loss of generality, we assume that $T^* = \{\boldsymbol{x}_1, \boldsymbol{x}_2\}$. The initial model $\text{LLM}_{\theta_{\text{init}}}(\boldsymbol{x}_1) = \epsilon$, $\text{LLM}_{\theta_{\text{init}}}(\boldsymbol{x}_2) = 1 - \epsilon$. Group $i$ has preference $rm_i(\boldsymbol{x}_1) = 1$ and $rm_i(\boldsymbol{x}_2) = 0$, and other groups have opposite preference: $rm_j(\boldsymbol{x}_1) = 0$ and $rm_j(\boldsymbol{x}_2) = 1$ for $j \neq i$. The group size is set to $w_k = 1$ for all $k \in [n]$.

In this case, as we have $\sum_{k=1}^n w_k rm_k(\boldsymbol{x}_1) < \sum_{k=1}^n w_k rm_k(\boldsymbol{x}_2)$, we can directly derived from the optimal condition Equation (OPT) that the final model satisfies that $\text{LLM}_\theta(\boldsymbol{x}_1) \leq \text{LLM}_{\theta_{\text{init}}}(\boldsymbol{x}_1)$. Since $p^{AFF}$ is always non-negative, the utility of group $i$ is at most $rm_i(\boldsymbol{x}_1) \cdot \text{LLM}_{\theta_{\text{init}}}(\boldsymbol{x}_1) = \epsilon$. To ensure that $p$ implements $\psi$ in IR, we have to set $g_i(\overrightarrow{rm}_{-i}, \vec{w}_{-i}, \theta_{\text{init}}) \leq \epsilon$ for this case. This is equivalent to $p_i(\overrightarrow{rm}, \vec{w}, \theta_{\text{init}}) \leq p_i^{AFF}(\overrightarrow{rm}, \vec{w}, \theta_{\text{init}})$. $\square$

## D  Omitted Proofs in Section 4.3

**Lemma D.1.** *For any $rm, rm'$, if $\max_{\boldsymbol{x} \in T^*} |rm(\boldsymbol{x}) - rm'(\boldsymbol{x})| = \epsilon$, then for any model $\theta$, we have*

$$|v(\theta; rm) - v(\theta; rm')| \leq \epsilon$$

*Proof.* We can derive that

$$|v(\theta; rm) - v(\theta; rm')| = |\sum_{\boldsymbol{x} \in T^*} \text{LLM}_\theta(\boldsymbol{x})(rm(\boldsymbol{x}) - rm'(\boldsymbol{x}))| \leq \sum_{\boldsymbol{x} \in T^*} \text{LLM}_\theta(\boldsymbol{x})|rm(\boldsymbol{x}) - rm'(\boldsymbol{x})|$$

$$\leq \sum_{\boldsymbol{x} \in T^*} \text{LLM}_\theta(\boldsymbol{x})\epsilon = \epsilon.$$

$\square$

**Lemma D.2.** *Assume that for any noisy input $\overrightarrow{\widehat{rm}}$ generated from $F(\cdot|\overrightarrow{rm})$, and $i \in [n]$, there is*

$$\max_{\boldsymbol{x} \in T^*} |\widehat{rm}_i(\boldsymbol{x}) - rm_i(\boldsymbol{x})| \leq \epsilon.$$

*Then for any $\psi \in \Psi^{SW}$ and $\overrightarrow{\widehat{rm}}$ generated from $F(\cdot|\overrightarrow{rm})$, the distance between the training outcome and the optimal is bounded by:*

$$\text{ASW}(\psi(\overrightarrow{\widehat{rm}}, \vec{w}, \theta_{init}); \overrightarrow{rm}, \vec{w}, \theta_{init}) \geq$$

$$\text{ASW}(\psi(\overrightarrow{rm}, \vec{w}, \theta_{init}); \overrightarrow{rm}, \vec{w}, \theta_{init}) - 2\epsilon \sum_{i=1}^{n} w_i.$$

*Proof.* Let $\hat{\theta} = \psi(\overrightarrow{\widehat{rm}}, \vec{w}, \theta_{init})$ and $\theta = \psi(\overrightarrow{rm}, \vec{w}, \theta_{init})$. $\hat{\theta}$ is the optimal parameter for biased input, and $\theta$ is the optimal parameter for the true input.

$$\text{ASW}(\hat{\theta}; \overrightarrow{rm}, \vec{w}, \theta_{init}) = \sum_{i=1}^{n} w_i v_i(\hat{\theta}; rm_i) - D_f(\text{LLM}_{\hat{\theta}} || \text{LLM}_{\theta_{init}})$$

$$\overset{(1)}{\geq} \sum_{i=1}^{n} w_i \left( v_i(\hat{\theta}; \widehat{rm}_i) - \epsilon \right) - D_f(\text{LLM}_{\hat{\theta}} || \text{LLM}_{\theta_{init}})$$

$$= \text{ASW}(\hat{\theta}; \overrightarrow{\widehat{rm}}, \vec{w}, \theta_{init}) - \sum_{i=1}^{n} w_i \epsilon$$

$$\overset{(2)}{\geq} \text{ASW}(\theta; \overrightarrow{\widehat{rm}}, \vec{w}, \theta_{init}) - \sum_{i=1}^{n} w_i \epsilon$$

$$= \sum_{i=1}^{n} w_i v_i(\theta; \widehat{rm}_i) - D_f(\text{LLM}_{\theta} || \text{LLM}_{\theta_{init}}) - \sum_{i=1}^{n} w_i \epsilon$$

$$\overset{(3)}{\geq} \sum_{i=1}^{n} w_i (v_i(\theta; rm_i) - \epsilon) - D_f(\text{LLM}_{\theta} || \text{LLM}_{\theta_{init}}) - \sum_{i=1}^{n} w_i \epsilon$$

$$= \text{ASW}(\theta; \overrightarrow{rm}, \vec{w}, \theta_{init}) - 2 \sum_{i=1}^{n} w_i \epsilon.$$

(1) and (3) can be directly induced by Lemma D.1, and (2) holds by the definition of $\hat{\theta}$.

$$\hat{\theta} = \psi(\overrightarrow{\widehat{rm}}, \vec{w}, \theta_{init}) = \arg\max_{\theta \in \Theta} \text{ASW}(\theta; \overrightarrow{\widehat{rm}}, \vec{w}, \theta_{init}).$$

$\square$

**Theorem 4.12.** *Assume that for any noisy input $\overrightarrow{\widehat{rm}}$ generated from $F(\cdot|\overrightarrow{rm})$, and $i \in [n]$, there is*

$$\max_{\boldsymbol{x} \in T^*} |\widehat{rm}_i(\boldsymbol{x}) - rm_i(\boldsymbol{x})| \leq \epsilon.$$

*Then, with a training rule $\psi \in \Psi^{SW}$, $(\psi, p^{AFF})$ ensures that each group $i$ can benefit at most $2w_i\epsilon$ from misreporting the reward model.*

*Proof.* Recall that the calculation of payment in $p^{AFF}$ is

$$p_i^{AFF}(\overrightarrow{rm}, \vec{w}, \theta_{init}) = \text{ASW}_{-i}(\psi(\overrightarrow{rm}_{-i}, \vec{w}_{-i}, \theta_{init}); \overrightarrow{rm}, \vec{w}, \theta_{init}) - \text{ASW}_{-i}(\psi(\overrightarrow{rm}, \vec{w}, \theta_{init}); \overrightarrow{rm}, \vec{w}, \theta_{init}).$$

Let $\vec{w} = (w_i, \vec{w}_{-i})$, the utility function can be written as:

$$u_i((rm'_i, \overrightarrow{rm}_{-i}), \vec{w}; \psi, p, rm_i, w_i) = w_i v_i(\theta; rm_i) - p_i^{AFF}((rm'_i, \overrightarrow{rm}_{-i}), \vec{w}, \theta_{init})$$

$$= w_i v_i(\theta; rm_i) - \text{ASW}_{-i}(\theta_{-i}; \overrightarrow{rm}, \vec{w}, \theta_{init}) + \text{ASW}_{-i}(\theta; \overrightarrow{rm}, \vec{w}, \theta_{init})$$

$$= \text{ASW}(\theta; \overrightarrow{rm}, \vec{w}, \theta_{init}) - \text{ASW}_{-i}(\theta_{-i}; \overrightarrow{rm}, \vec{w}, \theta_{init}),$$

where we define $\theta = \psi((rm'_i, \overrightarrow{rm}_{-i}), \vec{w}, \theta_{init})$, and $\theta_{-i} = \psi(\overrightarrow{rm}_{-i}, \vec{w}_{-i}, \theta_{init})$. Note that the term $\text{ASW}_{-i}(\theta_{-i}; \overrightarrow{rm}, \vec{w}, \theta_{init})$ is not influenced by the change of $rm_i$ or $w_i$.

Therefore, we can derive that for any $\overrightarrow{\mathrm{rm}}_{-i}, \vec{w}$, let $\theta_{-i} = \psi(\overrightarrow{\mathrm{rm}}_{-i}, \vec{w}_{-i}, \theta_{\mathrm{init}})$:

$$\mathbb{E}_{\widehat{\mathrm{rm}}_i \sim \mathcal{F}_i(\cdot|\mathrm{rm}_i)} \left[ u_i((\widehat{\mathrm{rm}}_i, \overrightarrow{\mathrm{rm}}_{-i}), \vec{w}; \psi, p, \mathrm{rm}_i, w_i) + \mathrm{ASW}_{-i}(\theta_{-i}; \overrightarrow{\mathrm{rm}}, \vec{w}, \theta_{\mathrm{init}}) \right]$$

$$= \mathbb{E}_{\widehat{\mathrm{rm}}_i \sim \mathcal{F}_i(\cdot|\mathrm{rm}_i)} \left[ \mathrm{ASW}(\hat{\theta}; \overrightarrow{\mathrm{rm}}, \vec{w}, \theta_{\mathrm{init}}) \right]$$

$$= \mathbb{E}_{\widehat{\mathrm{rm}}_i \sim \mathcal{F}_i(\cdot|\mathrm{rm}_i)} \left[ w_i v_i(\hat{\theta}; \mathrm{rm}_i) + \sum_{j \neq i} w_j v_j(\hat{\theta}; \mathrm{rm}_j) - D_f(\mathrm{LLM}_{\hat{\theta}} \| \mathrm{LLM}_{\theta_{\mathrm{init}}}) \right]$$

$$\overset{(1)}{\geq} \mathbb{E}_{\widehat{\mathrm{rm}}_i \sim \mathcal{F}_i(\cdot|\mathrm{rm}_i)} \left[ w_i v_i(\hat{\theta}; \widehat{\mathrm{rm}}_i) + \sum_{j \neq i} w_j v_j(\hat{\theta}; \mathrm{rm}_j) - D_f(\mathrm{LLM}_{\hat{\theta}} \| \mathrm{LLM}_{\theta_{\mathrm{init}}}) \right] - w_i \epsilon$$

$$\overset{(2)}{\geq} \mathbb{E}_{\widehat{\mathrm{rm}}_i \sim \mathcal{F}_i(\cdot|\mathrm{rm}_i)} \left[ w_i v_i(\theta; \widehat{\mathrm{rm}}_i) + \sum_{j \neq i} w_j v_j(\theta; \mathrm{rm}_j) - D_f(\mathrm{LLM}_{\theta} \| \mathrm{LLM}_{\theta_{\mathrm{init}}}) \right] - w_i \epsilon$$

$$\overset{(3)}{\geq} \mathbb{E}_{\widehat{\mathrm{rm}}_i \sim \mathcal{F}_i(\cdot|\mathrm{rm}_i)} \left[ w_i v_i(\theta; \mathrm{rm}_i) + \sum_{j \neq i} w_j v_j(\theta; \mathrm{rm}_j) - D_f(\mathrm{LLM}_{\theta} \| \mathrm{LLM}_{\theta_{\mathrm{init}}}) \right] - 2w_i \epsilon$$

$$\overset{(4)}{=} \mathbb{E}_{\widehat{\mathrm{rm}}_i \sim \mathcal{F}_i(\cdot|\mathrm{rm}_i')} \left[ w_i v_i(\theta; \mathrm{rm}_i) + \sum_{j \neq i} w_j v_j(\theta; \mathrm{rm}_j) - D_f(\mathrm{LLM}_{\theta} \| \mathrm{LLM}_{\theta_{\mathrm{init}}}) \right] - 2w_i \epsilon$$

$$\overset{(5)}{\geq} \mathbb{E}_{\widehat{\mathrm{rm}}_i \sim \mathcal{F}_i(\cdot|\mathrm{rm}_i')} \left[ w_i v_i(\hat{\theta}; \mathrm{rm}_i) + \sum_{j \neq i} w_j v_j(\hat{\theta}; \mathrm{rm}_j) - D_f(\mathrm{LLM}_{\hat{\theta}} \| \mathrm{LLM}_{\theta_{\mathrm{init}}}) \right] - 2w_i \epsilon$$

$$= \mathbb{E}_{\widehat{\mathrm{rm}}_i \sim \mathcal{F}_i(\cdot|\mathrm{rm}_i')} \left[ \mathrm{ASW}(\hat{\theta}; \overrightarrow{\mathrm{rm}}, \vec{w}, \theta_{\mathrm{init}}) \right] - 2w_i \epsilon$$

$$= \mathbb{E}_{\widehat{\mathrm{rm}}_i \sim \mathcal{F}_i(\cdot|\mathrm{rm}_i')} \left[ u_i((\widehat{\mathrm{rm}}_i, \overrightarrow{\mathrm{rm}}_{-i}), \vec{w}; \psi, p, \mathrm{rm}_i, w_i) + \mathrm{ASW}_{-i}(\theta_{-i}; \overrightarrow{\mathrm{rm}}, \vec{w}, \theta_{\mathrm{init}}) \right] - 2w_i \epsilon$$

All the $\hat{\theta}$ in the above inequalities refers to the optimal parameter for input $(\widehat{\mathrm{rm}}_i, \overrightarrow{\mathrm{rm}}_{-i}), \vec{w}, \theta_{\mathrm{init}}$, i.e. $\hat{\theta} = \psi((\widehat{\mathrm{rm}}_i, \overrightarrow{\mathrm{rm}}_{-i}), \vec{w}, \theta_{\mathrm{init}})$. Specifically, (1) and (3) come from the bounded distance between $\mathrm{rm}_i$ and $\widehat{\mathrm{rm}}_i$ (Lemma D.1), (2) and (5) hold by the definitions: $\hat{\theta} = \psi((\widehat{\mathrm{rm}}_i, \overrightarrow{\mathrm{rm}}_{-i}), \vec{w}, \theta_{\mathrm{init}})$ $= \arg\max_{\theta' \in \Theta} \mathrm{ASW}(\theta'; (\widehat{\mathrm{rm}}_i, \overrightarrow{\mathrm{rm}}_{-i}), \vec{w}, \theta_{\mathrm{init}})$ and $\theta = \psi((\mathrm{rm}_i, \overrightarrow{\mathrm{rm}}_{-i}), \vec{w}, \theta_{\mathrm{init}}) = \arg\max_{\theta' \in \Theta}$ $\mathrm{ASW}(\theta'; (\mathrm{rm}_i, \overrightarrow{\mathrm{rm}}_{-i}), \vec{w}, \theta_{\mathrm{init}})$. And (4) holds since the inner term is irrelevant to $\widehat{\mathrm{rm}}_i$.

Therefore, we get

$$U_i((\mathrm{rm}_i, \overrightarrow{\mathrm{rm}}_{-i}), \vec{w}; \psi, p, \mathrm{rm}_i, w_i)$$

$$= \mathbb{E}_{\widehat{\overrightarrow{\mathrm{rm}}} \sim \mathcal{F}(\cdot|(\mathrm{rm}_i, \overrightarrow{\mathrm{rm}}_{-i}))} \left[ u_i(\widehat{\overrightarrow{\mathrm{rm}}}, \vec{w}; \psi, p, \mathrm{rm}_i, w_i) \right]$$

$$= \mathbb{E}_{\widehat{\mathrm{rm}}_i \sim \mathcal{F}_i(\cdot|\mathrm{rm}_i)} \mathbb{E}_{\widehat{\overrightarrow{\mathrm{rm}}}_{-i} \sim \mathcal{F}_{-i}(\cdot|\overrightarrow{\mathrm{rm}}_{-i})} \left[ u_i((\widehat{\mathrm{rm}}_i, \widehat{\overrightarrow{\mathrm{rm}}}_{-i}), \vec{w}; \psi, p, \mathrm{rm}_i, w_i) \right]$$

$$\geq \mathbb{E}_{\widehat{\mathrm{rm}}_i \sim \mathcal{F}_i(\cdot|\mathrm{rm}_i')} \mathbb{E}_{\widehat{\overrightarrow{\mathrm{rm}}}_{-i} \sim \mathcal{F}_{-i}(\cdot|\overrightarrow{\mathrm{rm}}_{-i})} \left[ u_i((\widehat{\mathrm{rm}}_i, \widehat{\overrightarrow{\mathrm{rm}}}_{-i}), \vec{w}; \psi, p, \mathrm{rm}_i, w_i) - 2w_i \epsilon \right]$$

$$= \mathbb{E}_{\widehat{\overrightarrow{\mathrm{rm}}} \sim \mathcal{F}(\cdot|(\mathrm{rm}_i', \overrightarrow{\mathrm{rm}}_{-i}))} \left[ u_i(\widehat{\overrightarrow{\mathrm{rm}}}, \vec{w}; \psi, p, \mathrm{rm}_i, w_i) - 2w_i \epsilon \right]$$

$$= U_i((\mathrm{rm}_i', \overrightarrow{\mathrm{rm}}_{-i}), \vec{w}; \psi, p, \mathrm{rm}_i, w_i) - 2w_i \epsilon..$$

$\square$

# E  Further Discussion on General Training Rules

In practice, some other training principles do not belong to SW-Max training rules, including those that maximize the Nash Social Welfare and focus more on fairness issues, like MaxMin-RLHF [11]. As an initial study on the incentive property of the RLHF Game, we primarily consider the mainstream training rules, SW-Max training rules, that aim to maximize social welfare under certain regularization.

Therefore, analyzing the properties of general forms of training rules is out of the scope of this paper. However, we also make a preliminary step for analyzing the two questions proposed in Section 4.2. The second question is partly included in Theorem 4.9, and for the implementability of a training rule, we utilize the notion of *cycle monotonicity* proposed by Rochet [65], which is a generalized version of monotonicity defined in a single-parameter scenario [53]. In the RLHF Game, we use the notation $t_i$ to represent the combination of $(\text{rm}_i, w_i)$ with the same superscript and subscript. We define the function $l(t_i', t_i; \vec{t}_{-i}, \theta_{\text{init}}) := w_i v_i(\psi((t_i, \vec{t}_{-i}), \theta_{\text{init}}); \text{rm}_i) - w_i v_i(\psi((t_i', \vec{t}_{-i}, \theta_{\text{init}})); \text{rm}_i)$ to measure group $i$'s valuation gains from misreporting ($t_i'$) to truthfully reporting ($t_i$) under $\vec{t}_{-i}$ and $\theta_{\text{init}}$. The cycle monotonicity is defined based on this function:

**Definition E.1** (Cycle Monotonicity). The training rule $\psi$ satisfies cycle monotonicity if for any group $i$, $t_i, t_i' \in \mathcal{R} \times \mathcal{W}$, any finite, distinct sequence of reward models $[t_i, t_i^1, t_i^2, \ldots, t_i^k, t_i']$ ($k \geq 0$), and any $\vec{t}_{-i}, \theta_{\text{init}}$, defining $t_i^0 = t_i^{k+2} := t_i$ and $t_i^{k+1} := t_i'$, we have

$$\sum_{j=0}^{k+1} l(t_i^j, t_i^{j+1}; \vec{t}_{-i}, \theta_{\text{init}}) \geq 0.$$

For general training rules, cycle monotonicity is a sufficient and necessary condition for implementability.

**Proposition E.2** (Rochet [65]). *A training rule $\psi$ is implementable if and only if it satisfies cycle monotonicity.*

*Proof.* We fix the other groups' report $\overrightarrow{\text{rm}}_{-i}, \vec{w}_{-i}, \theta_{\text{init}}$, and also omit their notations for simplicity.

**We first prove the necessity:** if $\psi$ is implementable, it satisfies cycle monotonicity. Since $\psi$ is implementable, there exists $p$ such that $(\psi, p)$ satisfies DSIC. We use notation $t_i^j$ to represent the combination of $(\text{rm}_i^j, w_i^j)$. For any types $t_i, t_i' \in \mathcal{R} \times \mathcal{W}$, any finite and distinct sequence of types $[t_i, t_i^1, t_i^2, \ldots, t_i^k, t_i']$, $k \geq 0$, we let $t_i^0 = t_i^{k+2} := t_i$ and $t_i^{k+1} := t_i'$. By the property of DSIC, we have

$$w_i^{j+1} v_i(\psi(t_i^{j+1}); \text{rm}_i^{j+1}) - p_i(t_i^{j+1}) \geq w_i^{j+1} v_i(\psi(t_i^j); \text{rm}_i^{j+1}) - p_i(t_i^j) \quad \forall 0 \leq j \leq k+1.$$

By definition of the function $l$, this is equivalent to

$$l(t_i^j, t_i^{j+1}) \geq p_i(t_i^{j+1}) - p_i(t_i^j) \quad \forall 0 \leq j \leq k+1.$$

Sum over all $j$, we get

$$\sum_{j=0}^{k+1} l(t_i^j, t_i^{j+1}) \geq \sum_{j=0}^{k+1} \left( p_i(t_i^{j+1}) - p_i(t_i^j) \right) = 0.$$

By the arbitrariness of the sequence $[t_i, t_i^1, t_i^2, \ldots, t_i^k, t_i']$, this means that $\psi$ satisfies cycle monotonicity.

**Then, we prove the sufficiency:** By cycle monotonicity, we have that for any finite and distinct sequence $[t_i, t_i^1, t_i^2, \ldots, t_i^k, t_i']$,

$$\sum_{j=0}^{k} l(t_i^j, t_i^{j+1}) + l(t_i', t_i) = \sum_{j=0}^{k+1} l(t_i^j, t_i^{j+1}) \geq 0.$$

By the arbitrariness of the sequence, we can infer that

$$V(t_i, t_i') + l(t_i', t_i) \geq 0.$$

Since $l(t_i', t_i)$ is bounded, $V(t_i, t_i')$ is also finite and $V(t_i, t_i') \geq -l(t_i', t_i)$. Then, we can establish a payment rule $p$ such that for any agent $i$,

$$p_i(t_i) = V(t^*, t_i).$$

where $t^* \in \mathcal{R} \times \mathcal{W}$ is a certain type.

Then, for any $t_i = (\mathrm{rm}_i, w_i)$, we have

$$
\begin{aligned}
& w_i v_i(\psi(t_i); \mathrm{rm}_i) - p_i(t_i) \\
=& w_i v_i(\psi(t_i); \mathrm{rm}_i) - V(t^*, t_i) \\
=& w_i v_i(\psi(t_i'); \mathrm{rm}_i) + l(t_i', t_i) - V(t^*, t_i) \\
\overset{(2)}{\geq}& w_i v_i(\psi(t_i'), \mathrm{rm}_i) - V(t^*, t_i') \\
=& w_i v_i(\psi(t_i'); \mathrm{rm}_i) - p_i(t_i').
\end{aligned}
$$

Note that (2) comes from the definition of $V$ that:

$$
\begin{aligned}
V(t^*, t_i) =& \inf_{\substack{\text{A finite and distinct sequence} \\ [t_i^0 := t^*, t_i^1, \dots, t_i^k, t_i^{k+1} := t_i]}} \sum_{j=0}^{k} l(t_i^j, t_i^{j+1}) \\
\leq& \inf_{\substack{\text{A finite and distinct sequence} \\ [t_i^0 := t^*, t_i^1, \dots, t_i^k, t_i^{k+1} := t_i']}} \sum_{j=0}^{k} l(t_i^j, t_i^{j+1}) + l(t_i', t_i) \\
=& V(t^*, t_i') + l(t_i', t_i).
\end{aligned}
$$

This means that mechanism $(\psi, p)$ satisfies DSIC, and suffices to show that $\psi$ is implementable. $\quad\square$

Validating whether a training rule satisfies cycle monotonicity is a complex task. Thus, finding a more concise condition that can induce the implementability for a general training rule or a subset of training rules is a valuable further direction.

# F  Additional Experimental Results

**Synthetic RLHF Game.**  We construct a synthetic RLHF Game: We set the group number to be 5 and assume the size of the outcome space to be 10. Each group's preference is first sampled from a uniform distribution $U[0, 1]^{10}$ and then normalized. The group sizes are uniformly sampled from $\{1, 2, \dots, 10\}^{10}$.

We consider the misreporting strategy that is used to prove Theorem 4.2. Specifically, given a group's preference rm. We first find the most preferred and the least preferred outcome $x_1 = \arg\max_x \mathrm{rm}(x)$, $x_2 = \arg\min_x \mathrm{rm}(x)$. Then we set the reported reward model to be $\widetilde{\mathrm{rm}}(x_1) = \mathrm{rm}(x_1) + \epsilon$, $\widetilde{\mathrm{rm}}(x_2) = \mathrm{rm}(x_2) - \epsilon$, and $\widetilde{\mathrm{rm}}(x) = \mathrm{rm}(x)$ for other $x$s.

Table 1: Average changes in valuation and utility when adopting the misreporting strategy from Theorem 4.2, holding other groups' reports fixed. The parameter $\epsilon$ controls the extent of deviation from truthful reporting. As shown in the table, such a misreporting strategy brings valuation gain but decreases the utility.

| Reporting Parameter $\epsilon$ | Type | 0.001 | 0.002 | 0.005 | 0.01 | 0.02 | 0.05 | 0.1 |
|---|---|---|---|---|---|---|---|---|
| $\Delta$Valuation (*1e2) | Mean | 0.1073 | 0.2096 | 0.4881 | 0.8667 | 1.3674 | 1.7978 | 1.8154 |
| | Std | < 0.0001 | < 0.0001 | 0.0003 | 0.0004 | 0.0013 | 0.0026 | 0.0032 |
| $\Delta$Utility (*1e4) | Mean | -0.1064 | -0.4135 | -2.3696 | -8.1557 | -23.7334 | -53.1552 | -55.8977 |
| | Std | < 0.0001 | 0.0001 | 0.0011 | 0.0046 | 0.0196 | 0.0573 | 0.1415 |

We let group 1 use this strategy and maintain the other group truthfully reporting. The payment is set according to the mechanism introduced in Section 4.2. We take $100,000$ samples and the average change in valuation and utility for group 1 is reported in Table 1. The result shows that such a strategy can indeed improve the valuation and is, hence, beneficial when there is no payment. However, with the introduced payment, no strategy will bring higher utility than truthfully reporting.

**More Complex Preferences.**  The experiment setup of this part follows the Section 5. We consider two scenarios with more complex, multiple preferences.

1. We simulated scenarios from data reported in [70] (Table 6), involving three groups, each valuing helpfulness, harmlessness, and humor, respectively. Normalization and other settings follow our paper. The true group sizes and the numerical results are shown in the tables below.

2. We examined a scenario where the group's preference is a linear combination of two reward models. Specifically, group 1 values $0.2 \times$ Helpfulness $+ 0.8 \times$ Harmlessness, group 2 values $0.8 \times$ Helpfulness $+ 0.2 \times$ Harmlessness, and group 3 values Humor.

All of the above results show that truthfully reporting is among the optimal strategies under the mechanism.

Table 2: Valuation, utility, and social welfare outcomes when varying reporting parameters for Group 1, with other groups' reports held fixed ($\alpha = 1$ means truthful reporting). Group sizes are set as $(w_1, w_2, w_3) = (3, 2, 1)$. The three groups value Helpfulness, Harmlessness, and Humor, respectively. The highest value in each row is highlighted in **bold**.

| Reporting Parameter $\alpha$ | 0.2 | 0.5 | 1 | 1.5 | 2 | 3 |
|---|---|---|---|---|---|---|
| Valuation | 0.00 | 0.79 | 2.66 | **3.00** | **3.00** | **3.00** |
| Utility (= Valuation-Payment) | 0.00 | 0.44 | **0.57** | 0.50 | 0.50 | 0.50 |
| Social Welfare | 2.51 | 2.94 | **3.08** | 3.00 | 3.00 | 3.00 |

Table 3: Valuation, utility, and social welfare outcomes when varying reporting parameters for Group 1, with other groups' reports held fixed ($\alpha = 1$ means truthful reporting). Group sizes are set as $(w_1, w_2, w_3) = (4, 5, 3)$. The three groups value Helpfulness, Harmlessness, and Humor, respectively. The highest value in each row is highlighted in **bold**.

| Reporting Parameter $\alpha$ | 0.2 | 0.5 | 1 | 1.5 | 2 | 3 |
|---|---|---|---|---|---|---|
| Valuation | 0.00 | 0.00 | 1.05 | 1.05 | 3.54 | **4.00** |
| Utility (= Valuation-Payment) | 0.00 | 0.00 | **0.43** | **0.43** | -1.83 | -2.51 |
| Social Welfare | 6.51 | 6.51 | **6.94** | **6.94** | 4.68 | 4.00 |

Table 4: Valuation, utility, and social welfare outcomes when varying reporting parameters for Group 1, with other groups' reports held fixed ($\beta = 1$ means truthful reporting). Group sizes are set as $(w_1, w_2, w_3) = (5, 5, 2)$. The three groups value Helpfulness, Harmlessness, and Humor, respectively. The highest value in each row is highlighted in **bold**.

| Reporting Parameter $\beta$ | 0.5 | 0.8 | 1 | 1.5 | 2 | 3 |
|---|---|---|---|---|---|---|
| Valuation | 0.00 | 0.33 | 1.31 | 4.67 | **5.00** | **5.00** |
| Utility (= Valuation-Payment) | 0.00 | 0.09 | **0.20** | -0.72 | -1.01 | -1.01 |
| Social Welfare | 6.01 | 6.10 | **6.20** | 5.29 | 5.00 | 5.00 |

Table 5: Valuation, utility, and social welfare outcomes when varying reporting parameters for Group 1, with other groups' reports held fixed ($\beta = 1$ means truthful reporting). Group sizes are set as $(w_1, w_2, w_3) = (3, 1, 4)$. The three groups value Helpfulness, Harmlessness, and Humor, respectively. The highest value in each row is highlighted in **bold**.

| Reporting Parameter $\beta$ | 0.5 | 0.8 | 1 | 1.5 | 2 | 3 |
|---|---|---|---|---|---|---|
| Valuation | 0.79 | 0.79 | 0.79 | 0.79 | 2.66 | **3.00** |
| Utility (= Valuation-Payment) | **0.79** | **0.79** | **0.79** | **0.79** | -1.04 | -1.58 |
| Social Welfare | **5.37** | **5.37** | **5.37** | **5.37** | 3.54 | 3.00 |

Table 6: Valuation, utility, and social welfare outcomes when varying reporting parameters for Group 1, with other groups' reports held fixed ($\alpha = 1$ means truthful reporting). Group sizes are set as $(w_1, w_2, w_3) = (2, 3, 1)$. The three groups value $0.8 \times$ Helpfulness $+ 0.2 \times$ Harmlessness, $0.2 \times$ Helpfulness $+ 0.8 \times$ Harmlessness, and Humor, respectively. The highest value in each row is highlighted in **bold**.

| Reporting Parameter $\alpha$ | 0.2 | 0.5 | 1 | 1.5 | 2 | 3 |
|---|---|---|---|---|---|---|
| Valuation | 0.53 | 0.53 | 1.03 | 1.51 | **1.60** | **1.60** |
| Utility (= Valuation-Payment) | 0.52 | 0.52 | **0.61** | 0.39 | 0.31 | 0.31 |
| Social Welfare | 2.92 | 2.92 | **3.01** | 2.79 | 2.71 | 2.71 |

