# OpenReview forum: "Mechanism Design for LLM Fine-tuning with Multiple Reward Models"
_NeurIPS.cc/2025/Conference — NeurIPS 2025 poster_

### Official Review · Reviewer_a1Sa · 2025-06-23

**Clarity:** 3
**Significance:** 3
**Originality:** 4
**Rating:** 4
**Confidence:** 3

**Summary:**

This paper proposes integrating payment schemes into SW-Max rules to address incentive issues where agents might strategically misreport their preferences to influence outcomes during LLM fine-tuning via RLHF across diverse groups. The authors provide theoretical analyses and proofs demonstrating the necessity, uniqueness, and robustness of the proposed payment rule. Additionally, empirical experiments conducted on two datasets support the claims and validate the effectiveness of the proposed method.

**Questions:**

No futher questions.

**Ethical Concerns:**

["NO or VERY MINOR ethics concerns only"]

**Final Justification:**

The reviewer's rebuttal by incorporating these experimental details and illustrating key concepts earlier would significantly enhance the readability and quality of the paper. At this point, the rebuttal has satisfactorily addressed my concerns. And I'll keep my score for accept recommendation.

**Limitations:**

Yes.

**Paper Formatting Concerns:**

No concerns.

**Quality:**

3

**Strengths And Weaknesses:**

Strengths:

1. The paper is generally well-written, clearly presenting most methodology details.

2. The theoritical analysis and proofs seem solid and comprehensive, effectively convering the intuition, correctness and key characteristics of the proposed method.

3. Empirical experiments on two datasets are provided to substantiate the necessity and effectiveness of the proposed method.


Weaknesses:

1. One of the central concepts, the payment rule/scheme, is clearly defined and illustrated only midway through the paper, leading to confusion when reading the introduction and related work sections.

2. Some critical experimental details, such as the temperature, K, and P parameters of the LLMs, should be explicitly stated. Additionally, although the paper checklist states "For others, we have provided clarification on the error bars," these clarifications seem not clearly identifiable. It is recommended to further clarify the different experiments conducted under deterministic and statistical settings, and explicitly report the variances from repeated trials in the statistical experiments.

3. Despite the paper checklist stating, "The full information combined with the code is provided," code access was not readily found. Providing publicly accessible code would greatly enhance reproducibility and encourage further research.

---

> ### Author Rebuttal · Authors · 2025-07-31
>
> We thank the reviewer for the insightful feedback and constructive suggestions. We address the specific points below.
>
> ---
> **Weakness 1**: About the presentation.
> > One of the central concepts, the payment rule/scheme, is clearly defined and illustrated only midway through the paper, leading to confusion when reading the introduction and related work sections.
>
> **Response**: We thank the reviewer for this helpful suggestion. We agree that adding a concise description about the **payment rule and training rule** (currently defined below Line 139) in the earlier Introduction will enhance readability. We will revise the Introduction to explicitly and concisely outline the core elements of the model, including both the training and payment mechanisms. We believe this reorganization will significantly improve the logical flow and overall readability of the paper.
>
> ---
> **Weakness 2**: About the experimental details.
> > Some critical experimental details, such as the temperature, K, and parameters of the LLMs, should be explicitly stated.
>
> **Response**: We appreciate the reviewer's emphasis on transparency in the experimental setup. The hyperparameters we use are:
>
> ``generation_kwargs = {
>  "max_new_tokens": 128 if exp_type == 'assistant' else 48,
>  "min_length": -1,
>  "do_sample": False,
>  "num_beams": 1
> }``
>
> For other parameters, we follow the default settings of `AutoModelForCausalLM.generate()`. Specifically, **the temperature is set to $1.0$, and top-K is $50$. The base model used throughout our experiments is llama-2-7b-hf.** These details will be included in the Appendix in the final version for full reproducibility.
>
> ---
> > Additionally, although the paper checklist states, "For others, we have provided clarification on the error bars," these clarifications do not seem clearly identifiable. It is recommended to further clarify the different experiments conducted under deterministic and statistical settings, and explicitly report the variances from repeated trials in the statistical experiments.
>
> **Response**: **The main experiments presented in the paper are deterministic simulations**, meaning that no randomness is introduced in the evaluation process. The results are based on model performances without training or sampling, and hence, error bars are not applicable in that context.
>
> However, **for the additional statistical experiments in Appendix F** (Table 1), we have now included standard error values. The updated table is shown below and will be included in the final version:
>
> | Reporting Parameter $\epsilon$                  |      | 0.001   | 0.002   | 0.005   | 0.01    | 0.02     | 0.05     | 0.1       |
> |-------------------------------------------------|------|---------|---------|---------|---------|----------|----------|-----------|
> | $\Delta$ Valuation (*1e2)                       | Mean | 0.1073  | 0.2096  | 0.4881  | 0.8667  | 1.3674   | 1.7978   | 1.8154    |
> | $\Delta$ Valuation (*1e2)                       | Std  | <0.0001 | <0.0001 | 0.0003  | 0.0004  | 0.0013   | 0.0026   | 0.0032    |
> | $\Delta$ Utility (= Valuation - Payment) (*1e4) | Mean | -0.1064 | -0.4135 | -2.3696 | -8.1557 | -23.7334 | -53.1552 | -55.8977  |
> | $\Delta$ Utility (= Valuation - Payment) (*1e4) | Std  | <0.0001 | 0.0001  | 0.0011  | 0.0046  | 0.0196   | 0.0573   | 0.1415    |
>
> ---
> **Weakness 3**: About the code access.
> > Despite the paper checklist stating, "The full information combined with the code is provided," code access was not readily found. Providing publicly accessible code would greatly enhance reproducibility and encourage further research.
>
> **Response**: We appreciate the reviewer's interest in reproducibility. Currently, **the code has been provided in the Supplementary Material**, along with a detailed README that explains how to run the simulations and reproduce the results. The implementation builds upon the framework introduced by Shi et al. [2], particularly the MOD mechanism.
>
> Due to the double-blind review policy, we are currently unable to release a public GitHub repository. However, we will make the code publicly available upon acceptance of the paper to ensure transparency and foster further research in this area.
>
> ---
> **The README file, which is enclosed in the Supplementary Material, is as follows:**
>
> This code implements the empirical study section of the paper *"Mechanism Design for LLM Fine-tuning with Multiple Reward Models."* The implementation builds upon *Multiple Objective RLHF* [1] and *Rewarded Soup* [2].
>
> As outlined in the paper, we model two RLHF games: the "Harmless vs. Humor" game for the Helpful Assistants task, and the "Faithful vs. Summary" game for the Reddit Summary task. This code uses the "Faithful vs. Summary" game as a case study to demonstrate the training process.
>
> #### Training Procedure
>
> 1. **Supervised Fine-Tuning**
>  First, we perform supervised fine-tuning on the base model using the corresponding dataset. To do so, run the following command:
>  ```
>  cd ./sft
>  sh sft.sh
>  ```
>  Ensure that the base model (e.g., *llama-2-7b-hf*) is placed in the `./ppo` folder.
>
> 2. **RLHF Process**
>  Next, we apply RLHF by training two models with their respective reward models. To start this process, run:
>  ```
>  cd ./ppo
>  sh ppo.sh
>  ```
>  After completion, you will find the two RLHF fine-tuned models in the `./ppo` folder.
>
> 3. **Rewarded Soup**
>  We then use the *Rewarded Soup* method to generate a set of hybrid models. Execute the following code:
>  ```
>  eval_rs_summary.sh
>  ```
>
> 4. **Simulation and Visualization**
>  The mechanism is simulated and visualised in the file `./ppo/stat.ipynb`. The figures and training logs are stored in the `./ppo` folder.
>
> #### References
> [1] Rame A, Couairon G, Dancette C, et al. Rewarded soups: towards Pareto-optimal alignment by interpolating weights fine-tuned on diverse rewards. *Advances in Neural Information Processing Systems*, 2023, 36.
>
> [2] Shi R, Chen Y, Hu Y, et al. Decoding-time language model alignment with multiple objectives. *arXiv preprint* arXiv:2406.18853, 2024.

---

> > ### Comment · Reviewer_a1Sa · 2025-08-01
> >
> > Thanks for the response and additional results. In particular, incorporating these experimental details and illustrating key concepts earlier would significantly enhance the readability and quality of the paper. At this point, the rebuttal has satisfactorily addressed my concerns.

---

> > > ### Author Response · Authors · 2025-08-06
> > >
> > > Thank you for your additional feedback. We are grateful for the chance to engage in this discussion and will integrate your suggestions into the final version of our paper. We believe these revisions will improve the clarity and quality of our work.

---

### Official Review · Reviewer_N7K8 · 2025-06-30

**Clarity:** 2
**Significance:** 2
**Originality:** 3
**Rating:** 4
**Confidence:** 2

**Summary:**

This work sought to address an important issue in LLM finetuning using human feedback data where agents may intentionally misreport their preferences to manipulate the finetuned model to their advantage. It uses a mathematical framework for investigating this issue and a mitigation strategy that adds a payment model. The theoretical results show that truthful reporting is likely sub-optimal for individual groups when only a SW-Max rule is used. It then extends the VCG payment to implement SW-Max rules in dominant-strategy incentive compatibility (DSIC). Numerical experiments are conducted to assess the proposed approach.

**Questions:**

1) In Section 4.2, it seems that the affine maximizer payment rule $p^{AFF}$ depends on true $\overrightarrow{rm}\_{i}$ and $\overrightarrow{w}\_{i}$. It is unclear how to obtain $p^{AFF}$ in real world settings when true $\overrightarrow{rm}\_{i}$ and $\overrightarrow{w}\_{i}$ are unknown.

2) What happens when agents (from all $n$ groups) misreport in terms of theoretically results and/or empirical results?

3) I have some difficulty to see the connections between the mathematical framework in this paper and what happens in real-world LLMs fine-tuning. For example, more information is needed about what types of mis-reporting in real world can be represented by the mathematical definition of misreporting in this work. Does it consider the cases a) where agents may have knowledge about the utility function (and/or other agents’ preferences) so that they can design a reporting strategy to maximize their value or b) where agents without any such information simply over-represent their preference?

**Ethical Concerns:**

["NO or VERY MINOR ethics concerns only"]

**Final Justification:**

The authors have addressed my major comments, so I raised my score accordingly.

**Limitations:**

yes.

**Quality:**

3

**Strengths And Weaknesses:**

$\textbf{Strengths:}$
This work sought to address an important issue in LLM finetuning using human feedback data where agents may intentionally misreport their preferences to manipulate the finetuned model to their advantage. It uses a mathematical framework for investigating this issue and a mitigation strategy that adds a payment model, which seems quite elegant. The theoretical results show that truthful reporting is likely sub-optimal for individual groups when only a SW-Max rule is used. It then extends the VCG payment to implement SW-Max rules in dominant-strategy incentive compatibility (DSIC). Numerical results reported in this paper show the advantage of including a payment model.



$\textbf{Weakness:}$

$\underline{Section 4}$: The theoretical results in Section 4.1 show that in the absence of additional ‘intervention’/refinement, groups have incentives to misreport their preferences under most circumstances when the SW-Max Training rule is used for fine-tuning. Are there any other options to address this issue besides adding a payment rule? If yes, it’s unclear why the authors declared that the results in Section 4.1 demonstrate necessity of payment rule. I’d suggest a) presenting more accurate discussions about the implications of the results in Section 4.1 and b) discussing in Section1 other alternative approaches vs the approach of adding a payment rule used in this work, their respective pros and cons.

In Section 4.2, it seems that the affine maximizer payment rule $p^{AFF}$ depends on true $\overrightarrow{rm}\_{i}$ and $\overrightarrow{w}\_{i}$. It is unclear how to obtain $p^{AFF}$ in real world settings when true $\overrightarrow{rm}\_{i}$ and $\overrightarrow{w}\_{i}$ are unknown. Actually, if true $\overrightarrow{rm}\_{i}$ and $\overrightarrow{w}\_{i}$ are known, then it seems that there is no need to use the proposed approach.

$\underline{Section 5}$: In the numerical experiments, how was $p^{AFF}$ estimated? In addition, I have some difficulty understanding how the misreporting strategy (2) with $\beta$ greater than 1 translates to a real-world misreporting strategy. Plus, would this strategy require the knowledge of true $\overrightarrow{rm}\_{i}$ and $\overrightarrow{w}\_{-i}$? It would be helpful if some concrete examples for misreporting can be given. For instance, one example of $\tilde{w}\_{i}=\alpha w\_{i}$ could be one agent reports its preference multiple times.

$\underline{Section 6}$ is incomplete. A discussion about limitations is included in the Appendix.

Overall, while the mathematical framework is elegant, I have some difficulty to see the connections between the mathematical framework in this paper and what happens in real-world LLMs fine-tuning. For example, more information is needed about what types of mis-reporting in real world can be represented by the mathematical definition of misreporting in this work. Does it consider the cases a) where agents may have knowledge about the utility function (and/or other agents’ preferences) so that they can design a reporting strategy to maximize their value or b) where agents without any such information simply over-represent their preference? Also, is the proposed payment strategy actually implementable without prior knowledge of the true preference/awards of agents? This disconnect could diminish the potential impact of this work.

---

> ### Author Rebuttal · Authors · 2025-07-31
>
> We thank the reviewer for the insightful feedback and constructive suggestions. We address the specific points below.
>
> ---
> **Question 1**: About the payment rule.
> > In Section 4.2, it seems that the affine maximizer payment rule depends on true $\vec{rm}$ and $\vec w$. It is unclear how to obtain in real-world settings when true $\vec{rm}$ and $\vec w$ are unknown.
>
> **Response**: We would like to respectfully clarify that in our framework, **the payment rule takes the reported information as input, not the true values.** The model's timeline is as follows:
>
> 1.  The platform publicly announces the mechanism, which includes a training rule and a payment rule.
> 2.  After observing the mechanism, the groups simultaneously report their information.
> 3.  The LLM is fine-tuned, and payment is calculated using the **reported information** based on both the training rule and the payment rule.
>
> To keep our notation concise, we opted not to introduce explicit notation (e.g., tildes such as $\widetilde{\overrightarrow{rm}}$) for reported values. We will revise the paper to make this point clearer and prevent potential confusion.
>
> Furthermore, as the reviewer highlights, the unobservability of true preferences is precisely what motivates the study of incentive compatibility in the RLHF context. We believe our characterization of DSIC from a mechanism design perspective provides valuable insights and establishes a foundation for future extensions.
>
> ---
> **Question 2**: What happens when agents (from all groups) misreport in terms of theoretical results and/or empirical results?
>
> **Response**: **The main consequence of misreporting is a reduction in the fine-tuned model's performance.**. As illustrated in the simplified example in Figure 1, when a group misreports to bias the model toward its own preferences, this results in a decline in regularized social welfare and reduced utility for other groups.
>
> Theoretically, because training is conducted on reported information while evaluation (or true welfare) is with respect to true preferences, only truthful reporting ensures optimality with respect to the social welfare objective.
>
> Empirically, we provide detailed evidence in Tables 2–6 of Appendix F (page 33), showing that misreporting leads to significant drops in the regularized social welfare, which is the objective function for fine-tuning LLM.
>
> ---
> **Question 3**: The connection between the mathematical framework and real scenarios.
> > I have some difficulty seeing the connections between the mathematical framework in this paper and what happens in real-world LLMs fine-tuning. For example, more information is needed about what types of misreporting in the real world can be represented by the mathematical definition of misreporting in this work. Does it consider the cases (a) where agents may know the utility function (and/or other agents’ preferences) so that they can design a reporting strategy to maximize their value, or (b) where agents without any such information simply over-represent their preferences?
>
> **Response**: We are happy to elaborate on this. Our framework captures both (a) and (b) styles of misreporting, and we simulate both in our experiments.
>
> * Type (a) – *Strategic Misreporting Using Other Groups' Preferences (Line \~369)*:
>  In settings with known or predictable conflicts between groups, a group may benefit from **downplaying outcomes favored by others**. For example, if two groups have preferences $r_1 = (0.7, 0.3)$ and $r_2 = (0.3, 0.7)$, the first group might report $r = \beta \times r_1 + (1 - \beta) \times r_2$ for some $\beta > 1$. For instance, with $\beta = 1.2$, the reported preference becomes $r = (0.78, 0.22)$. This type of reporting aligns with more strategic manipulation and requires some information about the opposing group. Results from our paper (Figure 1) have shown that this strategy is profitable in certain cases.
>
> * Type (b) – *Naïve Overstatement (Line \~203)*:
>   **A group may exaggerate its top preference to pull the model closer to its most favored outcome.** For instance, if true preferences are $r = (0.6, 0.3, 0.1)$, the group might report $r' = (0.6+\epsilon, 0.3, 0.1-\epsilon)$ for an $\epsilon > 0$. This strategy requires **no knowledge of other groups' preferences**. Our experimental results (Table 1, Appendix F) show that this approach can be beneficial in the absence of payments.
>
> Finally, we emphasize that DSIC ensures truthful reporting yields higher utility than any misreporting strategy, including both types (a) and (b). Given the ease of implementing these misreporting strategies, particularly type (b), our study addresses a critical challenge in the RLHF context.
>
> ---
> **Weakness in Section 4** About the declaration of "necessity of a payment rule".
> > Are there any other options to address this issue besides adding a payment rule? If yes, it’s unclear why the authors declared that the results in Section 4.1 demonstrate the necessity of the payment rule. I'd suggest (a) presenting more accurate discussions about the implications of the results in Section 4.1 and (b) discussing in Section 1 other alternative approaches vs the approach of adding a payment rule used in this work, their respective pros and cons.
>
> **Response**: We appreciate the reviewer's constructive suggestion. In our paper, **the "necessity of a payment rule" is asserted within our specific modeling framework**, as detailed in our response to **Question 1**. Within this framework, Section 4.1 demonstrates that ensuring **Dominant Strategy Incentive Compatibility (DSIC)** is nearly impossible for commonly used training objectives without the inclusion of payment rules. Thus, a payment rule becomes necessary under these assumptions.
>
> We agree that alternative paradigms are both possible and valuable. For instance, one could **design training objectives to incentivize truthful reporting instead of introducing explicit payments.** We'll expand the discussion in our related work section to include a broader perspective. For example, a recent paper by Buening et al. [1] highlights that no RLHF objective function can simultaneously guarantee strict truthfulness and social welfare optimality. They propose a pessimistic median algorithm that offers approximate truthfulness and converges to the optimal policy. While we also identify incentive issues in RLHF, our study approaches this from a mechanism design perspective, focusing on payment design to address the problem. The advantage of introducing payment is that it allows us to ensure the fine-tuned LLM remains optimal, as we don't alter the training objective. The drawback, however, is that monetary transfers may be prohibited in certain scenarios.
>
> [1] Buening, T. K., Gan, J., Mandal, D., & Kwiatkowska, M. (2025). Strategyproof reinforcement learning from human feedback. arXiv preprint arXiv:2503.09561.
>
> ---
> **Weakness in Section 5**: About the experiments.
> > In the numerical experiments, how $p^{AFF}$ was estimated?
>
> **Response**: In our empirical study, we simplify the optimization by predefining a finite model set $\Theta'$, which is composed of LLM checkpoints. All `argmax` operations in the payment formula are then computed over $\Theta'$, rather than the entire parameter space $\Theta$. This approximation is computationally feasible due to the small size of $\Theta'$. Importantly, since $\Theta'$ is predefined and independent of group reports, this simplification preserves DSIC. Additional discussion on how $\Theta'$'s expressiveness impacts DSIC and payment outcomes is provided in our response to **Question 2 from Reviewer AgWA**.
>
> ---
> **Weakness in Section 6**: About the conclusion and the future work.
>
> **Response**: We appreciate that the reviewer raises this issue. There is an error that occurs during file submission, which we will correct in the final version. Additionally, **we will include a detailed discussion on future work in the appendix. A preliminary version is outlined below:**
>
> Building on our proposed scenario and formulated model, we identify several promising directions for future research from both theoretical and empirical perspectives. First, exploring and modeling more general training rules could enhance our understanding of the framework. As noted in Appendix E, cycle monotonicity is a necessary and sufficient condition for implementability, but its validation is complex. Identifying a simpler condition to ensure implementability and investigating properties like payment equivalence for these rules are critical next steps. Second, studying preference aggregation across multiple models, particularly with diversity considerations, is a valuable direction. Third, as discussed in Section 4.4, developing mechanisms or criteria that balance computational efficiency and incentive compatibility in the RLHF Game could improve its real-world applicability. Finally, applying mechanism design theory to other large language model contexts, such as API pricing, retrieval-augmented generation (RAG), and prompt engineering, offers significant opportunities for further exploration.

---

> > ### Comment · Reviewer_N7K8 · 2025-08-04
> >
> > I want to thank the authors for addressing my comments. I don't have additional major concerns and will raise my score.

---

> > > ### Author Response · Authors · 2025-08-06
> > >
> > > Thank you for your constructive feedback and for considering an increase in our score. We are grateful for this discussion and will carefully integrate your suggestions into the final version of our paper. We believe these revisions will enhance the clarity and overall quality of our work.

---

### Official Review · Reviewer_AgWA · 2025-07-01

**Clarity:** 2
**Significance:** 2
**Originality:** 3
**Rating:** 4
**Confidence:** 2

**Summary:**

This paper investigates the incentive issues that arise when fine-tuning large language models using multiple reward models contributed by different agents. In such a scenario, agents may strategically misreport their preferences to manipulate the aggregated training objective. The authors model this as a mechanism design problem, where the LLM provider defines both a training rule and a payment rule. They show that naive fine-tuning without a payment rule is vulnerable to strategic misreporting. To address this, they propose extending the VCG mechanism to LLM fine-tuning by introducing an affine maximizer payment rule that ensures dominant-strategy incentive compatibility (DSIC) and individual rationality (IR). The authors further prove payment equivalence under certain conditions and demonstrate robustness to valuation noise.

**Questions:**

Q1: How would the proposed mechanism extend to DPO-style preference reporting, where groups only provide pairwise comparisons rather than scalar reward functions?

Q2: How sensitive is the payment outcome (and DSIC property) to approximation errors in the hybrid model set Θ′ used in simulations? Would more expressive Θ′ increase DSIC robustness?

**Ethical Concerns:**

["NO or VERY MINOR ethics concerns only"]

**Final Justification:**

I believe the authors' rebuttal has addressed most of my concerns. After carefully reviewing the interactions between the authors and other reviewers, I think a score of 4 is fair, so I will maintain my initial score.

**Limitations:**

The paper assumes access to normalized, scalar-valued reward functions. In practical RLHF scenarios, preferences are often noisy, multi-modal, or provided in pairwise form (as in DPO).

**Quality:**

3

**Strengths And Weaknesses:**

## Strength.

- **Novel formulation.**  The paper is among the first to formalize the incentive misalignment in multi-preference LLM fine-tuning as a multi-parameter mechanism design problem.
- **Theoretical rigor.**  The authors provide comprehensive theoretical analysis, including necessary and sufficient conditions for DSIC, IR, and payment equivalence, extending classical results to the context of RLHF with regularization.
- **Practical insights.**  The affine payment mechanism is empirically validated on real-world LLM fine-tuning setups (e.g., LLAMA2-7B).

---

## Weaknesses.

- **W1: No consideration for model performance degradation**

While the paper proves DSIC under the affine payment rule, it does not analyze whether this mechanism leads to degradation in final model quality compared to the no-payment baseline. There may be a trade-off between alignment fidelity and incentive robustness.

- **W2: Scalability concern for payment calculation**

Computing the affine payment requires training n+1 models (or approximating them), which can be costly when the number of agents is large. Although heuristics are proposed (Section 4.4), their theoretical and empirical impacts on DSIC guarantees are not deeply analyzed.

- **W3: Limited experiments**

The empirical experiments only simulate two-agent settings. It is unclear whether the proposed mechanism remains robust in more complex settings involving many diverse agents with conflicting preferences. And I think the current experiments are not enough to support the author's claims in total.

- **W4: Limited human-in-the-loop validation**

The framework is motivated by human feedback, but its evaluation is still largely synthetic or based on pre-collected datasets. A live human evaluation or annotation study would greatly strengthen the argument for practical effectiveness.

- **W5: Presentation issue**

The conclusion section is incomplete and fails to summarize key findings, discuss limitations, or outline directions for future work. This weakens the overall narrative of the paper and leaves the reader without a clear takeaway or understanding of the work’s boundaries.

---

> ### Author Rebuttal · Authors · 2025-07-31
>
> We thank the reviewer for the insightful feedback and constructive suggestions. We address the specific points below.
>
> ---
> **Weakness 1**: About the trade-off between alignment fidelity and incentive robustness.
>
> **Response**: We would like to clarify that **incentive robustness does not negatively impact alignment fidelity in our framework.** Alignment fidelity is determined by the alignment objective (i.e., the training rule) and the learning algorithm used to optimize it. **Our work focuses on characterizing the landscape of payment rules that implement a fixed training rule under DSIC.** These payment rules ensure that groups truthfully report their private information, and this information directly contributes to correct alignment. Since we impose no additional restrictions on either the training objective or algorithm, the resulting alignment fidelity remains unaffected.
>
> ---
> **Weakness 2**: The impacts on DSIC guarantees of the approximated payments.
> > Computing the affine payment requires training $n+1$ models (or approximating them), which can be costly when the number of agents is large. Although heuristics are proposed (Section 4.4), their theoretical and empirical impacts on DSIC guarantees are not deeply analyzed.
>
> **Response**: We appreciate this insightful question regarding the computational cost of affine payments and the theoretical implications of our proposed heuristics.
>
> First, let's clarify the affine payment structure. For group $i$, the payment $p_i$ is defined as:
> $$p_i = \max_{\theta' \in \Theta_1} \sum_{j \neq i} v_j(\theta') - \sum_{j \neq i} v_j(\theta)$$
> where $\theta = \arg\max_{\theta' \in \Theta} \sum_j v_j(\theta')$ is the model obtained from the main fine-tuning process.
>
> In our original theoretical formulation, $\Theta_1$ is set to the entire model space, i.e., $\Theta_1 = \Theta$. This full optimization guarantees DSIC and ensures payment non-negativity. However, finding an exact optimal point in the entire model space necessitates an additional training run for each group $i$, leading to $N$ additional training processes for $N$ groups. This computational burden is precisely the concern the reviewer raises.
>
> To address this, we introduced two heuristics in Section 4.4:
>
> * **Heuristic 1: Restricting $\Theta_1$ to Intermediate Models.**
>  In this heuristic, we restrict $\Theta_1 = \Theta' \subset \Theta$, where $\Theta'$ comprises intermediate models saved during the main training process that yields $\theta$. Crucially, $\theta \in \Theta_1$ is maintained, which preserves payment non-negativity. Since the models in $\Theta'$ are themselves dependent on group $i$'s report (as they are part of the main training process), DSIC is no longer strictly theoretically guaranteed. **However, we emphasize that the nature of this dependency, from group $i$'s report to the specific models stored in $\Theta'$, is highly complex and non-trivial. This complexity makes it exceptionally difficult for a group to strategically compute a profitable misreporting strategy.** While strict DSIC is not guaranteed, the practical difficulty of manipulation suggests that groups are still likely to report truthfully. This heuristic offers the significant advantage of incurring no additional training costs.
>
> * **Heuristic 2: Early-Stopped Training for $\max_{\theta' \in \Theta_1}$ Term.**
>  Here, we perform an early-stopped training to compute the maximum in the first term, defining $\Theta_1 = \Theta'' \subset \Theta$. Similar to Heuristic 1, we ensure $\theta \in \Theta''$ to maintain non-negativity. The key distinction is that this early-stopped training is specifically constructed to be **independent of group $i$'s report**. Therefore, **under this method, DSIC remains theoretically guaranteed.** This heuristic introduces a slight additional training cost, but it fully preserves the DSIC guarantee.
>
> In conclusion, Heuristic 1 offers a computationally efficient solution with no additional training, accepting a relaxation of strict DSIC due to the practical difficulty of profitable manipulation. Heuristic 2, while incurring a minor additional training overhead, fully upholds the DSIC guarantee. We believe these two heuristics offer a practical trade-off spectrum between computational efficiency and theoretical guarantees, allowing for flexibility based on application requirements.
>
> ---
> **Weakness 3**: Limited experiments.
>
> **Response**: Due to space constraints, we included only the two-group experiments in the main paper. However, **additional experiments are provided in Appendix F**:
>
> 1. We conduct a synthetic RLHF Game scenario to validate the misreporting strategy constructed in the proof of Theorem 4.2. The results show that such a strategy can improve utility when payment is absent.
> 2. Based on the dataset used in Section 5, we conduct two scenarios with more complex, multiple preferences. In that setup, there are three groups, and each group's preferences are a linear combination of different reward models.
>
> While our experimental results support the theory, the main focus of this work is **theoretical analysis**. The empirical studies are intended to illustrate and validate theoretical claims.
>
> ---
> **Weakness 4**: Limited human-in-the-loop validation.
>
> **Response**: We agree that live human evaluation would provide stronger validation. However, such evaluations are costly and extend beyond the scope of this work, which primarily emphasizes theoretical foundations. Therefore, we mainly conduct simple experiments based on the previous empirical papers on multi-objective RLHF.
>
> ---
> **Weakness 5**: About the presentation issue.
>
> **Response**: We thank the reviewer for pointing this out. The issue in the conclusion section was caused by a formatting error during upload. This will be corrected in the final version.
>
>
> ---
> **Question 1 and Limitation**: The transformation to DPO-style preference reporting.
> > How would the proposed mechanism extend to DPO-style preference reporting, where groups only provide pairwise comparisons rather than scalar reward functions?
>
> **Response**: Adapting our proposed mechanism to DPO-style preference reporting, where groups provide pairwise comparisons instead of scalar reward functions, presents interesting challenges and opportunities. Currently, we envision two primary formulations for this extension:
>
> 1. We assume that each group possesses an underlying, latent reward function, but the groups' reports are expressed as pairwise comparisons. These comparisons are then used to train a reward model for each group, and these learned reward models subsequently guide the RLHF process. This approach aligns more closely with the existing framework of our paper. Therefore, we believe our mechanism could be applicable with slight modifications, primarily by defining the payment rule with respect to these inferred reward models. The key challenge here lies in establishing an accurate and robust mapping from the pairwise comparisons to a reward model that faithfully reflects the group's true preferences, as the inferred reward model may not perfectly capture the underlying reward function.
>
> 2. In this formulation, the pairwise comparisons are used directly within the DPO training framework, without an explicit intermediate reward model. Since DPO fundamentally alters the objective landscape, our current mechanism is not directly applicable. For our mechanism to extend here, the critical step would be to reformulate how these direct pairwise comparisons translate into influence on the LLM policy in a way that allows for "charging" or incentivizing each individual comparison provided by the group. This would require a novel design for incentive compatibility in a purely comparative setting.
>
> We acknowledge that a full exploration of DPO-style formulations is beyond the scope of our current paper. However, we believe that our results can inspire and inform solutions for these extended settings, particularly by considering the two formulations outlined above. The discussion part will also be added to this paper.
>
> ---
> **Question 2**: About the impact of using an approximated $\Theta'$.
> > How sensitive is the payment outcome (and DSIC property) to approximation errors in the hybrid model set $\Theta'$ used in simulations? Would a more expressive $\Theta'$ increase DSIC robustness?
>
> **Response**: In our experiments, we primarily consider the case where $\Theta'$ is fixed and independent of groups' reports. This ensures that DSIC holds regardless of the expressiveness of $\Theta'$.
>
> The **expressiveness of $\Theta'$ mainly affects the training outcome**. Intuitively, a more expressive $\Theta'$ better approximates the original space $\Theta$, leading to a training outcome that's closer to the universally optimal one. **The choice of $\Theta'$ also affects the payment, and the direction of this impact is ambiguous.** Increasing the expressiveness of $\Theta'$ could either raise or lower payments.
>
> We also note a potential fairness concern if a platform deliberately adjusts $\Theta'$ to influence groups' payments. This highlights an important area for future investigation.

---

> > ### Comment · Reviewer_AgWA · 2025-08-06
> >
> > Thank you for your efforts in addressing my concerns. Most of my concerns have been resolved, and I will keep my score unchanged.

---

> ### Author Response · Authors · 2025-08-06
>
> Dear Reviewer AgWA,
>
> Thank you once again for your time and effort in providing valuable feedback on our paper. As the author-reviewer discussion phase is approaching its end, we would like to confirm whether we have adequately addressed your concerns, or at least some of them. In particular, **we have expanded our explanations regarding the weaknesses you highlighted and provided additional discussion in response to your insightful questions.**
>
> Should any questions remain or if further clarification is needed, please do not hesitate to let us know. Your feedback is valuable for the further improvement of the quality of our paper. If you are satisfied with our responses, we would greatly appreciate your consideration in adjusting the evaluation scores accordingly.
>
> We sincerely look forward to your feedback.

---

### Official Review · Reviewer_gqdi · 2025-07-03

**Clarity:** 3
**Significance:** 3
**Originality:** 3
**Rating:** 4
**Confidence:** 3

**Summary:**

This paper explores the setting where an LLM provider offers the service to fine-tune a base LLM in accordance with the preferences of a number, $n$, of groups each comprising some (perhaps different) number of agents. In this setting, each group provides a reward model, $\mathrm{rm}_i$ which takes as input a prompt response pair and returns a number quantifying the group's satisfaction with the response according to their preferences. The agent's valuation of the given model is thus its expected reward across the prompt-response pairs. The provider thus wishes to fine-tune the model in accordance with the preferences of the groups.

In this context, the provider first announces the training rule and the payment rule. Then, the groups report their desired reward model and group size. The provider then fine-tunes the model and charges each group according to the payment rule. It is assumed that each group assesses the utility of the model as the difference between the valuation (times the number of individuals) minus the payment. The paper studies the possibility for groups to misreport their reward model or number of individuals in order to increase their benefit at the behest of the aggregate performance.

Towards this end, it is illustrated that, in the absence of a payment rule, there exists a misreporting strategy which yields a strictly higher utility regardless of the other groups' choices in most cases of interest. Next, it is shown that the so-called affine maximizer payment rule satisfies the propert that  truthfully reporting one's reward model will yield the highest utility regardless of the other group's choices, a property called dominant-strategy incentive compatibility (DSIC).  Next, for continuous training rules, a
connection between all payment rules satisfying the DSIC property is derived. Finally, the effect of noise on the reported reward model is studied and a bound on the benefit under the affine maximizer payment rule is derived.

The paper concludes with some implementation details and an empirical study which serves to validate the theoretical results. Notably, it is shown that, without a payment structure misreporting can lead to a higher utility and that, under the affine maximizer payment rule, misreporting is not beneficial.

**Questions:**

1. It is assumed throughout that the groups have quasi-linear utilities which does not appear to be a realistic assumption in general. Indeed, if a group has a much larger budget they may be willing to trade off a lower utility for a model that better matches their preferences by increasing their payment..

2, Related to the above point, I wonder if the notion of DSIC is a realistic aim to pursue. Indeed, it appears that, under DSIC, the underlying model cannot be modified too much in response to a group's preferences perhaps even if there is a consensus between multiple groups. Is this impression correct?

3. There are a few small issues with the writing in the current version. For instance, the statement of Theorem 4.3 does not appear to be complete. Similarly, the last line of the paper is not complete. I highly recommend that the author's give the paper a few more passes to improve clarity.

**Ethical Concerns:**

["NO or VERY MINOR ethics concerns only"]

**Final Justification:**

The authors have adequately answered the questions I had for this submission and alleviated my concern regarding the DSIC property. I believe that these points should have been addressed more clearly in the original submission. I maintain the initial score.

**Limitations:**

yes

**Quality:**

3

**Strengths And Weaknesses:**

Overall, I find the paper to be generally well written. The submission addresses an important question when aggregating user's preferences and provides some principled theoretical results. While the experimental validation of these results is somewhat limited in scope, I see the main results of this paper to be at the theoretical level.

My main questions about this paper revolve around some of the assumptions which I believe to be limiting, see the questions below.

---

> ### Author Rebuttal · Authors · 2025-07-31
>
> We thank the reviewer for the insightful feedback and constructive suggestions. We address the specific points below.
>
> ---
> **Weakness 1**: About the assumption on the quasi-linear utility.
> > It is assumed throughout that the groups have quasi-linear utilities, which does not appear to be a realistic assumption in general. Indeed, if a group has a much larger budget, they may be willing to trade off a lower utility for a model that better matches their preferences by increasing their payment.
>
> **Response**: We understand the reviewer's concern regarding the assumption of quasi-linear utility, and we appreciate the opportunity to clarify our modeling choices. There has, in fact, been a long debate in the economic literature on which utility function to use. We adopted quasi-linear utility because it is **more common in mechanism design theory, and we believe it is suitable for this initial work studying this scenario.**
>
> We acknowledge that in certain scenarios, groups might focus solely on the final outcome regardless of payment. **This case also induces strategic misreporting, as it is exactly the same as a mechanism without payment.** Our Section 4.1 discusses strategic behavior under such mechanisms, and the results (Theorems 4.2 and 4.3) show that commonly used objectives, like SW-Max training rules, cannot guarantee truthfulness. Further exploration in this direction could consider the design of training objectives for RLHF, which is an interesting and valuable avenue for future work. Overall, we believe the results presented in this paper serve as a crucial first step in this area, offering valuable insights that can inform these subsequent extensions.
>
> ---
> **Weakness 2**: About the DSIC property.
> > It appears that, under DSIC, the underlying model cannot be modified too much in response to a group's preferences, perhaps even if there is a consensus between multiple groups.
>
> **Response**: We appreciate the reviewer's comment regarding the DSIC property and its perceived implications for model modification. We would like to clarify that the DSIC property **does not restrict how much the LLM policy can be modified** in response to group preferences.
>
> The extent to which the LLM policy shifts is **determined solely by the training rule**, not by the payment rule. Our theoretical analysis focuses on characterizing the payment rules that can implement DSIC for fixed training rules. Consequently, our results don't involve altering the training objective or the training process itself. The DSIC property simply ensures that groups are incentivized to report their preferences truthfully.
>
> For instance, consider the **SW-Max training rule** defined in our paper. This includes a **regularization penalty** to constrain deviations from the initial model, a common practice in RLHF. In such a case, even if multiple groups reach a consensus, the regularization, as part of the training rule, might limit how much the LLM is modified. Crucially, within our framework, the choice of payment rule (and whether the mechanism is DSIC) does not affect the learned model.
>
> ---
> **Weakness 3**: About Theorem 4.3. and the Conclusion.
> > There are a few small issues with the writing in the current version. For instance, the statement of Theorem 4.3 does not appear to be complete. Similarly, the last line of the paper is not complete. I highly recommend that the authors give the paper a few more passes to improve clarity.
>
> **Response**: We thank the reviewer for highlighting these important issues.
>
> Regarding Theorem 4.3, we intentionally presented a simplified version in the main text to enhance readability and convey the core conceptual insight more effectively. This simplified statement aims to communicate the intuition that the SW-Max training rule is susceptible to strategic misreporting. **The full technical version of this theorem, including the concrete form of the function $t(\cdot)$, is provided as Theorem B.2 in the appendix.**
>
> Theorem 4.3 is stated as follows.
> >In the RLHF Game with mechanism $(\theta^\ast, p)$ that $\theta^\ast \in \Psi^{SW}$ and $p \equiv 0$, when $f$ is strongly convex and $C^2$-smooth, there exists a function $t$, when $t({x}, \overrightarrow{\mathrm{rm}}, \vec w, \theta_{\text{init}}) \neq 0$ for some ${x} \in T^{\ast}$, truthfully reporting is not the optimal strategy.
>
> Theorem B.2 is stated as follows.
> >In the RLHF Game with mechanism $(\theta^\ast, p)$ that $\theta^\ast \in \Psi^{SW}$ and $p \equiv 0$, when $f$ is $\alpha$-strongly convex and $C^2$-smooth, suppose group $i$ has preference $\mathrm{rm}\_i$ and group size $w_i$, other groups report $(\overrightarrow{\mathrm{rm}}\_{-i}, \vec w\_{-i})$ and the initial model $\theta\_{\text{init}}$, we define
> $$
> t({z}) := \sum_{{x} \in T^*} \frac{(\mathrm{rm}\_i({z}) - \mathrm{rm}\_i({x})) \mathrm{llm}\_{\text{init}}({x})}{f''\left(\frac{\mathrm{llm}({x})}{\mathrm{llm}\_{\text{init}}({x})} \right)},
> $$
> in which $\theta = \psi(\overrightarrow{\mathrm{rm}}, \vec w, \theta_{\text{init}})$. When $s_i \geq 2$ and $\underline{\mathrm{rm}_i} = 0$:
> > - For the maximum normalization case, if there exist ${x}_1 \in T^*$, $t({x}_1) \neq 0$ and $0 < \mathrm{rm}\_i({x}_1) < 1$, truthful reporting is not the optimal strategy.
> > - For the summation normalization case, if there exist ${x}_1 \in T^*$, $t({x}_1) < 0$ and $0 < \mathrm{rm}\_i({x}_1) < 1$, truthful reporting is not the optimal strategy.
> > - For the summation normalization case, if there exist $x_1 \in T^\ast$, $t(x_1) > 0$ and we can also find ${x}_2 \in T^*$, such that $1 > \mathrm{rm}\_i({x}_1) \geq \mathrm{rm}\_i({x}_2) > 0$ and $\frac{1}{\mathrm{llm}\_{\text{init}}({x}_1)} f''\left(\frac{\mathrm{llm}({x}_1)}{\mathrm{llm}\_{\text{init}}({x}_1)}\right) < \frac{1}{\mathrm{llm}\_{\text{init}}({x}_2)} f''\left(\frac{\mathrm{llm}({x}_2)}{\mathrm{llm}\_{\text{init}}({x}_2)}\right)$, truthful reporting is not the optimal strategy.
>
> To further improve clarity and prevent any misunderstanding, we will modify the title of Theorem 4.3 in the main text to explicitly state its relationship to Theorem B.2 in the appendix.
>
> Concerning the conclusion, we acknowledge that an error occurred during the manuscript upload process, resulting in an incomplete final sentence. We assure the reviewer that this will be corrected in the final version.

---

> > ### Comment · Reviewer_gqdi · 2025-08-04
> >
> > I thank the authors for clarifying the points I raised and for correcting my concern regarding the DSIC property.
> >
> > I believe that the added discussion will improve the overall quality of the paper, but should have been included in the original submission. As such, I will maintain the current score.

---

> > > ### Author Response · Authors · 2025-08-06
> > >
> > > Thank you for your additional feedback. We appreciate this discussion and will carefully integrate your valuable suggestions into the final version of our paper. We believe these revisions will improve the clarity and quality of our work. Additionally, we are open to any further questions or concerns you may have.

---

### Note · Authors · 2025-08-12

We sincerely thank all reviewers for their constructive feedback and engaging discussions during the rebuttal phase. Particularly, we appreciate that **all reviewers** confirmed our rebuttal effectively addressed their concerns and provided positive scores at the end of the discussion period.

We are encouraged by the reviewers' **recognition of the strength of our theoretical results and the significance of the problem we address**. This paper explores incentive issues in RLHF with multiple strategic preference providers. As misreporting can undermine training performance, mitigating such behavior is critical. We frame the problem within a mechanism design framework and demonstrate that commonly used RLHF objectives are susceptible to strategic manipulation, a finding validated through simulations on real RLHF outcomes and synthetic scenarios. Our main theoretical contribution characterizes the complete set of payment rules that ensure truthful reporting for widely used training objectives. This also highlights inherent trade-offs between certain game-theoretic desiderata and efficiency.

During the discussion period, we received **valuable suggestions from reviewers**. Reviewers gqdi, AgWA, and N7K8 raised concerns about the clarity of the Conclusion section. Reviewer N7K8 recommended adding intuition for key mathematical formulas to enhance readability. Reviewer a1Sa suggested including error bars for Table 1 in Appendix F and encouraged code publication upon acceptance. We are committed to incorporating these suggestions into the final version of the paper. As these revisions **do not alter the core contributions and were addressed in the rebuttal**, we are confident the final version will fully resolve all reviewer concerns.

We again express our deep gratitude to the reviewers and Area Chairs for their time, insights, and constructive engagement.

---

### Decision · Program_Chairs · 2025-09-17

**Decision:**

Accept (poster)

**Comment:**

This paper studies a very natural problem, that should be relevant to the NeurIPS community. It studies the following mechanism design problem: There are n groups with sizes w_i, and preferences rm_i mapping (prompt, reply)-pairs of an LLM to a value. The goal is to fine-tune a central LLM to these groups and preferences, with particular focus on SW-Max Training Rules which maximize a weighted sum of the preferences minus a regularization term (based on the divergence of the initial LLM and the fine-tuned LLM). The paper argues that quite generically payment-free models are susceptible to manipulation, it proposes an approach based on affine-maximization to determine payments that render truthful reporting a dominant (and IR) strategy. It then continues to examine conditions under which payment equivalence is obtained, and shows that under small noise the mechanism's incentive properties are preserved.

Overall, I think this is a worthy candidate for being accepted to NeurIPS (all reviewers have a score of 4), and I think this is a fair evaluation. The paper isn't rocket science from a mechanism design perspective, but I see clear value in formulating the problem in the right language accessible to the broader ML and NeurIPS audience. The paper also nicely pinpoints ways to manipulate system w/o payment (which should already be a useful contribution in and by itself). Wearing my AGT head, I also appreciate some some of the results that go beyond mere application of what's known in the mechanism design literature. I think this might become an influential Mechanism Design for LLMs paper.

----
Comment: There seems to be a rather closely related paper by Buening et al. "Strategyproof Reinforcement Learning
from Human Feedback" (https://arxiv.org/pdf/2503.09561). The authors websites don't mention whether this paper has already been published, it may have been submitted to NeurIPS as well. The current paper is not yet citing it, but the authors brought it up in the discussions and may wish to discuss it in the final version.